# WHEN SCORES LEARN GEOMETRY: RATE SEPARATIONS UNDER THE MANIFOLD HYPOTHESIS

**Xiang Li, Zebang Shen, Ya-Ping Hsieh & Niao He**
Department of Computer Science
ETH Zurich
8092 Zurich, Switzerland
{xiang.li,zebang.shen,yaping.hsieh,niao.he}@inf.ethz.ch

## ABSTRACT

Score-based methods, such as diffusion models and Bayesian inverse problems, are often interpreted as learning the **data distribution** in the low-noise limit ($\sigma \to 0$). In this work, we propose an alternative perspective: their success arises from implicitly learning the **data manifold** rather than the full distribution. Our claim is based on a novel analysis of scores in the small-$\sigma$ regime that reveals a sharp **separation of scales**: *information about the data manifold is $\Theta(\sigma^{-2})$ stronger than information about the distribution*. We argue that this insight suggests a paradigm shift from the less practical goal of distributional learning to the more attainable task of **geometric learning**, which provably tolerates $O(\sigma^{-2})$ larger errors in score approximation. We illustrate this perspective through three consequences: i) in diffusion models, concentration on data support can be achieved with a score error of $o(\sigma^{-2})$, whereas recovering the specific data distribution requires a much stricter $o(1)$ error; ii) more surprisingly, learning the **uniform distribution** on the manifold—an especially structured and useful object—is also $O(\sigma^{-2})$ easier; and iii) in Bayesian inverse problems, the **maximum entropy prior** is $O(\sigma^{-2})$ more robust to score errors than generic priors. Finally, we validate our theoretical findings with preliminary experiments on large-scale models, including Stable Diffusion.

## 1 INTRODUCTION

*Score learning* has emerged as a particularly powerful paradigm for modeling complex probabilistic distributions, driving breakthroughs in generative modeling, Bayesian inverse problems, and sampling (Laumont et al., 2022; Saremi et al., 2023; Ho et al., 2020; Song & Ermon, 2019; Song et al., 2021). Let $\mu_{\text{data}}$ be a data measure over $\mathbb{R}^d$ and define a Gaussian-smoothed measure as

$$\mu_\sigma := \text{law}\left(X + \sigma Z\right) \text{ or } \mu_\sigma := \text{law}\left(\sqrt{1 - \sigma^2}\, X + \sigma Z\right), \text{ where } X \sim \mu_{\text{data}}, Z \sim \mathcal{N}(0, I). \quad (1)$$

Let $p_\sigma$ be its density function w.r.t. the Lebesgue measure over $\mathbb{R}^d$. A key step in the score learning framework is to approximate the score function $\nabla \log p_\sigma$ and to sample from the target distribution $\mu_\sigma$, possibly across a spectrum of different $\sigma$ values (Vincent, 2011; Hyvärinen & Dayan, 2005).

A central challenge in this framework is understanding the *low-temperature limit*, i.e., learning the score of $\mu_\sigma$ as $\sigma \to 0$, which encodes the most detailed information about the data distribution. Empirically, this regime is also the most valuable: low-temperature scores underpin many probabilistic learning frameworks (Laumont et al., 2022; Saremi et al., 2023; Janati et al., 2024; Kadkhodaie & Simoncelli, 2020), including the influential diffusion model framework (Ho et al., 2020; Song et al., 2020; Karras et al., 2022), whose noise schedules are specifically designed to emphasize low temperatures and often require substantial post-training engineering to stabilize the learned scores.

Despite its importance, accurately estimating the score function in the low-$\sigma$ regime remains notoriously difficult (Song et al., 2021; Karras et al., 2022; Arts et al., 2023; Raja et al., 2025; Stanczuk et al., 2024). Motivated by this challenge, this paper establishes a new qualitative phenomenon under the widely adopted *manifold hypothesis*, which posits that the data distribution $\mu_{\text{data}}$ is supported on a low-dimensional manifold $\mathcal{M}$ embedded in a high-dimensional ambient space.

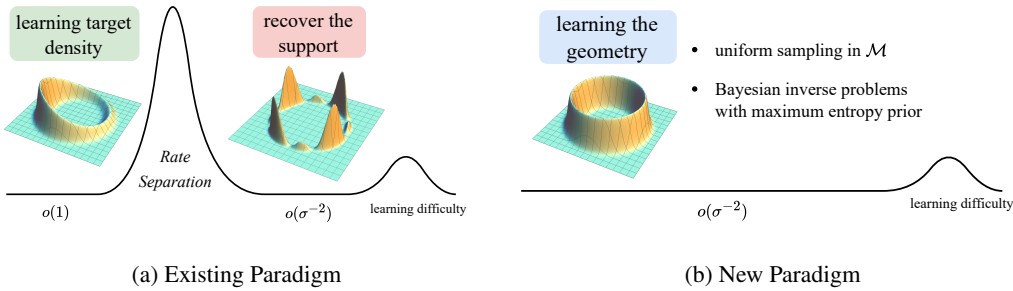

(a) Existing Paradigm                    (b) New Paradigm

Figure 1: Toy examples illustrating recovered distributions under different regimes, with the manifold represented as a one-dimensional circle embedded in $\mathbb{R}^2$.

Our key finding, formalized in Theorem 3.1, is that in the small-$\sigma$ regime of score learning there is a **sharp separation of scales**: *geometric information about the data manifold appears at order* $\Theta(\sigma^{-2})$, *whereas density information of* $\mu_{\text{data}}$ *emerges only at order* $\Theta(1)$. As shown in Section 3, this implies that distribution learning of $\mu_\sigma$ (e.g., in diffusion models) **necessarily** first recovers the support of the data distribution before any information about the density can be learned. This perspective naturally separates score learning into two fundamental tasks: *geometric learning*, which targets the manifold geometry, and *density learning*, which targets the specific data density on that manifold, with the latter being order of magnitude more difficult. It also suggests that the practical success of score-based models (e.g., diffusion models) stems from constraining generated samples to the manifold, thereby producing realistic data even without fully recovering the underlying distribution. According to our analysis, to achieve this, a score error even as large as $o(\sigma^{-2})$ is sufficient.

However, our analysis reveals a critical limitation: unless the score is learned to a stringent accuracy that is beyond $O(1)$, attempts to recover the data distribution may yield *arbitrary* densities supported on the manifold. This amounts to only a partial recovery of geometry and can compromise the reliability of downstream tasks and analyses. Such an observation motivates us to pursue *full geometric learning*—that is, learning to sample *uniformly* with respect to the manifold's intrinsic (Riemannian) volume measure, as it is well-known that uniform samples can best support tasks that depend solely on the underlying geometry (e.g., Laplace–Beltrami and heat-kernel approximation, geodesic and diffusion distances) (Coifman & Lafon, 2006; Belkin & Niyogi, 2008; Jost, 2005). In addition, they also facilitate principled manifold exploration, yielding diverse samples while mitigating potential biases present in $\mu_{\text{data}}$ (De Santi et al., 2025).

In this light, a central contribution of this work is to show that a simple, one-line modification to standard algorithms can *provably* generate the *uniform distribution* on the manifold—requiring only $o(\sigma^{-2})$ score accuracy, in stark contrast to the $o(1)$ accuracy needed for exact distributional recovery. In summary, we advocate a paradigm shift: from the demanding goal of *distributional learning* toward the more practical and robust objective of *geometric learning*.

We substantiate the aforementioned rate separation phenomenon by three key results (see also Figure 1):

- Theorem 4.1 shows that, in existing frameworks, the score accuracy required to force concentration on the data manifold is $O(\sigma^{-2})$ weaker than that needed to exactly recover $\mu_{\text{data}}$. Nevertheless, the resulting distribution can still be *arbitrary*.

- In contrast, Theorems 5.1 to 5.2 establish a new paradigm centered on extracting precise *geometric* information of the data manifold by producing the *uniform distribution*. Notably, we show that a simple one-line modification of a widely used sampling algorithm suffices to obtain samples from the uniform distribution under the relaxed score error condition $o(\sigma^{-2})$, substantially weaker than the $o(1)$ required for full recovery of $\mu_{\text{data}}$.

- In the context of Bayesian inverse problems (Venkatakrishnan et al., 2013), Theorem 6.1 establishes a rate separation in posterior sampling depending on the choice of prior. When the prior is uniform, posterior sampling requires only $o(\sigma^{-2})$ score accuracy. By contrast, when the prior is taken to be the commonly used data distribution $\mu_{\text{data}}$, substantially stronger accuracy guarantees are needed to ensure provable success in existing works (Laumont et al., 2022; Pesme et al., 2025).

We validate these theoretical results with preliminary experiments on both synthetic and real-world data, including an application of our algorithm to a large-scale image generation model (Stable Diffusion 1.5 (Rombach et al., 2022)).

## 1.1 RELATED WORK

**Diffusion models for distribution learning.** Prior theory shows that diffusion/score-based samplers converge to the target law when the learned score is accurate, with error bounds that scale directly with the score mismatch (De Bortoli, 2022; Chen et al., 2023; Lee et al., 2023); related works study other factors such as dimension dependence (Azangulov et al., 2024; Tang & Yang, 2024). However, these results do not separate geometry from density in the score error but instead consider them together, therefore they do not imply any scale separation.

**Diffusion models detect data manifold.** There is a growing body of work probing whether diffusion models learn the full data distribution or primarily the underlying low-dimensional manifold. A number of studies suggest that these models often capture the data support while missing fine-grained distributional structure. However, these results are obtained under restricted settings: Stanczuk et al. (2024) focuses on estimating the intrinsic dimension of the data manifold; Ventura et al. (2024) analyzes only linear manifolds (linear subspaces); and Pavlova & Wei (2025) provides primarily empirical evidence. Pidstrigach (2022) establishes sufficient regularity conditions under which high-accuracy scores concentrate mass near the manifold, but does not address how approximation errors scale with $\sigma$ and therefore does not reveal a separation of scales. By contrast, our analysis quantifies how inaccuracies in the learned score propagate differently to geometry versus distribution learning, exhibiting distinct error rates that lead to a sharp scale separation in the small-$\sigma$ regime. Furthermore, prior work does not address full geometric recovery via uniform sampling.

**Asymptotic behavior of the score.** It is established that under the manifold hypothesis, the score function develops a singularity in the small-noise regime, becoming orthogonal to the data manifold. Recent works characterize this behavior mathematically, showing that the score effectively acts as a geometric projection operator onto the manifold (Lu et al., 2023; Lyu et al., 2025; Liu et al., 2025). This aligns with the leading-order term in our expansion (Equation (6)), which governs geometric concentration. However, these analyses generally subsume the distributional information into a generic bounded remainder term. Crucially, they do not explicitly isolate the higher-order terms involving $p_{\text{data}}$ and thus do not characterize the separation between geometry and density. Our analysis reveals that these missing terms are not merely residuals but are essential for establishing the rate separation between recovering the manifold support and learning the underlying density.

**Uniform sampling on manifolds.** Classical approaches achieve uniform-on-manifold sampling via graph-based normalizations that cancel the sampling density so that the limiting operator is the Laplace–Beltrami operator (Coifman & Lafon, 2006; Hein et al., 2007). While foundational, these methods are designed to approximate geometric operators from neighborhood graphs and do not readily scale to high-dimensional, large-scale generative modeling. Recently, De Santi et al. (2025) proposed fine-tuning diffusion models to produce uniform samples. In contrast, our approach operates entirely at inference time, achieving uniform sampling without the cost of fine-tuning.

## 2 PRELIMINARIES AND NOTATION

In this work, we adopt the manifold assumption (Song & Ermon, 2019; De Bortoli, 2022; Loaiza-Ganem et al., 2024) as follows:

**Assumption 2.1** (The Manifold Hypothesis). *We assume that the data distribution $\mu_{\text{data}}$ is supported on a compact, boundaryless $C^4$ embedded submanifold $\mathcal{M} \subset \mathbb{R}^d$, with $\dim(\mathcal{M}) = n$.*

**Local coordinates and manifold geometry.** Under the manifold hypothesis, the $n$-dimensional manifold $\mathcal{M}$ can be described locally using coordinates from a flat, Euclidean space. This is done via a set of smooth mappings, or charts, $\Phi : U \to \mathcal{M}$, where each chart maps an open set of parameters $U \subset \mathbb{R}^n$ to a patch on the manifold. For notational simplicity, we will work with a single chart, where $u \in U$ represents the local coordinates of a point $\Phi(u)$ on $\mathcal{M}$. The manifold's intrinsic,

and generally non-Euclidean, geometry is captured by the Riemannian metric tensor, $g(u)$. This tensor provides the means to measure lengths and angles on the curved surface. The metric gives rise to the Riemannian volume measure, $d\mathcal{M}(x)$, which is the natural way to integrate a function $f : \mathcal{M} \to \mathbb{R}$ over the manifold. In local coordinates, this integral is expressed as $\int_{\mathcal{M}} f(x)\, d\mathcal{M}(x) = \int_U f(\Phi(u)) \sqrt{\det(g(u))}\, du$, w.r.t. the Lebesgue measure on $U$. Here, the term $\sqrt{\det(g(u))}$ is the volume correction factor. While we use a single chart for clarity, integration over the entire compact manifold is handled by stitching together multiple charts via a partition of unity. The set of points in $\mathbb{R}^d$ that are sufficiently close to the manifold forms the tubular neighborhood: $T_{\mathcal{M}}(\epsilon) := \{x \in \mathbb{R}^d : \mathrm{dist}(x, \mathcal{M}) < \epsilon\}$. For any point $x$ within this neighborhood, there exists a unique closest point on the manifold, given by the $P_{\mathcal{M}}(x) : T_{\mathcal{M}}(\epsilon) \to \mathcal{M}$. This projection allows us to define the squared distance function to the manifold, a quantity of central importance to our analysis:

$$d_{\mathcal{M}}(x) := \frac{1}{2}\mathrm{dist}^2(x, \mathcal{M}) = \min_{\bar{x} \in \mathcal{M}} \frac{1}{2}\|x - \bar{x}\|^2. \tag{2}$$

Further details and notations regarding the manifold hypothesis are provided in Appendix A.

## 2.1 THE GAUSSIAN SMOOTHED MEASURE AND CONNECTION TO DIFFUSION MODELS

With Assumption 2.1, we define the corresponding density $p_{\mathrm{data}}$ of $\mu_{\mathrm{data}}$ with respect to the Lebesgue measure on $U$: $p_{\mathrm{data}}(u) := \frac{d(\Phi^* \mu_{\mathrm{data}})}{du}(u)$, where $\Phi^* \mu_{\mathrm{data}}(S) := \mu_{\mathrm{data}}(\Phi(S))$ for $S \subseteq U$, and assume the following regularity assumption:

**Assumption 2.2** (Regularity and Converage of $p_{\mathrm{data}}$). *The probability density $p_{\mathrm{data}} : U \to \mathbb{R}$ defined w.r.t. the Lebesgue measure on $U$ is $C^1(U)$ and strictly positive.*

Recall the two Gaussian–smoothed measures $\mu_\sigma$ introduced in Equation (1). We follow the naming convention of Song et al. (2021) and denote by $\mu_\sigma^{\mathrm{VE}}$ the variance–exploding (VE) smoothing and by $\mu_\sigma^{\mathrm{VP}}$ the variance–preserving (VP) smoothing. Their densities w.r.t. the Lebesgue measure on $\mathbb{R}^d$ is

$$p_\sigma(x) := \int_{\mathcal{M}} \frac{1}{(2\pi\sigma^2)^{d/2}} \exp\left(-\frac{\|x - \gamma(\sigma)\Phi(u)\|^2}{2\sigma^2}\right) p_{\mathrm{data}}(u)\, du, \tag{3}$$

where the densities are denoted $p_\sigma^{\mathrm{VE}}$ for VE with $\gamma(\sigma) = 1$ and $p_\sigma^{\mathrm{VP}}$ for VP with $\gamma(\sigma) = \sqrt{1-\sigma^2}$. We take $p_{\mathrm{data}}$ to be the true population density rather than a finite-sample empirical approximation.

These smoothed distributions correspond to the marginals of the forward noising processes used in diffusion and score-based generative modeling. In SMLD or VE-SDE (Song et al., 2021), Gaussian noise with variance $\sigma^2(t) : \mathbb{R}_+ \to \mathbb{R}_+$ is added to the data at time $t$, a model is trained to progressively denoise, and in the reverse process the objective is to sample from $p_{\sigma(t)}^{\mathrm{VE}}$, recovering $p_{\mathrm{data}}$ as $t \to 0$ (equivalently, $\sigma(t) \to 0$). Similarly, DDPM or VP-SDE (Ho et al., 2020; Song et al., 2021) corresponds to the VP density $p_{\sigma(t)}^{\mathrm{VP}}$, again with the goal of recovering $p_{\mathrm{data}}$ in the limit $t \to 0$. Beyond the reverse process, one may also directly use the learned score to run a Langevin sampler targeting $p_\sigma^{\mathrm{VE}}$ (Song & Ermon, 2019) or $p_\sigma^{\mathrm{VP}}$, or combine Langevin sampling with the reverse process, as in the Predictor–Corrector algorithm (Song et al., 2021). Since our results apply to both VE and VP settings, we adopt the unified notation $p_\sigma$ whenever no ambiguity arises.

## 2.2 BAYESIAN INVERSE PROBLEMS

Another important algorithmic implication of our results concerns Plug-and-Play (PnP) methods for Bayesian inverse problems (Venkatakrishnan et al., 2013). Let $x \in \mathbb{R}^d$ be the latent signal and $y \in \mathcal{Y} \subseteq \mathbb{R}^m$ the observation $y = A(x) + \xi$, where $A : \mathbb{R}^d \to \mathbb{R}^m$ is the measurement map and $\xi \in \mathbb{R}^m$ is noise. Under standard assumptions on $A$ and $\xi$ (e.g., $A$ linear, $\xi \sim \mathcal{N}(0, s^2 I)$), the likelihood admits a density $p(y \mid x) \propto \exp(-v(x; y))$ (for the Gaussian case, $v(x; y) = \frac{1}{2s^2}\|A(x) - y\|^2$). In the Bayesian framework we endow $x$ with a prior $p_{\mathrm{prior}}$. Inference is cast as sampling from the posterior $p(x \mid y) = p(y \mid x)\, p_{\mathrm{prior}}(x) / \int p(y \mid \bar{x})\, p_{\mathrm{prior}}(\bar{x}) d\bar{x}$.

**Plug-and-Play (PnP).** PnP methods address the case where the prior is (i) known up to a normalizing constant, e.g. a Gibbs measure or (ii) only accessible via samples (common in ML). A unifying sampling paradigm is posterior Langevin with a (possibly learned) prior score $\hat{s} \simeq \nabla \log p_{\mathrm{prior}}$,

$$dX_t = -\nabla_x v(X_t; y)\, dt + \hat{s}(X_t)\, dt + \sqrt{2}\, dW_t. \tag{4}$$

In case (ii), $\hat{s}$ is a score estimator obtained, e.g., by score matching on prior samples. A common choice of $p_{\text{prior}}$ would be the density $p_\sigma$ defined in eq. (3) with small $\sigma$. In this context, to ensure update (4) yields samples matching the target posterior distribution, existing works require the learned score $\hat{s}$ to be at least $o(1)$ accurate (Laumont et al., 2022), or even exact (Pesme et al., 2025).

## 2.3    STATIONARY DISTRIBUTION FOR NON-REVERSIBLE DYNAMICS

In score learning, one typically learns a score function $s(x, \epsilon)$ for a target density and then runs Langevin dynamics (equivalently, the corrector step in the Predictor–Corrector algorithm for diffusion models (Song et al., 2021)) until near stationarity to sample from that density:

$$dX_t = s(X_t, \epsilon)\, dt + \sqrt{2}\, dW_t.$$

If $s(x, \epsilon) = -\nabla f_\epsilon(x)$, the stationary distribution is proportional to $\exp(-f_\epsilon(x))$. In practice, however, the score is often produced by a parameterized model and need not be a gradient field (this is also the case for our proposed algorithms). The resulting Langevin dynamics is then generally *non-reversible*, and its stationary distribution need not admit a closed form—an open problem studied in, e.g., (Graham & Tél, 1984; Maes et al., 2009; Rey-Bellet & Spiliopoulos, 2015).

Several works have sought to characterize the stationary distribution of non-reversible SDEs. Notably, Matkowsky & Schuss (1977); Maier & Stein (1997); Graham & Tél (1984); Bouchet & Reygner (2016) employ the WKB ansatz (Wentzel, 1926; Kramers, 1926; Brillouin, 1926), which is commonly used in matched asymptotic expansions (Holmes, 2012). This approach posits that the stationary density takes the form

$$\exp\left(-\frac{V(x)}{\epsilon}\right) c_\epsilon(x), \quad \text{with} \quad c_\epsilon(x) = \sum_{i=0}^{k} c_i(x)\, \epsilon^i, \tag{5}$$

for some $k \in \mathbb{N}$. The functions $V$ and $\{c_i\}$ are then identified by inserting (5) into the stationary Fokker–Planck equation and balancing terms order by order in $\epsilon$. Importantly, prior analyses typically focus on low-dimensional special examples or on drifts with a *single* stable point. The difficulty of removing such restrictions turn out to be central to our analysis; see Section 5 for details.

## 3    CENTRAL INSIGHT: GAUSSIAN SMOOTHING RECOVERS GEOMETRY BEFORE DISTRIBUTION

This section presents the central insight of the paper: While the proofs of our later main results are technically involved, they are all guided by a common intuition that is transparent and can be understood through a simple Taylor expansion of $\log p_\sigma$ at $\sigma = 0$:

**Theorem 3.1** (Informal Theorem B.2). *Assume Assumptions 2.1 and 2.2 holds. For any $x \in T_{\mathcal{M}}(\epsilon)$,*

$$\log p_\sigma(x) = -\frac{1}{\sigma^2} d_{\mathcal{M}}(x) + \log p_{\text{data}}(\Phi^{-1}(\mathrm{P}_{\mathcal{M}}(x))) - \frac{d-n}{2}\log(2\pi\sigma^2) + H(x) + o(1), \tag{6}$$

*where $H(x)$ contains the curvature information of the manifold and $\epsilon$ is some sufficiently small constant; both of them are independent of $\sigma$. The small $o(1)$ term is uniform for $x \in T_{\mathcal{M}}(\epsilon)$.*

From Equation (6), it follows immediately that the scaled log-density recovers the distance function to the manifold in the small $\sigma$ limit:

$$\lim_{\sigma \to 0} \sigma^2 \log p_\sigma(x) = -d_{\mathcal{M}}(x) \quad \text{uniformly for all } x \in T_{\mathcal{M}}(\epsilon). \tag{7}$$

The appearance of $d_{\mathcal{M}}(x)$ under the manifold hypothesis should not come as a surprise; indeed, as $p_\sigma \to p_{\text{data}}$ when $\sigma \to 0$, and since $p_{\text{data}}$ is supported entirely on $\mathcal{M}$, any point $x$ with $d_{\mathcal{M}}(x) > 0$ must be assigned zero probability, which explains the divergent scaling factor $\sigma^{-2}$ in the coefficient. What is more surprising is that *only* $d_{\mathcal{M}}(x)$ appears at leading order, with *no dependence on* $p_{\text{data}}$: Information about $p_{\text{data}}$ enters only at the higher-order terms of $\Theta(1)$.

This reveals a fundamental *rate separation*: for *any* distribution supported on $\mathcal{M}$, one must first recover $d_{\mathcal{M}}(x)$ *exactly* before learning anything about $p_{\text{data}}$, as any inaccuracy in $d_{\mathcal{M}}(x)$ gets blown

up by the diverging factor $\sigma^{-2}$. Moreover, coefficients encoding $p_{\text{data}}$ appear at order $O(\sigma^{-2})$ higher, meaning that extracting information about $p_{\text{data}}$ requires a level of accuracy orders of magnitude stricter than that needed to recover the manifold geometry, i.e., the distance function $d_{\mathcal{M}}$.

As demonstrated in Sections 4 to 5, this observation entails several significant consequences for machine learning. Each of these can be understood as a manifestation of the fundamental rate separation between geometric recovery *vs.* distributional learning established in Theorem 3.1.

## 4 SCALE SEPARATION IN EXISTING GENERATIVE LEARNING: GEOMETRY VERSUS DISTRIBUTION

In this section, we study the paradigm of existing generative learning where algorithms target to learn the Gaussian-smoothed measure $\mu_\sigma$, such as the diffusion models discussed in Section 2.1. We denote the corresponding perfect score function by $s^*(x, \sigma) := \nabla \log p_\sigma(x)$.

In practice, however, the generated samples may follow a different distribution due to imperfections such as errors in training or discretization of the reverse differential equation. We therefore let $\pi_\sigma(x) : \mathbb{R}^d \to \mathbb{R}$ denote the density of the distribution actually produced by an empirical algorithm, and define its associated score as $s_{\pi_\sigma}(x) := \nabla \log \pi_\sigma(x)$. Our analysis focuses on $\pi_\sigma$ in terms of discrepancies between $s_{\pi_\sigma}(x)$ and the ideal score $s^*(x, \sigma)$.

Before presenting our result, we impose the following assumption on the recovered distribution.

**Assumption 4.1.** *We denote the log-density of the recovered distribution as $-f_\sigma := \log \pi_\sigma(x)$, and assume that $f_\sigma$ is $C^1(K)$. Furthermore, we impose the following conditions:*

1. *There exists a compact set $K \subset \mathbb{R}^d$ with $T_{\mathcal{M}}(\epsilon) \subset K$ such that the density concentrates on $K$ as $\sigma \to 0$, i.e., $\lim_{\sigma \to 0} \int_K \pi_\sigma(x) \, dx = 1$.*

2. *$K$ is uniformly rectifiably path-connected, meaning that for any two points $x, y \in K$, there exists a path in $K$ connecting $x$ and $y$ whose length is uniformly bounded for all $x, y \in K$.*

*Remark* 4.1. We believe our assumptions are already reflected in practice: Since $\pi_\sigma$ represents the effective distribution of the generated samples, it can incorporate standard constraints such as data clipping (e.g., to $[-1, 1]$) used in many diffusion models (Ho et al., 2020; Saharia et al., 2022). This ensures the generated density concentrates on a compact set $K$ as required. Furthermore, such regular sets are naturally uniformly rectifiably path-connected.

We are ready to state our main result in this section; see Appendix B.3 for the proof.

**Theorem 4.1.** *Suppose Assumptions 2.1, 2.2 and 4.1 hold. Denote the score error as*

$$E_\sigma := \|s_{\pi_\sigma} - s^*(\cdot, \sigma)\|_{L^\infty(K)}.$$

1. ***Concentration on Manifold.*** *If we have that $\boldsymbol{E_\sigma = o(\sigma^{-2})}$, then $\pi_\sigma$ concentrates on $\mathcal{M}$, i.e.,*

$$\lim_{\sigma \to 0} \int_{\text{dist}(x, \mathcal{M}) > \delta} \pi_\sigma(x) dx = 0 \quad \text{for any} \quad \delta > 0.$$

2. ***Arbitrary Distribution Recovery.*** *For any distribution $\hat{\pi}$ supported on $\mathcal{M}$ with $C^1$ density, one can construct $f_\sigma$ such that $\boldsymbol{E_\sigma = \Omega(1)}$ as $\sigma \to 0$, and $\pi_\sigma$ converges weakly to $\hat{\pi}$.*

3. ***Recovering $p_{\text{data}}$.*** *If we have that $\boldsymbol{E_\sigma = o(1)}$ as $\sigma \to 0$, then $\pi_\sigma$ converges weakly to $p_{\text{data}}$.*

This result formalizes the intuitive fact that recovering $p_{\text{data}}$ requires $\nabla \log \pi_\sigma$ to match the true score to within $o(1)$ accuracy as $\sigma \to 0$. The reason is clear from the expansion (6): the distribution $p_{\text{data}}$ only appears in the $\Theta(1)$ term, and any larger error would overwhelm this information. In practice, however, achieving such accuracy is extremely challenging, particularly in the small-$\sigma$ regime. However, recovering the manifold is simple—only $o(1/\sigma^2)$ accuracy is required such that as $\sigma \to 0$, the density will concentrate on $\mathcal{M}$—a shape separation from recovering $p_{\text{data}}$.

**Implications to Diffusion Models.** As we mentioned before, the paradigmatic example to which our results can be applied is diffusion models. Our Theorem 4.1 then reveals a sharp scale separation

in terms of the score error: *well before the true distribution $p_{\text{data}}$ is fully recovered, one can already recover a distribution supported on the same data manifold*. In practice, this often suffices, as what truly matters is capturing the *structural features* of the manifold—realistic images, plausible protein conformations, or meaningful material geometries. This insight provides a potential new explanation for the remarkable success of diffusion models.

# 5 NEW PARADIGM OF GEOMETRIC LEARNING: RECOVER UNIFORM DISTRIBUTIONS WITH $o(\sigma^{-2})$ SCORE ERROR

As shown in Theorem 4.1, while concentration on the manifold is orders of magnitude simpler, the recovered distribution can still be **arbitrary** unless the score is learned with $o(1)$ accuracy. In contrast, we show in this section the striking fact that even with score errors as large as $o(\sigma^{-2})$, with a simple modification of the existing algorithm, one can recover the *uniform distribution on the manifold*—a fundamental distribution that plays a key role in scientific discovery and encodes rich geometric information about the manifold (De Santi et al., 2025; Belkin & Niyogi, 2008).

Unlike in Section 4, where we compared errors by evaluating a learned *distribution* $\pi_\sigma$ against the ideal $p_\sigma$ through their score functions, in this section we assume direct access to an estimated *score oracle* $s(\cdot, \sigma)$, such as those learned via score matching in diffusion models. Given access to such an oracle, our proposed algorithm consists of running the following SDE for some $\alpha > 0$:

$$dX_t = \sigma^\alpha s(X_t, \sigma)\, dt + \sqrt{2}\, dW_t, \tag{8}$$

which we refer to as the *Tempered Score* (TS) Langevin dynamics. We claim that, under mild error assumptions, the stationary distribution of this SDE, denoted $\tilde{\pi}_\sigma$, converges to the uniform distribution on the manifold as $\sigma \to 0$.

Our analysis proceeds in two steps. First, we establish the result in a simplified setting where the score oracle $s(\cdot, \sigma)$ is guaranteed to be a gradient field, with a proof analogous to Section 4. Second, we tackle the substantially more challenging case in which no *a priori* gradient structure is assumed. Full proofs are provided in Appendix B.5.

**Warm-up: Score Oracle is a Gradient Field.** We use the same notation as in Section 4, namely $s(x, \sigma) = -\nabla f_\sigma(x)$. In this case, the stationary distribution of Equation (8) admits the explicit form

$$\tilde{\pi}_\sigma(x) \propto \exp\left(-\sigma^\alpha f_\sigma(x)\right).$$

We then obtain the following result, using a proof technique similar to that of Theorem 4.1.

**Theorem 5.1.** *Assume Assumptions 2.1, 2.2 and 4.1 hold, with $\pi_\sigma$ replaced by $\tilde{\pi}_\sigma$. Suppose*

$$\|s(\cdot, \sigma) - s^*(\cdot, \sigma)\|_{L^\infty(K)} = o\left(\sigma^\beta\right) \quad \text{for some } \beta > -2. \tag{9}$$

*Then for any $\max\{-\beta, 0\} < \alpha < 2$, as $\sigma \to 0$, $\tilde{\pi}_\sigma$ converges weakly to the **uniform distribution** on the manifold $\mathcal{M}$ with respect to the intrinsic volume measure. More precisely, the limiting distribution $\tilde{\pi}$ with respect to the Lebesgue measure on $U$ satisfies*

$$\tilde{\pi}(u) \propto \frac{d\mathcal{M}}{du}(u),$$

*where $(d\mathcal{M}/du)(u) = \sqrt{\det(g(u))}$ is the Riemannian volume element on $\mathcal{M}$.*

**General Non-Gradient Score Oracle.** While theorem 5.1 already illustrates the rate separation phenomenon we wish to emphasize, it relies on the highly impractical assumption that the estimated scores $s(\cdot, \sigma)$ are exact gradient fields. To enhance the applicability of our framework, it is crucial to relax this stringent assumption.

As discussed in Section 2.3, existing approaches to non-gradient scores (and hence non-reversible dynamics) typically assume the existence of a unique point $x^*$ such that $\lim_{\sigma \to 0} \sigma^\alpha s(x^*, \sigma) = 0$, with the key consequence of collapsing the prefactor $c_0$ in (5) to a normalization constant $c_0(x^*)$. Our framework, however, explicitly violates this assumption: we require that $\lim_{\sigma \to 0} \sigma^\alpha s(\cdot, \sigma)$ stabilizes to a *manifold* rather than a singleton. Under this setting, the limiting behavior of $c_0$ is far from obvious, and the resolution of this issue turns out to be highly nontrivial.

To this end, a central part of our proof is devoted to showing that $c_0$ nevertheless remains constant, albeit for an entirely different reason: we prove that the higher-order terms in the Fokker–Planck expansion enforce $c_0$ to satisfy a *parabolic PDE* on the manifold, and by the strong maximum principle (Gilbarg et al., 1977), the only solutions on a compact manifold are constants.

With these techniques, we obtain the same conclusion as Theorem 5.1:

**Theorem 5.2.** *Assume Assumptions 2.1 and 2.2 and eq. (9) hold, and further suppose $p_{\text{data}} \in C^2(U)$. For any $\max\{-\beta, 0\} < \alpha < 2$, assume that the SDE admits a unique stationary distribution, denoted $\tilde{\pi}_\sigma$, which locally admits a WKB form (Assumption B.2 with $\theta = \sigma^{2-\alpha}$). Then the conclusion of Theorem 5.1 holds.*

Setting $\alpha = 0$ in eq. (8) recovers the standard Langevin sampler or the "Corrector" step commonly used in diffusion-based sampling (Song et al., 2021). Our results in Theorems 5.1 and 5.2 therefore imply that a simple, one-line modification of these standard schemes is enough to recover the uniform distribution on the data manifold *from samples of* $p_{\text{data}}$, even when the score error is as large as $o(\sigma^{-2})$—a substantially weaker requirement than the $o(1)$ accuracy needed to recover $p_{\text{data}}$ itself. *Remark* 5.1. In Appendix D, we provide further discussion on the convergence (mixing time) of TS Langevin compared to standard Langevin dynamics. While characterizing the general convergence rate is a non-trivial problem left for future work, our analysis indicates that TS Langevin maintains comparable algorithmic efficiency. In fact, by analyzing the Poincaré constant, we identify concrete examples where TS Langevin converges provably exponentially faster than standard, untempered Langevin dynamics.

## 6 UNIFORM PRIOR IS MORE ROBUST BAYESIAN INVERSE PROBLEMS

In Bayesian learning, one often sets the prior $p_{\text{prior}}$ to the Gaussian-smoothed data distribution $p_\sigma$ defined in Equation (3) with some small smoothing parameter $\sigma$. To ensure asymptotically correct posterior samples under this choice, the learned score typically must be exact (Pesme et al., 2025), $\hat{s} = \nabla \log p_\sigma$, or achieve vanishing error, $\|\hat{s} - \nabla \log p_\sigma\|_{\mathcal{L}^\infty} = o(1)$ (Laumont et al., 2022, Proposition 3.3 and H2). In contrast, under our framework, if one adopts the manifold volume measure (i.e., the uniform distribution on $\mathcal{M}$) as the prior, then correct posterior sampling can be attained under a substantially weaker requirement: it suffices that the score error scales as $o(\sigma^{-2})$. The precise statement is given in the theorem below.

**Theorem 6.1.** *Under the same assumptions as in Theorem 5.2, and suppose $v : \mathbb{R}^d \to \mathbb{R}$ is bounded on $\mathbb{R}^d$, and $C^1$ on $T_{\mathcal{M}}(\epsilon)$. Then, as $\sigma \to 0$, the stationary distribution of the SDE*

$$dx_t = -\nabla v(x_t)\, dt - \sigma^\alpha \nabla f_\sigma(x_t)\, dt + \sqrt{2}\, dW_t, \tag{10}$$

*converges weakly to a distribution supported on $\mathcal{M}$ with density $\propto \exp\big(-v(\Phi(u))\big) \frac{d\mathcal{M}}{du}(u)$.*

**Diffusion Models with Classifier-Free Guidance.** The above result can also be applied to diffusion models. The drift term in Equation (10) represents the effective score of a diffusion model with classifier-free guidance (Ho & Salimans, 2022). In this formulation, $-\nabla f_\sigma$ denotes the unconditional score estimate, while the guidance term $-\nabla v$ equals the guidance scale $w$ times the difference between the conditional and unconditional score estimates. Our tempered score can be applied directly to CFG diffusion models with a Predictor–Corrector sampler: in the corrector (Langevin) step, replace the score by its tempered version according to Equation (10) (i.e., scale the unconditional score by $\sigma^\alpha$). We will demonstrate the effectiveness of this modification empirically in Section 7.2.

## 7 EXPERIMENTS

To empirically validate our theory, we present preliminary experiments on both simple synthetic manifolds and a real-world image–generation setting with diffusion models. On synthetic manifolds, we directly verify the claims of Section 5, demonstrating recovery of the uniform distribution on the manifold. In the image domain, we show that our proposed algorithm yields samples that are both more diverse and high-quality. Further experimental details are provided in Appendix C.

### 7.1 NUMERICAL SIMULATIONS ON ELLIPSE

In this subsection, we illustrate our theoretical results with numerical simulations. We consider a simple manifold given by an ellipse embedded in the two-dimensional Euclidean space, $\mathcal{M} =$

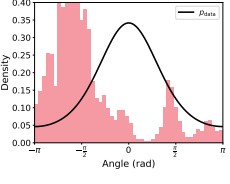 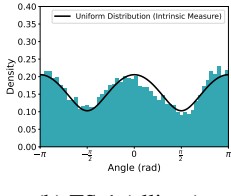 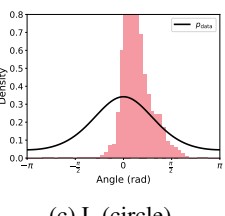 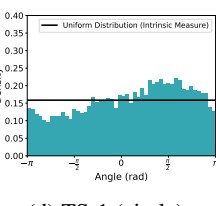

(a) L (ellipse)    (b) TS-1 (ellipse)    (c) L (circle)    (d) TS-1 (circle)

Figure 2: Comparison of stationary sample distributions generated with standard Langevin dynamics (L) versus our Tempered Score Langevin dynamics Equation (8) with $\alpha = 1$ (TS-1). The circle and ellipse correspond to manifolds with $(a, b) = (1, 1)$ and $(a, b) = (1, 2)$, respectively.

| Prompt | Furniture | | Car | | Architecture | |
|--------|-----------|---------|---------|-------|--------------|-------|
| **Method** | P-sim↑ | I-sim↓ | P-sim | I-sim | P-sim | I-sim |
| DDPM | 29.56 | 80.78 | 26.23 | 87.30 | **27.36** | 81.53 |
| PC | 29.40 | 81.24 | 26.30 | 87.20 | 27.13 | 81.03 |
| *TS (ours)* | **30.20** | **80.76** | **26.62** | **87.14** | 27.32 | **80.76** |

Table 1: Comparison of images generated by DDPM, PC, and TS. The prompts used are "Creative furniture," "An innovative car design," and "A creative architecture." For PC and TS, the number of corrector steps and $\alpha$ (for TS) are tuned.

$\{(x, y) \in \mathbb{R}^2 \mid (x/a)^2 + (y/b)^2 = 1\}, \quad a, b > 0$, and $p_{\text{data}}$ is chosen to be a von Mises distribution supported on the angular parameterization of the ellipse. The score function is parameterized using a transformer-based neural network, trained with the loss function introduced in (Song & Ermon, 2019). After training, we evaluate the learned score function with $\sigma = 10^{-2}$ and perform Langevin dynamics until convergence. Training hyperparameters are tuned to minimize the test loss.

As shown in Figure 2, the stationary distribution produced by standard Langevin dynamics deviates substantially from $p_{\text{data}}$, even in this simple elliptical setting, highlighting the difficulty of accurately learning the score function at small $\sigma$. In contrast, our TS Langevin dynamics reliably recovers the uniform distribution on the manifold, in agreement with Theorem 5.2.

## 7.2 IMAGE GENERATION WITH DIFFUSION MODELS

To validate our theoretical findings in a practical, large-scale setting, we conducted experiments on image generation. We demonstrate that a one-line modification to the widely-used Predictor-Corrector (PC) sampling algorithm (Song et al., 2021) can enhance both the quality and diversity of images generated by a pre-trained diffusion model. These experiments serve as a proof of concept, applying our proposed Tempered Score (TS) method to off-the-shelf diffusion models. Our modification targets the corrector step of the PC algorithm, which uses Langevin dynamics to refine the sample at each stage of the reverse process. In our TS method, we scale the unconditioned score prediction by a factor of $\sigma^{\alpha}$, as motivated by our analysis and discussion in Section 6. The standard classifier-free guidance term, i.e., $\nabla v$ in Equation (10), remains unchanged. Specifically, we compare Stable Diffusion 1.5 (Rombach et al., 2022) with a DDPM sampler (Ho et al., 2020), DDPM with PC sampler, and DDPM with our TS sampler.

We evaluate the performance using two metrics derived from CLIP scores (Hessel et al., 2021), which measure the cosine similarity between feature embeddings. **Quality**: We use the CLIP Prompt Similarity (P-sim), defined as the average CLIP score between the generated images and their corresponding text prompt. A higher P-sim value indicates better alignment with the prompt and thus higher image quality. **Diversity**: We use the CLIP Inter-Image Similarity (I-sim), which is the average pairwise CLIP score between all images generated with the same prompt. A lower I-sim value means greater diversity among the samples.

The experimental results in Table 1 and Table 2 provide empirical validation of our theoretical framework. Our proposed TS method consistently generates more diverse images than the DDPM and standard PC baselines across three distinct prompts, while maintaining very high image quality. In particular, Table 2 shows that, for all numbers of corrector steps considered, TS outperforms

| Num. Corrector Steps | | 5 | | 10 | | 15 | | 20 | | 30 | |
|---|---|---|---|---|---|---|---|---|---|---|---|
| **Prompt** | **Method** | P-sim↑ | I-sim↓ | P-sim | I-sim | P-sim | I-sim | P-sim | I-sim | P-sim | I-sim |
| **Furniture** | PC | 29.40 | 81.34 | 29.30 | 81.24 | 29.32 | 81.64 | 28.98 | 81.72 | 28.67 | 82.33 |
| | *TS (ours)* | **29.54** | **81.11** | **29.58** | **80.95** | **29.68** | **81.34** | **29.52** | **81.15** | **29.43** | **81.87** |
| **Car** | PC | 26.20 | 87.20 | 26.30 | 87.57 | 26.24 | 87.98 | 26.26 | **88.06** | 26.17 | 87.94 |
| | *TS (ours)* | **26.23** | **87.14** | **26.37** | **87.42** | **26.32** | **87.88** | **26.28** | 88.07 | **26.20** | **87.87** |
| **Architect.** | PC | 27.13 | 81.83 | 27.13 | 81.81 | 26.92 | 81.64 | 26.87 | 81.60 | 26.60 | 81.03 |
| | *TS (ours)* | **27.23** | **81.58** | **27.27** | **81.57** | **27.14** | **81.54** | **27.06** | **80.97** | **26.84** | **80.76** |

Table 2: Comparison of images generated by PC and TS across different numbers of corrector steps. For TS, $\alpha = 1$ is used without further tuning. The prompts are the same as in Table 1.

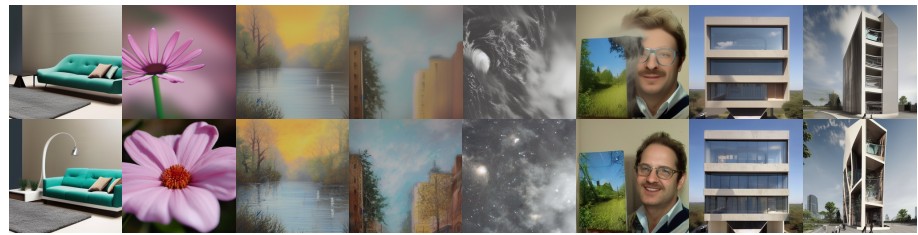

Figure 3: Top row: PC. Bottom row: *TS (ours)*. Samples in the same column are generated using the same prompt, the same number of corrector steps, and the same random seed. As shown, TS produces samples that appear more authentic and contain richer details.

standard PC in nearly every case. Crucially, these improvements are robust to the choice of $\alpha$ and are not merely the result of a larger tuning budget; as demonstrated in Table 2, simply setting $\alpha = 1$ without further tuning is sufficient to consistently enhance both quality and diversity compared to the baseline. Examples of the generated images by PC and TS are shown in Figure 3.

## 8 CONCLUSION

This paper advocates for a paradigm shift in score-based learning, moving from the difficult goal of full distributional recovery to a more robust, geometry-first approach. We demonstrate a fundamental rate separation in the low-noise limit, where information about the data manifold is encoded at a significantly stronger scale ($\Theta(\sigma^{-2})$) than details about the on-manifold distribution ($\Theta(1)$). This finding explains why models often succeed at capturing the data support even with imperfect score estimates. Building on this insight, we introduce Tempered Score (TS) Langevin dynamics, a simple one-line modification that robustly targets the uniform volume measure on the manifold, tolerating score errors up to $o(\sigma^{-2})$. This geometric approach not only provides a more stable foundation for Bayesian inverse problems but also, as shown in our experiments with models like Stable Diffusion, empirically improves the diversity and fidelity of generated samples.

**Limitations and future work.** Key limitations and future directions include: a) The implications for diffusion models are presently limited: we do not track cumulative error along the sampling trajectory; instead, we analyze a simplified setting that assumes access to the error of the final generated distribution. b) Our $L^{\infty}$ score–error assumption could potentially be relaxed to an $L^2$ bound, thereby aligning our theoretical framework with practical training objectives like denoising score matching (Fisher divergence) that minimize $L^2$ error. c) It remains to generalize the rate separation in score estimation into corresponding results on statistical sample complexity. d) Our analyses on the uniform sampling are in continuous time; we do not quantify discretization error arising in practical implementations. e) Our experiments are preliminary; we have not conducted a large-scale study with state-of-the-art diffusion models.

## ACKNOWLEDGMENT

The work is supported by ETH research grant, Swiss National Science Foundation (SNSF) Project Funding No. 200021-207343, and SNSF Starting Grant.

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

## A    ADDITIONAL NOTATION AND PRELIMINARIES

In this section, we provide some notation and preliminaries complementary to Section 2.

We denote by $W_t$ a standard Brownian motion, with its dimension clear from context. The Gaussian density with mean $\mu$ and covariance $\Sigma$, evaluated at $x$, is written as $\mathcal{N}(x \mid \mu, \Sigma)$. The symbol $*$ denotes the convolution operator. We use $\propto$ to indicate proportionality, i.e., that the left-hand side and right-hand side are equal up to a constant factor. For a set $S$, we write $\overline{S}$ for its closure, $\partial S$ for its boundary, and $S^c$ for its complement. Throughout the paper, by the term *limiting distribution* or by convergence of a distribution/density function, we mean convergence of the corresponding measures in the weak sense.

### A.1    THE MANIFOLD HYPOTHESIS

We outline few notations and standard results from differential geometry. By the tubular neighborhood theorem (Milnor & Stasheff, 1974; Weyl, 1939), there exists $\epsilon > 0$ such that the normal tube
$$T_{\mathcal{M}}(\epsilon) := \{x \in \mathbb{R}^d : \text{dist}(x, \mathcal{M}) < \epsilon\}.$$
admits local $C^4$ coordinate
$$\Phi : U \times R \to T_{\mathcal{M}}(\epsilon), \quad \text{where} \quad U \subset \mathbb{R}^n, R := \{r \in \mathbb{R}^{d-n} : \|r\| < \epsilon\},$$
such that $\Phi$ is a diffeomorphism mapping from local coordinates to ambient Euclidean space. With this result, we can then work with local coordinates to describe the manifold. For notational simplicity, we work with a single chart and suppress indices: $u \in U$ denote tangential coordinates and $r \in R$ denote normal coordinates. The slice $r = 0$ corresponds to points on $\mathcal{M}$, and we write $\Phi(u) := \Phi(u, 0)$. Let $J(u, r)$ denote the Jacobian of $\Phi(u, r)$ with respect to $(u, r)$, i.e., $J(u, r) = \partial \Phi(u, r)/\partial(u, r)$. Furthermore, let $g(u)$ denote the Riemannian metric tensor of the manifold $\mathcal{M}$, defined as $g(u) := J(u, 0)^\top J(u, 0)$. Intuitively, the Riemannian metric tensor gives a way to measure lengths and angles of the manifold geometry.

## B    PROOFS OF MAIN THEOREMS

In this section, we prove the main theorems of the paper. We begin by developing a general framework for characterizing the limiting distribution when the density admits a specific form. This framework will then be applied to establish the results in Section 4, where such a density form was assumed.

The results in Section 5 require a different approach, since no explicit form of the density is available. In this case, we employ the WKB approximation to obtain an approximate stationary distribution, which we then substitute into the general framework to derive the limiting distribution.

### B.1    A GENERAL FRAMEWORK FOR THE CONVERGENCE OF THE LIMITING DISTRIBUTION

In this subsection, we will establish a general framework for the limiting distribution of density proportional to
$$\exp\left(-\left(f_\theta(x)\right)/\theta\right), \quad \text{with} \quad f_\theta(x) = f_0(x) + \theta f_1(x) + \hat{f}(x, \theta), \tag{11}$$
where $f_0$'s minimizer is on the manifold $\mathcal{M}$ and $\hat{f}(x, \theta)$ is a perturbation that is uniformly $o(\theta)$ so that it does not affect the limiting distribution. This general result is stated in Theorem B.1. Our main results fall into this framework by letting $\theta = \sigma^2$ for Theorem 4.1 and $\theta = \sigma^{2-\alpha}$ for Theorem 5.2.

In all cases the theorems we will prove later, the density will concentrate on the tubular neighborhood of $M$, i.e., $T_{\mathcal{M}}(\epsilon)$. Therefore, we will discuss the lemmas and intermediate results in such a neighborhood and use local coordinates $(u, r)$. The notations used can be found in Section 2. When we use local coordinates, we assume the discussion is in the closure of $T_{\mathcal{M}}(\epsilon)$. We define the local coordinate versions of the functions: $f_\theta(u, r) := f_\theta(\Phi(u, r))$, $f_0(u, r) := f_0(\Phi(u, r))$, $f_1(u, r) := f_1(\Phi(u, r))$, and $\hat{f}(u, r, \theta) := \hat{f}(\Phi(u, r), \theta)$.

Our assumptions are stated as follows.

**Assumption B.1.** *We assume that*

1. $\mathcal{M} \subset \mathbb{R}^d$ *is a compact* $C^4$ *manifold without boundary with dimension* $n < d$.

2. $\mathcal{M} = \arg\min_{x \in T_{\mathcal{M}}(\epsilon)} f_0(x)$. *In addition, we assume that there exists* $0 < \hat{\epsilon} < \epsilon$ *such that* $\inf_{x \in T_{\mathcal{M}}(\epsilon) \setminus \overline{T_{\mathcal{M}}(\hat{\epsilon})}} f_0(x) - \min_{x \in T_{\mathcal{M}}(\epsilon)} f_0(x)$ *is bounded away from zero.*

3. *The absolution value of* $\hat{f}(u, r, \theta)$ *is* $o(\theta)$ *as* $\theta \to 0$ *uniformly for all* $u \in U$ *and* $\|r\| < \epsilon$.

4. $f_0 \geq 0$ *is* $C^3$, $f_1$ *is* $C^1$, *and* $f_\theta$ *is continuous on coordinates* $(u, r)$ *for all* $u \in U$ *and* $\|r\| \leq \epsilon$, *i.e., in the closure of* $T_{\mathcal{M}}(\epsilon)$.

5. *Further, we assume that the smallest eigenvalue of* $\frac{\partial^2 f_0}{\partial r^2}(u, r)$ *is uniformly bounded away from zero for all* $u \in U$ *and* $\|r\| < \epsilon$.

*Remark* B.1 (Compactness of the manifold implies boundedness of gradients.). Consider $f \in C^k(\overline{T_{\mathcal{M}}(\epsilon)})$. In local coordinates $(u, r)$ induced by a tubular atlas, we write $f(u, r) := f(\Phi(u, r))$. Since $\mathcal{M}$ is compact, one can choose a finite atlas with precompact coordinate domains. Let the cover be $\{U_i\}$. By the Shrinking Lemma (Munkres (2000, Theorem 32.3) combined with Willard (2012, Theorem 15.10)), there exist open subsets $\{V_i\}$ with $\overline{V_i} \subset U_i$ such that $\{V_i\}$ still forms a cover. We use these $\{V_i\}$ as the new atlas. The transition maps $\Phi$ and their derivatives are then bounded on these sets (since $\overline{V_i}$ is compact), and by the chain rule the same holds for $f(u, r)$ and its derivatives up to order $k$. Thus, throughout our arguments we may freely assume uniform boundedness of such derivatives without loss of generality. The same reasoning applies to $p_{\text{data}}$, we can use the same constructed atlas such that $p_{\text{data}}$ is uniformly lower and upper bounded, and gradients of $p_{\text{data}}$ are uniformly upper bounded.

During our proofs, we will frequently use Laplace's method for integrals. We adapt the error estimate from Łapiński (2019) as follows.

**Corollary B.1** (Theorem 2 of Łapiński (2019)). *Let* $\Omega \subset \mathbb{R}^m$ *be an open set and let* $\Omega' \subset \Omega$ *be a closed ball. Let* $c_1 := \text{Vol}(\Omega')$. *Let* $F, g : \Omega \to \mathbb{R}$ *with the following assumptions:*

1. $F|_{\Omega'} \in C^3(\Omega')$ *and* $F \geq 0$ *on* $\Omega$. *There is a unique minimizer* $x^* \in \text{int}(\Omega')$ *of* $F$ *on* $\Omega$. *Define*

$$m_1 := \inf_{x \in \Omega \setminus \Omega'} \{F(x) - F(x^*)\} > 0, \qquad m_2 := \inf_{x \in \Omega'} \lambda_{\min}(\nabla^2 F(x)) > 0.$$

   *Let*

$$c_2 := \sup_{x \in \Omega'} \|\nabla^2 F(x)\|, \qquad c_3 := \sup_{x \in \Omega'} \|\nabla^3 F(x)\|.$$

2. $g|_{\Omega'} \in C^1(\Omega')$ *and* $\int_\Omega |g(x)|\, dx < \infty$. *Let*

$$c_4 := \sup_{x \in \Omega'} |g(x)|, \qquad c_5 := \sup_{x \in \Omega'} \|\nabla g(x)\|, \qquad c_6 := \int_\Omega |g(x)|\, dx.$$

*Then, for every* $\theta > 0$,

$$\int_\Omega g(x)\, e^{-F(x)/\theta}\, dx = \exp(-F(x^*)/\theta)\frac{(2\pi\theta)^{m/2}}{\sqrt{|\nabla^2 F(x^*)|}}\left(g(x^*) + h(\theta)\right),$$

*where* $|h(\theta)|$ *can be upper bounded by a function of* $(c_1, \ldots, c_6, m_1, m_2)$. *Moreover,* $h(\theta) = O(\sqrt{\theta})$ *as* $\theta \to 0$. *The* $O(\sqrt{\theta})$ *is* uniform *over any class of pairs* $(F, g)$ *for which* $c_1, \ldots, c_6$ *are bounded above and* $m_1, m_2$ *are bounded below by strictly positive constants uniformly over the class.*

*Proof.* The result follows directly from Łapiński (2019, Theorem 2). $\qquad\square$

To show the convergence of the distribution to a distribution on the manifold, a key step is to integrate out the normal direction so as to obtain a distribution on $u$, such as what Hwang (1980) did. The following lemma proves Laplace's type of result for integrating out $r$.

**Lemma B.1.** *Assume Assumption B.1, and let $h(x) : \mathbb{R}^d \to \mathbb{R}$ be $C^1$ and uniformly bounded in $T_{\mathcal{M}}(\epsilon)$. Define $h(u, r) := h(\Phi(u, r))$. Then we have*

$$\int_{\|r\|<\epsilon} \exp\left(-\frac{f_\theta(u, r)}{\theta}\right) h(u, r) dr$$

$$= \exp\left(-\frac{f_0(u, 0)}{\theta}\right) \exp(-f_1(u, 0)) \frac{(2\pi\theta)^{(d-n)/2}}{\sqrt{\left|\frac{\partial^2 f_0}{\partial r^2}(u, 0)\right|}} (h(u, 0) + o(1)),$$

*where the $o(1)$ term is uniform for $u$.*

*Proof.* We have that

$$\int_{\|r\|<\epsilon} \exp\left(-\frac{f_\theta(u, r)}{\theta}\right) h(u, r) dr$$

$$= \int_{\|r\|<\epsilon} \exp\left(-\frac{f_0(u, r)}{\theta}\right) \exp(-f_1(u, r)) h(u, r) \left(\exp\left(-\frac{\hat{f}(u, r, \theta)}{\theta}\right)\right) dr$$

$$= \int_{\|r\|<\epsilon} \exp\left(-\frac{f_0(u, r)}{\theta}\right) \exp(-f_1(u, r)) h(u, r) dr +$$

$$\int_{\|r\|<\epsilon} \exp\left(-\frac{f_0(u, r)}{\theta}\right) \exp(-f_1(u, r)) h(u, r) \left(\exp\left(-\frac{\hat{f}(u, r, \theta)}{\theta}\right) - 1\right) dr.$$

For the first term, we can directly apply Corollary B.1 with $F(r) = f_0(u, r)$, $g(r) = \exp(-f_1(u, r)) h(u, r)$, and $\Omega'$ being the ball $\{r \mid \|r\| \le \hat{\epsilon}\}$. Define

$$J = \exp\left(-\frac{f_0(u, 0)}{\theta}\right) \exp(-f_1(u, 0)) \frac{(2\pi\theta)^{(d-n)/2}}{\sqrt{\left|\frac{\partial^2 f_0}{\partial r^2}(u, 0)\right|}}.$$

The first term can be approximated as $J(h(u, 0) + o(1))$. The boundedness of the quantities in Corollary B.1 will be discussed later. The second term can be upper bounded by

$$\sup_r |h(u, r)| \cdot \sup_r \left|\exp\left(-\frac{\hat{f}(u, r, \theta)}{\theta}\right) - 1\right| \int_{\|r\|<\epsilon} \exp\left(-\frac{f_0(u, r)}{\theta}\right) \exp(-f_1(u, r)) dr$$

$$= o(1) J(1 + o(1)) = o(1) J,$$

where we used Corollary B.1 for the integral. The lower bound can be obtained similarly. The result follows.

Regarding the uniform boundedness of the quantities in Corollary B.1, $\{c\}_1^5$ is uniformly bounded by the compactness of the manifold. The constant $c_6$ is uniformly bounded by our assumption on $h$. The uniform lower bounds of $m_1$ and $m_2$ is guaranteed by Assumption B.1. $\square$

Next, we will prove that the support of the limiting distribution will concentrate on the minimizers of the leading term. Previously, we considered $f_\theta$ consisting of $f_0 + \Theta(\theta) + o(\theta)$. Next, we will show that as long as $f_\theta$ is $f_0 + o(1)$, the concentration on $f_0$'s minimizers will happen.

**Lemma B.2.** *Let $f_\theta(x) = f_0(x) + \tilde{f}(x, \theta)$, such that $\exp(-f_\theta(x)/\theta)$ is a normalized density function on $\mathbb{R}^d$. Suppose $\mathcal{M}$ is a connected and compact $C^4$ manifold without boundary. Assume that:*

*1. $f_0(x)$ is continuous with $\arg\min_{x \in \overline{T_{\mathcal{M}}(\epsilon)}} f_0(x) = \mathcal{M}$ and $\min_{x \in x \in \overline{T_{\mathcal{M}}(\epsilon)}} f_0(x) = 0$.*

*2. $\tilde{f}(x, \theta)$ is continuous and uniformly $o(1)$ as $\theta \to 0$ for all $x \in \overline{T_{\mathcal{M}}(\epsilon)}$.*

*3. The density concentrates in $T_{\mathcal{M}}(\epsilon)$, i.e.,*

$$\lim_{\theta \to 0} \int_{T_{\mathcal{M}}(\epsilon)} \exp\left(-\frac{f_\theta(x)}{\theta}\right) dx = 1.$$

*For any $\eta > 0$, define the set $C_\eta = \{x \mid f_0(x) > \eta\}$. Then,*

$$\int_{C_\eta \cup T_\mathcal{M}(\epsilon)^c} \exp(-f_\theta(x)/\theta)\, dx \to 0 \quad as \quad \theta \to 0.$$

*If in addition, $\exp(-f_\theta(x)/\theta)$ converges weakly to a distribution as $\theta \to 0$, the support of the limiting distribution is contained in $\mathcal{M}$.*

*Proof.* Since we have that $\int_{T_\mathcal{M}(\epsilon)} \exp(-f_\theta(x)/\theta)dx \to 1$, for the first result, it suffices to show that $\int_{T_\mathcal{M}(\epsilon) \cap C_\eta} \exp(-f_\theta(x)/\theta)dx \to 0$. According to the assumptions, we have that for any $\delta$, $\exists \theta_0$, such that $\forall \theta < \theta_0$, $|\tilde{f}(x,\theta)| < \delta$. Therefore, we have

$$\int_{T_\mathcal{M}(\epsilon) \cap C_\eta} \exp(-f_\theta(x)/\theta)dx \leq \int_{T_\mathcal{M}(\epsilon) \cap C_\eta} \exp((-\eta + \delta)/\theta)dx \leq \mathrm{Vol}(T_\mathcal{M}(\epsilon)) \exp((-\eta + \delta)/\theta).$$

We choose $\delta = \eta/2$, then the right-hand side goes to zero as $\theta \to 0$.

Let the limiting measure be $P$, and $P_\theta$ be the probability measure corresponding to the density $\exp(-f_\theta(x)/\theta)$. Since $C_\eta$ is an open set, we have that

$$P(C_\eta) \leq \liminf_{\theta \to 0} P_\theta(C_\eta) = 0.$$

We also have that

$$P\left(\overline{T_\mathcal{M}(\epsilon)}^c\right) \leq \liminf_{\theta \to 0} P_\theta\left(\overline{T_\mathcal{M}(\epsilon)}^c\right) \leq \liminf_{\theta \to 0} P_\theta\left(T_\mathcal{M}(\epsilon)^c\right) = 0.$$

Denote $C := \mathcal{M}^c$. We have that $C = \cup_{m=1}^\infty C_{1/m} \cup \overline{T_\mathcal{M}(\epsilon)}^c$. Then we have

$$P(C) \leq \sum_{m=1}^\infty P(C_{1/m}) + P\left(\overline{T_\mathcal{M}(\epsilon)}^c\right) = 0.$$

which concludes the proof. $\qquad \square$

**Theorem B.1.** *Assume Assumption B.1. Define*

$$\pi_\theta(x) \propto \exp\left(-\frac{f_\theta(x)}{\theta}\right),$$

*Assume that $1 - \int_{x \in T_\mathcal{M}(\epsilon)} \pi_\theta(x)dx \to 0$ as $\theta \to 0$. Then we have that as $\theta \to 0$, $\pi_\theta$ converges weakly to the following distribution:*

$$\pi(u) = \frac{\exp(-f_1(u,0))\left|\frac{\partial^2 f_0(u,0)}{\partial r^2}\right|^{-1/2} d\mathcal{M}(u)/du}{\int_\mathcal{M} \exp(-f_1(u,0))\left|\frac{\partial^2 f_0(u,0)}{\partial r^2}\right|^{-1/2} d\mathcal{M}(u)/du},$$

*where $d\mathcal{M}$ is the intrinsic measure on the manifold $\mathcal{M}$, i.e., $d\mathcal{M}(u) = |g(u)|^{1/2}du$, and $du$ is the Lebesgue measure on the local parameterization domain $U$.*

*Proof.* The proof follows the same as the proof in Hwang (1980, Theorem 3.1). The only difference is that we replace the estimate of Hwang (1980, Equation (3.2)) with our Lemma B.1. Note that the Q in Hwang (1980, Theorem 3.1) is assumed as a probability measure, thus $f$ (in his notation) integrates to one. However, the proof technique of Hwang (1980, Theorem 3.1) remains valid even if $f$ is not a probability density, so applying to our case. $\qquad \square$

## B.2 Proof for Theorem 3.1

The remaining of the proof is to expand the true log-density w.r.t. $\sigma$, analyze the error of the learned log-density, and then to plug in the result obtained from Appendix B.1.

**Theorem B.2.** *Assume Assumptions 2.1 and 2.2 holds. Suppose $x \in T_{\mathcal{M}}(\epsilon)$. Then we have that*

$$\log p_{\sigma}^{\text{VE}}(x) = -\frac{1}{2\sigma^2}\|x - P_{\mathcal{M}}(x)\|^2 + \log p_{\text{data}}(\Phi^{-1}(P_{\mathcal{M}}(x))) - \frac{d-n}{2}\log(2\pi\sigma^2) -$$

$$\log\sqrt{\left|\hat{H}(\Phi^{-1}(P_{\mathcal{M}}(x)), x)\right|} + \hat{p}^{\text{VE}}(x, \sigma),$$

$$\log p_{\sigma}^{\text{VP}}(x) = -\frac{1}{2\sigma^2}\|x - P_{\mathcal{M}}(x)\|^2 + \log p_{\text{data}}(\Phi^{-1}(P_{\mathcal{M}}(x))) - \frac{d-n}{2}\log(2\pi\sigma^2) -$$

$$\log\sqrt{\left|\hat{H}(\Phi^{-1}(P_{\mathcal{M}}(x)), x)\right|} - \frac{1}{2}\langle P_{\mathcal{M}}(x), x - P_{\mathcal{M}}(x)\rangle + \hat{p}^{\text{VP}}(x, \sigma),$$

*where $\hat{p}^{\text{VE}}(x, \sigma)$ and $\hat{p}^{\text{VP}}(x, \sigma)$ are functions that are $o(1)$ uniformly for $x \in T_{\mathcal{M}}(\epsilon)$. The matrix $\hat{H}(u, x)$ is such that*

$$\hat{H}(u, x)_{i,j} = \left\langle \frac{\partial^2 \Phi(u)}{\partial u_i \partial u_j}, \Phi(u) - x \right\rangle + \left\langle \frac{\partial \Phi(u)}{\partial u_i}, \frac{\partial \Phi(u)}{\partial u_j} \right\rangle.$$

*Proof.* We can apply Corollary B.1 as an error estimate for Laplace's method, to the integral in $p_{\sigma}$. The minimizer of $F(u)$ is $\Phi^{-1}(P_{\mathcal{M}}(x))$ for both VE and VP.

We first consider the case of VE. By letting $F(u) = \|x - \Phi(u)\|^2/2$, $g(u) = p_{\text{data}}(u)$ and $\theta = \sigma^2$ we can obtain that

$$p_{\sigma}(x) = \exp\left(-\frac{\|x - P_{\mathcal{M}}(x)\|^2}{2\sigma^2}\right) \frac{(2\pi\sigma^2)^{(n-d)/2}}{\sqrt{\left|\hat{H}(\Phi^{-1}(P_{\mathcal{M}}(x)), x)\right|}} \left(p_{\text{data}}(\Phi^{-1}(P_{\mathcal{M}}(x))) + h(\sigma^2)\right) \tag{12}$$

where $|h(\sigma^2)|$ is $O(\sigma)$. Now we take logarithmic and use the fact that $\log(A + B) = \log(A) + \log(1 + B/A)$, we obtain

$$\log p_{\sigma}(x)$$
$$= -\frac{\|x - P_{\mathcal{M}}(x)\|^2}{2\sigma^2} + \frac{n-d}{2}\log(2\pi\sigma^2) + \log\left|\hat{H}(\Phi^{-1}(P_{\mathcal{M}}(x)), x)\right|^{-1/2} +$$
$$\log\left(p_{\text{data}}(\Phi^{-1}(P_{\mathcal{M}}(x))) + h(\sigma^2)\right)$$
$$= -\frac{\|x - P_{\mathcal{M}}(x)\|^2}{2\sigma^2} + \frac{n-d}{2}\log(2\pi\sigma^2) + \log p_{\text{data}}(\Phi^{-1}(P_{\mathcal{M}}(x))) +$$
$$\log\left|\hat{H}(\Phi^{-1}(P_{\mathcal{M}}(x)), x)\right|^{-1/2} + \log\left(1 + \frac{h(\sigma^2)}{p_{\text{data}}(\Phi^{-1}(P_{\mathcal{M}}(x)))}\right).$$

Therefore, we have

$$\hat{p}(x, \sigma) = \log\left(1 + \frac{h(\sigma^2)}{p_{\text{data}}(\Phi^{-1}(P_{\mathcal{M}}(x)))}\right)$$

The remaining is to show that $h(\sigma^2)/p_{\text{data}}(\Phi^{-1}(P_{\mathcal{M}}(x)))$ is uniformly $o(1)$ for all $x \in T_{\mathcal{M}}(\epsilon)$. Since the manifold is compact, $p_{\text{data}}(u)$ is uniformly bounded away from zero (see Remark B.1). The remaining is to find a suitable $\Omega'$ and upper and lower bound the constants in Corollary B.1. We will discuss this later.

Now let us look at the case of VP. The only difference is in the exponential, we changed from $\|x - \Phi(u)\|^2$ to

$$\left\|x - \sqrt{1 - \sigma^2}\Phi(u)\right\|^2 = \left\|x - \Phi(u) + \left(1 - \sqrt{1 - \sigma^2}\right)\Phi(u)\right\|^2.$$

If we do a Taylor expansion of $1 - \sqrt{1 - \sigma^2}$:

$$1 - \sqrt{1 - \sigma^2} = \frac{1}{2}\sigma^2 + o(\sigma^2).$$

Using this expansion, we have that

$$\left\| x - \Phi(u) + \left( 1 - \sqrt{1 - \sigma^2} \right) \Phi(u) \right\|^2$$
$$= \| x - \Phi(u) \|^2 + \sigma^2 \langle \Phi(u), x - \Phi(u) \rangle + o(\sigma^2) \langle x, \Phi(u) \rangle.$$

Then we can use the same argument as in the proof Lemma B.1 to show that the $o(\sigma^2)$ does not affect the approximation. Specifically, let

$$J := \exp \left( -\frac{\| x - \mathrm{P}_{\mathcal{M}}(x) \|^2}{2\sigma^2} \right) \frac{\left( 2\pi\sigma^2 \right)^{(n-d)/2}}{\sqrt{\left| \hat{H}(\Phi^{-1}(\mathrm{P}_{\mathcal{M}}(x)), x) \right|}},$$

and

$$K := \frac{1}{(2\pi\sigma^2)^{d/2}} \exp \left( -\frac{\| x - \Phi(u) \|^2}{2\sigma^2} \right) \exp \left( -\frac{1}{2} \langle \Phi(u), x - \Phi(u) \rangle \right) p_{\mathrm{data}}(u).$$

We have

$$\int_{\mathcal{M}} \frac{1}{(2\pi\sigma^2)^{d/2}} \exp \left( -\frac{\| x - \sqrt{1 - \sigma^2} \Phi(u) \|^2}{2\sigma^2} \right) p_{\mathrm{data}}(u) du$$
$$= \int_{\mathcal{M}} K \, du + \int_{\mathcal{M}} K \left( \exp \left( o(1) \langle \Phi(u), x \rangle \right) - 1 \right) du$$
$$\leq \int_{\mathcal{M}} K \, du + \int_{\mathcal{M}} K o(1) du$$
$$\leq J \left( p_{\mathrm{data}}(\Phi^{-1}(\mathrm{P}_{\mathcal{M}}(x))) \exp \left( -\frac{1}{2} \langle \mathrm{P}_{\mathcal{M}}(x), x - \mathrm{P}_{\mathcal{M}}(x) \rangle \right) + o(1) \right).$$

The rest of the proof follows similarly to the proof of the VE case.

Then we need to discuss the upper and lower bounds in Corollary B.1. For the upper bounds, since the manifold is compact, there exists such uniform upper bounds for $\{c_i\}_1^6$ (see Remark B.1). For the lower bounds we first consider $\lambda_{\min} \left( \hat{H}(u, x) \right)$. The part $\frac{\partial \Phi(u)}{\partial u}^{\mathsf{T}} \frac{\partial \Phi(u)}{\partial u}$ is positive definite and uniformly bounded away from zero for all $u$. The eigenvalues of other part, i.e., $\left\langle \frac{\partial \Phi(u)}{\partial u_i \partial u_j}, \Phi(u) - x \right\rangle$, may be negative. However, as long as its eigenvalues are small enough, by Weyl's inequality, we can still lower bound the smallest eigenvalue of $\hat{H}(u, x)$. The eigenvalues of $\left\langle \frac{\partial \Phi(u)}{\partial u_i \partial u_j}, \Phi(u) - x \right\rangle$, can then be bounded by $\| \nabla^2 \Phi(u) \| \, \| \Phi(u) - x \|$. Therefore, as long as the tubular neighborhood and the set $\Omega'$ is small enough, we can lower bound $\lambda_{\min} \left( \hat{H}(u, x) \right)$. Formally, let $G > 0$ be the lower bound of the smallest eigenvalue of $\frac{\partial \Phi(u)}{\partial u}^{\mathsf{T}} \frac{\partial \Phi(u)}{\partial u}$. Let $C_2$ be the uniform upper bound of $\| \nabla^2 \Phi(u) \|$, and $C_1$ be that of $\| \nabla \Phi(u) \|$. Those constants are uniform for a fixed finite atlas since the manifold is compact. Let the radius of $\Omega'$ be $r_0$. We have that in $\Omega'$, $\lambda_{\min} \left( \hat{H}(u, x) \right) \geq G - C_2(\| \Phi(u) - \mathrm{P}_{\mathcal{M}}(x) \|) + \| \mathrm{P}_{\mathcal{M}}(x) - x \|) \geq G - C_2(C_1 r_0 + \epsilon)$. Therefore, we can choose $r_0$ and $\epsilon$ small enough (but away from zero) such that $\lambda_{\min} \left( \hat{H}(u, x) \right) \geq G/2$, e.g., $\epsilon$ is the minimum of $G/(4C_2)$ and the original $\epsilon$ in the tubular neighborhood definition, and $r_0 = G/(4C_1 C_2)$. This way, $m_1$ can be lower bounded by $Gr_0^2/2$. $\qquad\square$

## B.3    PROOFS FOR SECTION 4

The results in Appendices B.1 and B.3 consider only points in $T_{\mathcal{M}}(\epsilon)$. Therefore, to use the results, we need first show that the density outside the tubular neighborhood becomes negligible as $\sigma \to 0$. In the following two lemmas, we will show the concentration of the density for $p_\sigma$ and $\exp(-f_\sigma)$.

**Lemma B.3.** *Assume Assumptions 2.1 and 2.2 holds. We have that* $\lim_{\sigma \to 0} \int_{x \in T_{\mathcal{M}}(\epsilon)} p_\sigma(x) dx = 1$.

*Proof.* We have that

$$\int_{x \in \mathbb{R}^d / T_{\mathcal{M}}(\epsilon)} p_\sigma(x) dx$$

$$= \int_{x \in \mathbb{R}^d / T_{\mathcal{M}}(\epsilon)} \int_{u \in \mathcal{M}} \frac{1}{(2\pi\sigma^2)^{d/2}} \exp\left(-\frac{\|x - \Phi(u)\|^2}{2\sigma^2}\right) p_{\text{data}}(u) du dx$$

$$= \int_{u \in \mathcal{M}} p_{\text{data}}(u) \int_{x \in \mathbb{R}^d / T_{\mathcal{M}}(\epsilon)} \frac{1}{(2\pi\sigma^2)^{d/2}} \exp\left(-\frac{\|x - \Phi(u)\|^2}{2\sigma^2}\right) dx du$$

$$\leq \int_{u \in \mathcal{M}} p_{\text{data}}(u) \int_{\|x - \Phi(u)\| \geq \epsilon} \frac{1}{(2\pi\sigma^2)^{d/2}} \exp\left(-\frac{\|x - \Phi(u)\|^2}{2\sigma^2}\right) dx du,$$

where the exchange of the integral is justified by Tonelli's theorem with the non-negativity of the integrand. The last inequality holds since any point in $\mathbb{R}^d / T_{\mathcal{M}}(\epsilon)$ is at least $\epsilon$ away from any point on the manifold. Now the inner integral is the integral of a Gaussian density with distance to the origin at least $\epsilon$. It will decay exponentially fast as $\sigma \to 0$. Let $Z$ be a standard Gaussian random variable of dimension $d$, and then the above integral is equivalent to

$$\int_{u \in \mathcal{M}} p_{\text{data}}(u) P\left(\|Z\| \geq \frac{\epsilon}{\sigma}\right) du = P\left(\|Z\| \geq \frac{\epsilon}{\sigma}\right).$$

The RHS can be shown to decay exponentially fast by the Gaussian concentrations. $\qquad\square$

**Lemma B.4.** *Assume Assumptions 2.1 and 2.2 holds. Further assume that*

$$\sup_{x \in K} \|\nabla f_\sigma(x) + \nabla \log p_\sigma(x)\| = o\left(\sigma^{-2}\right)$$

*We have that*

$$\lim_{\sigma \to 0} \int_{x \in K \setminus T_{\mathcal{M}}(\epsilon)} \exp(-f_\sigma(x)) dx = 0.$$

*Proof.* For $x \notin T_{\mathcal{M}}(\epsilon)$, the points are at least $\epsilon$ away from the manifold. Therefore, we have that

$$p_\sigma(x) \leq \int_{u \in \mathcal{M}} \frac{1}{(2\pi\sigma^2)^{d/2}} \exp\left(-\frac{\epsilon^2}{2\sigma^2}\right) p_{\text{data}}(u) du = \frac{1}{(2\pi\sigma^2)^{d/2}} \exp\left(-\frac{\epsilon^2}{2\sigma^2}\right),$$

as $p_{\text{data}}$ is a density function. Therefore, we have that

$$\exp(-f_\sigma(x)) \leq \frac{1}{(2\pi\sigma^2)^{d/2}} \exp\left(-\frac{\epsilon^2}{2\sigma^2} + o\left(\sigma^{-2}\right)\right),$$

There exists $\sigma_0$, such that for all $\sigma < \sigma_0$, the $o(\sigma^{-2})$ term is upper bounded by $\epsilon^2/4\sigma^2$. Then we have that

$$\int_{x \in K \setminus T_{\mathcal{M}}(\epsilon)} \exp(-f_\sigma(x)) dx \leq \text{Vol}(K) \frac{1}{(2\pi\sigma^2)^{d/2}} \exp\left(-\frac{\epsilon^2}{4\sigma^2}\right).$$

The RHS goes to zero as $\sigma \to 0$ as $p_{\text{data}}$ is bounded. $\qquad\square$

Now we are ready to prove our main theorems.

*Proof of Theorem 4.1.* First, since both $f_\sigma$ and $\log p_\sigma$ are $C^1$ functions on $K$, we have the that $L^\infty$ norm of their gradients is the same as the supremum. First we will show that for any $\eta \geq -2$,

$$\sup_{x \in K} \|\nabla f_\sigma(x) + \nabla \log p_\sigma(x)\| = o(\sigma^\eta) \quad \text{as } \sigma \to 0,$$

implies that

$$\sup_{x \in K} |f_\sigma(x) + \log p_\sigma(x)| = o(\sigma^\eta) \quad \text{as } \sigma \to 0.$$

Given our assumption, for any two points $x, y \in K$, there exists a finite length path, say $\gamma_{x,y}(\cdot)$ : $[0,1] \to K$ with and $\|\gamma'\|$ being upper bounded uniformly. Consider an arbitrary point $x_0 \in K$, then we have

$$\Delta_\sigma(x) \coloneqq -f_\sigma(x) - \log p_\sigma(x)$$

$$= -f_\sigma(x_0) - \log p_\sigma(x_0) + \int_0^1 (-\nabla f_\sigma(\gamma(t)) - \nabla \log p_\sigma(\gamma(t))) \cdot \gamma'(t) dt$$

$$= \Delta_\sigma(x_0) + g(x, \sigma),$$

where $\sup_x |g(x, \sigma)|$ is $o(\sigma^\eta)$ uniformly for $x \in K$ according to the assumption. Further, we have that

$$\int_{x \in K} \exp(-f_\sigma(x)) dx = \int_{x \in K} p_\sigma(x) \exp(\Delta_\sigma(x)) dx$$

$$= \int_{x \in K} p_\sigma(x) \exp(\Delta_\sigma(x_0) + g(x, \sigma)) dx,$$

which then imply that

$$\Delta_\sigma(x_0) \geq \log \int_{x \in K} \exp(-f_\sigma(x)) dx - \log \int_{x \in K} p_\sigma(x) dx - \sup_{x \in K} |g(x, \sigma)|,$$

and

$$\Delta_\sigma(x_0) \leq \log \int_{x \in K} \exp(-f_\sigma(x)) dx - \log \int_{x \in K} p_\sigma(x) dx + \sup_{x \in K} |g(x, \sigma)|.$$

The first two terms on the right-hand side is $o(1)$ as $\sigma \to 0$ as our assumption about $f_\sigma$ and $\int_K p_\sigma(x) dx \geq \int_{T_\mathcal{M}(\epsilon)} p_\sigma(x) dx \to 1$ according to Lemma B.3. Thus, $|\Delta_\sigma(x_0)|$ is $o(\sigma^\eta)$. Therefore, $|\Delta_\sigma(x)|$ is $o(\sigma^\eta)$ uniformly for $x \in K$. Further we can apply Lemma B.4 to conclude that the density of $\exp(-f_\sigma)$ concentrates in $T_\mathcal{M}(\epsilon)$ as $\sigma \to 0$.

Then, we can prove the first conclusion that the support is on the manifold. By the expansion of $\log p_\sigma$ in Theorem B.2, we have that

$$f_\sigma(x) = \frac{1}{2\sigma^2} \|x - \mathrm{P}_\mathcal{M}(x)\|^2 + o\left(1/\sigma^2\right).$$

Then we can apply Lemma B.2 with $f_\theta(x) = \sigma^2 f_\sigma(x)$, $\theta = \sigma^2$ and $\eta = \delta^2/2$ to conclude the claim.

To prove that the limiting distribution is $p_{\text{data}}$ on the manifold, we have

$$f_\sigma(x) = \frac{1}{2\sigma^2} \|x - \mathrm{P}_\mathcal{M}(x)\|^2 - \log p_{\text{data}}(\Phi^{-1}(\mathrm{P}_\mathcal{M}(x))) +$$

$$\log \sqrt{\left|\hat{H}(\Phi^{-1}(\mathrm{P}_\mathcal{M}(x)), x)\right|} + \frac{d - n}{2} \log(2\pi\sigma^2) + o(1).$$

Then we can apply Theorem B.1. Then the $f_0$ becomes the distance function (changed to local coordinates), and $f_1$ is $-\log p_{\text{data}} + \log \sqrt{\left|\hat{H}(u), \Phi(u, r))\right|}$, In addition, we note that for $r = 0$, $\sqrt{\left|\hat{H}(u), \Phi(u, r))\right|} = d\mathcal{M}(u)/du$, and therefore, we recover $p_{\text{data}}$. The $(d - n)\log(2\pi\sigma^2)$ term is simply a constant and does not affect the result after normalization. One can replace $f_\sigma$ with $f_\sigma + \frac{d-n}{2}\log(2\pi\sigma^2)$ and then apply Theorem B.1, and this does not change the distribution after normalization.

What remains is to ensure Assumption B.1 holds, especially the second condition, i.e., to ensure that the Hessian of $\|\Phi(u, r) - \Phi(u)\|^2/2$ w.r.t. $r$ is uniformly bounded away from zero. We can write $\Phi(u, r)$ as $\Phi(u) + \mathcal{N}(u)r$, where $\mathcal{N}(u)$ is the normal vector field on the manifold $\mathcal{M}$ at point $\Phi(u)$ (Weyl, 1939). We have that

$$\frac{\partial}{\partial r} \frac{\|\Phi(u, r) - \Phi(u)\|^2}{2} = \frac{\partial \Phi(u, r)}{\partial r}^\mathsf{T} (\Phi(u, r) - \Phi(u)) = \mathcal{N}(u)^\mathsf{T} \mathcal{N}(u) r = r,$$

since the columns of $\mathcal{N}(u)$ are orthonormal. Therefore, the Hessian of $\|\Phi(u,r) - \Phi(u)\|^2/2$ w.r.t. $r$ is simply the identity matrix, which satisfies the assumption.

To construct a $s(\sigma, x)$ such that the limiting distribution is arbitrarily, say $\hat{\pi}$, we let $s(\sigma, x)$ being the gradient of

$$-\frac{1}{2\sigma^2}\|x - \mathrm{P}_{\mathcal{M}}(x)\|^2 + \log \hat{\pi}(\Phi^{-1}(\mathrm{P}_{\mathcal{M}}(x))) - \log \sqrt{\left|\hat{H}(\Phi^{-1}(\mathrm{P}_{\mathcal{M}}(x)), x)\right|} + o(1).$$

The difference between $f_\sigma$ and $\log p_\sigma$ is then $\Omega(1)$. $\qquad\square$

### B.4 MANIFOLD WKB ANALYSIS OF THE STATIONARY DISTRIBUTION

A key difference between our theorem in Section 5 and the results in Section 4 is that, in the former, the density does not admit an explicit form. When $s(x, \sigma)$ is a gradient field, a closed-form expression for the density is readily available; however, this property is not guaranteed for most parameterized models, such as neural networks. We therefore resort to the WKB approximation to approximate the stationary distribution. Similarly to Appendix B.1, we first present a general framework and then apply it to our specific setting. We will show that SDE with the following form admits a stationary distribution of the form Equation (11). Interested readers may refer to Bouchet & Reygner (2016); Bonnemain & Ullmo (2019) for more details on WKB applied on Fokker-Planck equation.

We consider the following SDE:

$$dx_t = b_\theta(x_t)dt + \sqrt{2\theta}dW_t, \quad \text{with} \quad b_\theta(x) = -\nabla f_0(x) - \theta\nabla f_1(x) + \hat{b}(x, \theta),$$

or the following SDE with the same stationary distribution,

$$dx_t = \frac{b_\theta(x_t)}{\theta}dt + \sqrt{2}dW_t. \tag{13}$$

We assume that $\hat{b}(x, \theta)$ is uniformly $o(\theta)$ in $T_{\mathcal{M}}(\epsilon)$ as $\theta \to 0$. Also, we have $\arg\min f_0(x) = \mathcal{M}$. This framework is general enough to cover the cases of Theorems 5.2 and 6.1. We will see later that in these two cases, the function $f_0$ is the distance function to the manifold, and $\theta$ will be chosen differently in different cases. We make the following assumptions about the SDE.

Let $\pi_\theta(x)$ be the stationary distribution of the SDE Equation (13). First we assume the WKB ansatz:

**Assumption B.2** (Local WKB ansatz). *We assume that* $\lim_{\theta \to 0} \int_{T_{\mathcal{M}}(\epsilon)} \pi_\theta(x)dx = 1$, *and that* $\pi_\theta(x)$ *admits a local WKB form within compact set* $T_{\mathcal{M}}(\epsilon)$:

$$\pi_\theta(x) \propto \exp\left(-\frac{V(x)}{\theta}\right) c_\theta(x) \quad \text{with} \quad c_\theta(x) = c_0(x) + \hat{c}(x, \theta),$$

*where* $c_0 \in C^2(T_{\mathcal{M}}(\epsilon))$ *is positive, and* $c_\theta \to c_0$ *in* $C^2(T_{\mathcal{M}}(\epsilon))$. *We further assume that* $V \in C^3(T_{\mathcal{M}}(\epsilon))$ *admits a unique solution.*

The normalization constant can be explicitly written as

$$\int_{x \in T_{\mathcal{M}}(\epsilon)} \pi_\theta(x)dx \Big/ \int_{x \in T_{\mathcal{M}}(\epsilon)} \exp\left(-\frac{V(x)}{\theta}\right) c_\theta(x)dx,$$

since we have for $x \in T_{\mathcal{M}}(\epsilon)$,

$$\pi_\theta(x) = \pi_\theta(x) \cdot \mathbf{1}_{T_{\mathcal{M}}(\epsilon)}(x) = \pi_\theta(x \mid x \in T_{\mathcal{M}}(\epsilon))\pi_\theta(T_{\mathcal{M}}(\epsilon))$$

$$= \frac{c_\theta(x)\exp\left(-\frac{V(x)}{\theta}\right)}{\int_{x \in T_{\mathcal{M}}(\epsilon)} c_\theta(x)\exp\left(-\frac{V(x)}{\theta}\right)dx}\pi_\theta(T_{\mathcal{M}}(\epsilon)).$$

Our goal would be to solve for $V(x)$ and $c_0(x)$ with the Fokker-Planck equation. Once solved, to study the limit of $\pi_\theta$, we can use results in Appendix B.1 as

$$\pi_\theta(x) \propto \exp\left(-\frac{V(x) - \theta\log c_0(x) + o(\theta)}{\theta}\right).$$

**Theorem B.3.** *Consider the SDE described in Equation* (13). *Assume Assumption B.2 holds. Then we have that*

$$V(x) = f_0(x), \quad c_0(x) = C \exp(-f_1(x)),$$

*for some constant $C$.*

*Proof.* By Fokker-Planck equation for the stationary distribution, we have that

$$0 = \operatorname{div}\left(-b_\theta(x)\pi_\theta(x) + \theta\frac{\partial \pi_\theta(x)}{\partial x}\right).$$

By plugging in the WKB ansatz, we have that

$$
\begin{aligned}
-\operatorname{div}(b_\theta)c_\theta - \left\langle b_\theta, \frac{\partial c_\theta}{\partial x} - \frac{1}{\theta}\frac{\partial V}{\partial x}c_\theta \right\rangle + \theta \operatorname{Tr}\left[\frac{\partial^2 c_\theta}{\partial x^2}\right] - 2\left\langle \frac{\partial c_\theta}{\partial x}, \frac{\partial V}{\partial x}\right\rangle \\
- \operatorname{Tr}\left[\frac{\partial^2 V}{\partial x^2}\right]c_\theta + \frac{1}{\theta}\left\|\frac{\partial V}{\partial x}\right\|^2 c_\theta = 0.
\end{aligned}
\tag{14}
$$

Next by the method of WKB, we will equate different orders of $\theta$ in the above equation to solve for $V(x)$ and $c_0(x)$, starting from the lowest order $\theta^{-1}$. It is easier to show a function is constant, therefore, for $c_0$, we will define $\tilde{c}_0(x) = \exp(f_1(x))c_0(x)$, and try to show that it is constant.

**Order $\theta^{-1}$** In this order, we have that

$$\left\langle \frac{\partial f_0}{\partial x}, \frac{\partial V}{\partial x}\right\rangle = \left\|\frac{\partial V}{\partial x}\right\|^2.$$

This corresponds to the Hamilton-Jacobi equation typically appears in the WKB approximation. The equation gives the solution for $V(x)$ as $V(x) = f_0(x)$. Plugging this solution into Equation (14), we can get

$$
\begin{aligned}
-\operatorname{Tr}\left[-\theta\frac{\partial^2 f_1}{\partial x^2} + \frac{\partial \hat{b}}{\partial x}\right]c_\theta - \left\langle b_\theta, \frac{\partial c_\theta}{\partial x}\right\rangle + \left\langle -\theta\frac{\partial f_1}{\partial x} + \hat{b}, \frac{1}{\theta}\frac{\partial f_0}{\partial x}c_\theta\right\rangle + \theta\operatorname{Tr}\left[\frac{\partial^2 c_\theta}{\partial x^2}\right] \\
-2\left\langle \frac{\partial c_\theta}{\partial x}, \frac{\partial f_0}{\partial x}\right\rangle = 0.
\end{aligned}
$$

We will work with this equation for equating the higher orders.

**Order $\theta^0$** In this order, we have that

$$\left\langle \frac{\partial f_1}{\partial x}, \frac{\partial f_0}{\partial x}\right\rangle c_0 + \left\langle \frac{\partial c_0}{\partial x}, \frac{\partial f_0}{\partial x}\right\rangle = 0.$$

This is known as the transport equation (Bouchet & Reygner, 2016). It shows how $c_0$ changes along the gradient of $f_0$. Next, we express the equation in terms of $\tilde{c}_0$:

$$\left\langle \frac{\partial \tilde{c}_0}{\partial x}, \frac{\partial f_0}{\partial x}\right\rangle = 0. \tag{15}$$

This implies that along the gradient of $f_0$, $\tilde{c}_0$ is constant. Since the manifold $\mathcal{M}$ consists of the minimizers of $f_0$, for any point $x$ in $K$, the value of $\tilde{c}_0(x)$ is the same as the value at the corresponding minimizer $y$ on $\mathcal{M}$ following the gradient flow of $f_0$. Formally, we have

$$\tilde{c}_0(x) = \tilde{c}_0(\psi^x(+\infty)),$$

where $\psi^x(t)$ follows $d\psi^x(t)/dt = -\nabla f_0(\psi^x(t))$ with $\psi^x(0) = x$ given the initial condition $\psi^x(0) = x$. Therefore, we see that to solve for $\tilde{c}_0$, we need to know the value of it on $\mathcal{M}$. We find that the next order equation will help us to solve for $\tilde{c}_0$ on $\mathcal{M}$.

**Order $\theta^1$** In this order, if we directly find all terms in Equation (14) that are of order $\theta^1$, we will find that it includes higher order terms, e.g., $\hat{c}(x, \theta)$. However, since we only care about the solution on $\mathcal{M}$, we evaluate the equation on $\mathcal{M}$ and interestingly find that it does not include such higher order terms, as crucially the factor $\partial f_0 / \partial x$ becomes 0 at $\mathcal{M}$. Specifically, for $x \in \mathcal{M}$, we have that

$$\mathrm{Tr}\left[\frac{\partial^2 f_1}{\partial x^2}\right] c_0 + \left\langle \frac{\partial f_1}{\partial x}, \frac{\partial c_0}{\partial x} \right\rangle + \mathrm{Tr}\left[\frac{\partial^2 c_0}{\partial x^2}\right] = 0.$$

Replacing $c_0$ with $\tilde{c}_0 \exp(-f_1)$, we have that

$$\mathrm{Tr}\left[\frac{\partial^2 \tilde{c}_0}{\partial x^2}\right] - \left\langle \frac{\partial \tilde{c}_0}{\partial x}, \frac{\partial f_1}{\partial x} \right\rangle = 0.$$

Our goal here would be to solve for $\tilde{c}_0$ on $\mathcal{M}$, and apparently it would be helpful to convert the equation to the local coordinates and establish a PDE for the manifold chart coordinate $u$.

**Local coordinates** We convert the above order $\theta^1$ equation about $\tilde{c}_0$ to the local coordinates $z = (u, r)$ and get that for $r = 0$, i.e., points on $\mathcal{M}$,

$$\begin{aligned}
0 &= \frac{1}{|J|} \mathrm{div}_z \left(|J|G^{-1}\frac{\partial \tilde{c}_0}{\partial z}\right) - \left\langle \frac{\partial \tilde{c}_0}{\partial z}, G^{-1}\frac{\partial f_1}{\partial z} \right\rangle \\
&= \frac{1}{|J|} \left( \left\langle \mathrm{div}_z \left(|J|G^{-1}\right), \frac{\partial \tilde{c}_0}{\partial z} \right\rangle + \mathrm{Tr}\left[|J|G^{-1}\frac{\partial^2 \tilde{c}_0}{\partial z^2}\right] \right) - \left\langle \frac{\partial \tilde{c}_0}{\partial z}, G^{-1}\frac{\partial f_1}{\partial z} \right\rangle,
\end{aligned} \tag{16}$$

where $G = J^\mathsf{T} J$ and the divergence of a matrix is understood as the divergence of the column vectors. Note that we cannot simply conclude from the above equation that $\tilde{c}_0$ is constant, by say, the strong maximum principle, since the gradients of $\tilde{c}_0$ include not only the manifold chart coordinate $u$ but also the normal coordinate $r$. Therefore, we have to further derive a PDE about $u$ and any gradients of $\tilde{c}_0$ w.r.t. $r$ should be replaced by known functions. Fortunately those gradients can be solved by the equation we obtain at order $\theta^0$.

First, let us derive from Equation (16) a PDE about $u$:

**Lemma B.5.** *From Equation* (16)*, we have that for $r = 0$,*

$$\Delta_{\mathcal{M}} \tilde{c}_0(u) - \left\langle \frac{\partial \tilde{c}_0}{\partial u}, g^{-1}\frac{\partial f_1}{\partial u} \right\rangle + \frac{1}{\sqrt{|g|}} \left\langle \frac{\partial |J|}{\partial r}, \frac{\partial \tilde{c}_0}{\partial r} \right\rangle - \left\langle \frac{\partial \tilde{c}_0}{\partial r}, \frac{\partial f_1}{\partial r} \right\rangle + \mathrm{Tr}\left[\frac{\partial^2 \tilde{c}_0}{\partial r^2}\right] = 0, \quad (17)$$

*where $\Delta_{\mathcal{M}}$ is the Laplace-Beltrami operator on $\mathcal{M}$.*

*Proof.* Let the index $i, j$ when showing at $\partial$ be derivatives w.r.t. the $i$ or $j$-th coordinate of $u$, and let $p, q$ be the derivatives w.r.t. $r$ respectively. Any index variable that is not explicitly defined is understood to be summed over. From Equation (16), by carefully expanding the divergence, the inner product term becomes

$$\left\langle \mathrm{div}\left(|J|G^{-1}\right), \nabla \tilde{c}_0 \right\rangle \big|_{r=0}$$
$$= \sqrt{|g|}\partial_j \left[g^{-1}\right]_{i,j} \partial_i \tilde{c}_0 + \left[g^{-1}\right]_{i,j}\partial_j\sqrt{|g|}\partial_i\tilde{c}_0 - \sqrt{|g|}\left[g^{-1}\right]_{i,k}\partial_p d_{p,k}\big|_{r=0}\partial_i\tilde{c}_0 + \partial_p|J|\partial_p\tilde{c}_0.$$

For the trace term, we have

$$\mathrm{Tr}\left[|J|G^{-1}\nabla^2 \tilde{c}_0\right]\big|_{r=0} = \sqrt{|g|}\left[g^{-1}\right]_{i,j}\partial_{j,i}\tilde{c}_0 + \sqrt{|g|}\partial_{p,p}\tilde{c}_0.$$

Now we look at Equation (17). From the definition of Laplace-Beltrami operator, we have

$$\begin{aligned}
\Delta_{\mathcal{M}} \hat{c}_0(u) &= \frac{1}{\sqrt{|g|}}\partial_i\left(\sqrt{|g|}\left[g^{-1}\right]_{i,j}\partial_j\tilde{c}_0\right) \\
&= \frac{1}{\sqrt{|g|}}\partial_i\sqrt{|g|}\left[g^{-1}\right]_{i,j}\partial_j\tilde{c}_0 + \partial_i\left[g^{-1}\right]_{i,j}\partial_j\tilde{c}_0 + \left[g^{-1}\right]_{i,j}\partial_{i,j}\tilde{c}_0.
\end{aligned}$$

Since $G^{-1}$ evaluated at $r = 0$ is $\begin{bmatrix} g^{-1} & 0 \\ 0 & I \end{bmatrix}$, the term $-\left\langle \frac{\partial \tilde{c}_0}{\partial z}, G^{-1}\frac{\partial f_1}{\partial z} \right\rangle$ in Equation (16) matches $-\left\langle \frac{\partial \tilde{c}_0}{\partial u}, g^{-1}\frac{\partial f_1}{\partial u} \right\rangle - \left\langle \frac{\partial \tilde{c}_0}{\partial r}, \frac{\partial f_1}{\partial r} \right\rangle$ in Equation (17). Now compare the terms of Equation (17) and Equation (16), the only remaining term is

$$\left[g^{-1}\right]_{i,k}\partial_p d_{p,k}\big|_{r=0}\partial_i\tilde{c}_0,$$

which we will prove is 0. We will show that $\sum_p \partial_p d_{p,k}\big|_{r=0} = 0$.

Since the columns of $\mathcal{N}$ are orthonormal, we have for any $p$, $\sum_i (\mathcal{N}_{i,p})^2 = 1$. Taking derivative for both sides to $u_j$, we have for any $p, j$, $\sum_i \mathcal{N}_{i,p}\partial_j\mathcal{N}_{i,p} = 0$. We also have by definition that for any $p, j$,

$$\left[\mathcal{N}^{\mathsf{T}}\nabla\mathcal{N}r\right]_{p,j} = \mathcal{N}_{i,p}\partial_j\mathcal{N}_{i,l}r_l.$$

Using the above two results, we have for any $j$,

$$\sum_p \partial_p d_{p,j} = \sum_p \partial_p \left(\mathcal{N}_{i,p}\partial_j\mathcal{N}_{i,l}r_l\right) = \sum_p \mathcal{N}_{i,p}\partial_j\mathcal{N}_{i,p} = 0.$$

$\square$

From Equation (17), we see that it contains gradients of $\tilde{c}_0$ w.r.t. $r$, which we will solve by the order $\theta^0$ equation.

**Lemma B.6.** *From Equation* (15)*, we have that on the manifold $\mathcal{M}$,*

$$\frac{\partial \tilde{c}_0}{\partial r}(u, 0) = 0, \quad and \quad \mathrm{Tr}\left[\frac{\partial^2 \tilde{c}_0}{\partial r^2}\right](u, 0) = \left\langle h(u), \frac{\partial \tilde{c}_0}{\partial u}(u, 0)\right\rangle,$$

*where $h(u)$ does not contain the unknown function $\tilde{c}_0$.*

*Proof.* Since we care about the evaluation of the equation on $\mathcal{M}$, we start by changing the coordinates to the local coordinates $z = (u, r)$ from Equation (15) to get that

$$\left\langle \frac{\partial \tilde{c}_0}{\partial z}, G^{-1}\frac{\partial f_0}{\partial z}\right\rangle = 0.$$

Next, we compute the gradient w.r.t. $z$:

$$\frac{\partial^2 \tilde{c}_0}{\partial z^2}G^{-1}\frac{\partial f_0}{\partial z} + \frac{\partial^2 f_0}{\partial z^2}G^{-1}\frac{\partial \tilde{c}_0}{\partial z} + \left(\frac{\partial \,\mathrm{vec}\left[G^{-1}\right]}{\partial z}\right)^T \left(\frac{\partial f_0}{\partial z} \otimes \frac{\partial \tilde{c}_0}{\partial z}\right) = 0, \quad (18)$$

where $\otimes$ is the Kronecker product. When we evaluate this equation at $r = 0$, the factor $\partial f_0/\partial r$ becomes 0, $G^{-1}(u, 0) = \begin{bmatrix} g^{-1} & 0 \\ 0 & I \end{bmatrix}$ and $\frac{\partial^2 f_0}{\partial z^2}(u, 0) = \begin{bmatrix} 0 & 0 \\ 0 & \partial^2 f_0/\partial r^2(u, 0) \end{bmatrix}$. Then we have

$$\frac{\partial^2 f_0}{\partial r^2}(u, 0)\frac{\partial \tilde{c}_0}{\partial r}(u, 0) = 0.$$

Since $\frac{\partial^2 f_0}{\partial r^2}(u, 0)$ is full-rank, we have that $\frac{\partial \tilde{c}_0}{\partial r}(u, 0) = 0$.

Next, we compute gradient again for Equation (18), and evaluate at $r = 0$. Ignoring $\partial f_0/\partial z$ which is 0, we have the $i, j$-th element of the matrix is

$$\left[\frac{\partial^2 \tilde{c}_0}{\partial z^2}G^{-1}\frac{\partial^2 f_0}{\partial z^2}\right]_{i,j} + \left[\frac{\partial^2 f_0}{\partial z^2}G^{-1}\frac{\partial^2 \tilde{c}_0}{\partial z^2}\right]_{i,j} + \frac{\partial^3 f_0}{\partial z_i\partial z_k\partial z_j}\left[G^{-1}\frac{\partial \tilde{c}_0}{\partial z}\right]_k$$

$$+\frac{\partial^2 f_0}{\partial z_i\partial z_k}\frac{\partial G_{k,p}^{-1}}{\partial z_j}\frac{\partial \tilde{c}_0}{\partial z_p} + \frac{\partial \tilde{c}_0}{\partial z_k}\frac{\partial G_{k,p}^{-1}}{\partial z_i}\frac{\partial^2 f_0}{\partial z_p\partial z_j} = 0, \quad (19)$$

where $\partial \tilde{c}_0/\partial r$ is 0. The first two terms have nice structure when evaluated at $r = 0$, as

$$\frac{\partial^2 \tilde{c}_0}{\partial z^2}G^{-1}\frac{\partial^2 f_0}{\partial z^2} = \begin{bmatrix} 0 & \frac{\partial^2 \tilde{c}_0}{\partial u\partial r}\frac{\partial^2 f_0}{\partial r^2} \\ 0 & \frac{\partial^2 \tilde{c}_0}{\partial r^2}\frac{\partial^2 f_0}{\partial r^2} \end{bmatrix} \quad and \quad \frac{\partial^2 f_0}{\partial z^2}G^{-1}\frac{\partial^2 \tilde{c}_0}{\partial z^2} = \begin{bmatrix} 0 & 0 \\ \frac{\partial^2 f_0}{\partial r^2}\frac{\partial^2 \tilde{c}_0}{\partial r\partial u} & \frac{\partial^2 f_0}{\partial r^2}\frac{\partial^2 \tilde{c}_0}{\partial r^2} \end{bmatrix}.$$

We then multiply Equation (19) by matrix $\begin{bmatrix} 0 & 0 \\ 0 & \left(\frac{\partial^2 f_0}{\partial r^2}\right)^{-1} \end{bmatrix}$ from the left, and get

$$\begin{bmatrix} 0 & 0 \\ 0 & \left(\frac{\partial^2 f_0}{\partial r^2}\right)^{-1}\frac{\partial^2 \tilde{c}_0}{\partial r^2}\frac{\partial^2 f_0}{\partial r^2} \end{bmatrix} + \begin{bmatrix} 0 & 0 \\ \frac{\partial^2 \tilde{c}_0}{\partial r\partial u} & \frac{\partial^2 \tilde{c}_0}{\partial r^2} \end{bmatrix} + \text{remaining terms} = 0.$$

Since $\partial \tilde{c}_0/\partial r$ is 0, the element of the remaining terms all have one and only one factor of $\partial \tilde{c}_0/\partial u_i$ for some $i$. Taking the trace of the above equation, and we have proved the second statement. $\square$

Now we plug in Lemma B.6 to Lemma B.5, and obtain a PDE about $\tilde{c}_0(\cdot, 0)$ on $u$ whose second order derivatives are the Laplace-Beltrami operator, and the zero-th order term, i.e., the term that includes the function value $\tilde{c}_0(\cdot, 0)$, is 0. Therefore, we can conclude by strong maximum principle (Gilbarg et al., 1977, Theorem 3.5) that $\tilde{c}_0(\cdot, 0)$ is a constant. According to the equation at order $\theta^0$, we obtain that $\tilde{c}_0$ off-manifold is the same constant. □

### B.5 Proof for Section 5

We will first prove Theorem 5.1, which follows similar proof technique as Theorem 4.1, and then turn to the harder case of Theorem 5.2.

*Proof of Theorem 5.1.* The proof follows the same as Theorem 4.1, except that now we use Theorem B.1 with $\theta = \sigma^{2-\alpha}$. In this case, $f_0(x) = \|x - \mathrm{P}_{\mathcal{M}}(x)\|^2/2$, $f_1 \equiv 0$ and all other terms are asymptotically small compared to $\sigma^{2-\alpha}$. According to the proof of Theorem 4.1, the determinant of the Hessian of $f_0$ in the normal direction is the same for all $u$, therefore, we recover the uniform distribution on the manifold.

The only thing remains to verify is to ensure

$$\lim_{\sigma \to 0} \int_{\mathbb{R}^d \setminus T_{\mathcal{M}}(\epsilon)} \tilde{\pi}_\sigma(x) dx = \lim_{\sigma \to 0} \frac{\int_{\mathbb{R}^d \setminus T_{\mathcal{M}}(\epsilon)} \exp(-\sigma^\alpha f_\sigma(x)) dx}{\int_{\mathbb{R}^d} \exp(-\sigma^\alpha f_\sigma(x)) dx} = 0.$$

Since we have $\lim_{\sigma \to 0} \int_K \tilde{\pi}(x) dx \to 1$, we only need to consider within $K$. For the numerator, we can do similarly as Lemma B.4 to obtain

$$\int_{K \setminus T_{\mathcal{M}}(\epsilon)} \exp(-\sigma^\alpha f_\sigma(x)) dx \le \mathrm{Vol}(K) \left( \frac{1}{(2\pi\sigma^2)^{d/2}} \right)^{\sigma^\alpha} \exp\left( -\frac{\epsilon^2}{4\sigma^{2-\alpha}} + o\left( \sigma^{\alpha+\beta} \right) \right),$$

where $2 - \alpha > 0$ and $\alpha + \beta > 0$. There exists $\sigma_0$, such that for all $\sigma < \sigma_0$, the $o(\sigma^{\alpha+\beta})$ term is upper bounded by $\epsilon^2/8\sigma^{2-\alpha}$. Then we have the numerator upper bounded by

$$\mathrm{Vol}(K) \left( \frac{1}{(2\pi\sigma^2)^{d/2}} \right)^{\sigma^\alpha} \exp\left( -\frac{\epsilon^2}{8\sigma^{2-\alpha}} \right).$$

For the denominator, it is lower bounded by

$$\int_{T_{\mathcal{M}}(\epsilon/2)} \left( \frac{1}{(2\pi\sigma^2)^{d/2}} \right)^{\sigma^\alpha} \exp\left( -\frac{\|x - \Phi(x)\|^2}{2\sigma^{2-\alpha}} + o\left( \sigma^{\alpha+\beta} \right) \right) dx$$

$$\ge \int_{T_{\mathcal{M}}(\epsilon/2)} \left( \frac{1}{(2\pi\sigma^2)^{d/2}} \right)^{\sigma^\alpha} \exp\left( -\frac{\epsilon^2}{8\sigma^{2-\alpha}} + o\left( \sigma^{\alpha+\beta} \right) \right) dx.$$

There exists $\sigma_1$, such that for all $\sigma < \sigma_1$, the $o(\sigma^{\alpha+\beta})$ term is lower bounded by $\epsilon^2/16\sigma^{2-\alpha}$. Then the denominator is lower bounded by

$$\mathrm{Vol}(T_{\mathcal{M}}(\epsilon/2)) \left( \frac{1}{(2\pi\sigma^2)^{d/2}} \right)^{\sigma^\alpha} \exp\left( -\frac{\epsilon^2}{16\sigma^{2-\alpha}} \right).$$

Therefore, the ratio is upper bounded by

$$\frac{\mathrm{Vol}(K)}{\mathrm{Vol}(T_{\mathcal{M}}(\epsilon/2))} \exp\left( -\frac{\epsilon^2}{16\sigma^{2-\alpha}} \right),$$

which goes to zero as $\sigma \to 0$. □

Next, for Theorem 5.2, we use results in Appendix B.4 to find an approximate stationary distribution of the SDEs considered in Section 5, and then use results in Appendix B.1 to prove the main theorem.

*Proof of Theorem 5.2.* The SDE we consider can be also written as

$$dX_t = \frac{\sigma^2 s(X_t, \sigma)}{\sigma^{2-\alpha}} dt + \sqrt{2} dW_t,$$

Therefore, we want to apply Theorem B.3 with $\theta = \sigma^{2-\alpha}$ and $b_\theta = \sigma^2 s(X_t, \sigma)$. We assert that under our assumption of Theorem 5.2, we can write

$$b_\theta(x) = -\frac{\partial \|x - \mathrm{P}_\mathcal{M}(x)\|^2/2}{\partial x} + o\left(\sigma^{2-\alpha}\right),$$

meaning that $f_0 = \|x - \mathrm{P}_\mathcal{M}(x)\|^2/2$ and $f_1 \equiv 0$. We will discuss the proof of this later. If we have the above, by Theorem B.3, the stationary distribution in $T_\mathcal{M}(\epsilon)$ is given by

$$\pi_\sigma(x) \propto \exp\left(-\frac{\|x - \mathrm{P}_\mathcal{M}(x))\|^2/2}{\sigma^{2-\alpha}} + o(1)\right),$$

where the error in the prefactor is equivalent to the error in the exponent. The remaining proof follows the same as Theorem 5.1.

It remains to prove the assertion about $b_\theta$. A sufficient condition is that

$$\sup_{x \in T_\mathcal{M}(\epsilon)}\left\|\nabla \log p_\sigma(x) + \frac{1}{\sigma^2}\frac{\partial \|x - \mathrm{P}_\mathcal{M}(x)\|^2/2}{\partial x}\right\| = O(1). \tag{20}$$

Because if Equation (20) holds, we have uniformly for any $x \in T_\mathcal{M}(\epsilon)$,

$$\left\|b_\theta(x) + \frac{\partial \|x - \mathrm{P}_\mathcal{M}(x)\|^2/2}{\partial x}\right\|$$

$$= \left\|\sigma^2 s(x, \sigma) + \frac{\partial \|x - \mathrm{P}_\mathcal{M}(x)\|^2/2}{\partial x}\right\|$$

$$= \left\|\sigma^2 s(x, \sigma) - \sigma^2 \nabla \log p_\sigma(x) + \sigma^2 \nabla \log p_\sigma(x) + \frac{\partial \|x - \mathrm{P}_\mathcal{M}(x)\|^2/2}{\partial x}\right\|$$

$$\leq \left\|\sigma^2 s(x, \sigma) - \sigma^2 \nabla \log p_\sigma(x)\right\| + \left\|\sigma^2 \nabla \log p_\sigma(x) + \frac{\partial \|x - \mathrm{P}_\mathcal{M}(x)\|^2/2}{\partial x}\right\|$$

$$= o(\sigma^{2+\beta}) + O(\sigma^2)$$

$$= o(\sigma^{2-\alpha}),$$

where the last inequality holds because $\alpha > \max\{-\beta, 0\}$. In the theorem, we assumed $L^\infty\left(T_\mathcal{M}(\epsilon)\right)$ norm, which is the same as $\sup_{x \in T_\mathcal{M}(\epsilon)}$ since $s(x, \sigma)$ and $\nabla \log p_\sigma(x)$ are continuous.

Therefore, it remains to prove Equation (20). We will prove for the case of VE, and the case of VP holds with similar argument. The gradient of the distance function can be written as:

$$\frac{\partial \|x - \mathrm{P}_\mathcal{M}(x)\|^2/2}{\partial x} = \left(I - \left(\frac{\partial \mathrm{P}_\mathcal{M}(x)}{\partial x}\right)^\mathsf{T}\right)(x - \mathrm{P}_\mathcal{M}(x)) = x - \mathrm{P}_\mathcal{M}(x),$$

where the last equality holds because $x - \mathrm{P}_\mathcal{M}(x)$ is orthogonal to the manifold and the image of $\frac{\partial \mathrm{P}_\mathcal{M}(x)}{\partial x}$ is in the tangent space of the manifold (Leobacher & Steinicke, 2021). Then note that

$$\nabla \log p_\sigma(x) = \frac{\nabla p_\sigma(x)}{p_\sigma(x)} = \frac{\int_\mathcal{M} \mathcal{N}(x; u, \sigma^2 I) p_{\text{data}}(u)\frac{\Phi(u)-x}{\sigma^2}du}{\int_\mathcal{M} \mathcal{N}(x; u, \sigma^2 I) p_{\text{data}}(u)du}.$$

For the denominator, follow the same as in the proof of Theorem B.2 to obtain that

$$p_\sigma(x) = \exp\left(-\frac{\|x - \mathrm{P}_\mathcal{M}(x)\|^2}{2\sigma^2}\right)\frac{\left(2\pi\sigma^2\right)^{(n-d)/2} p_{\text{data}}(\Phi^{-1}(\mathrm{P}_\mathcal{M}(x)))}{\sqrt{\left|\hat{H}(\Phi^{-1}(\mathrm{P}_\mathcal{M}(x)), x)\right|}}(1 + O(\sigma)),$$

since Equation (12) holds and $p_{\text{data}}$ is uniformly bounded away from zero. We could do the same for the numerator, however, the $O(\sigma)$ error is not enough here. Intuitively, the numerator would be

$$\exp\left(-\frac{\|x - \mathrm{P}_\mathcal{M}(x)\|^2}{2\sigma^2}\right)\frac{\left(2\pi\sigma^2\right)^{(n-d)/2} p_{\text{data}}(\Phi^{-1}(\mathrm{P}_\mathcal{M}(x))}{\sqrt{\left|\hat{H}(\Phi^{-1}(\mathrm{P}_\mathcal{M}(x)), x)\right|}}\left(\frac{\mathrm{P}_\mathcal{M}(x) - x}{\sigma^2} + O(1/\sigma)\right).$$

Apparently, the error term is not enough to prove Equation (20).

Therefore, we turn to stronger Laplace's method result that has an error term of $O\left(\sigma^2\right)$, i.e., the $h(\theta)$ term in Corollary B.1 could be improved to $O(\theta)$ instead of $O(\sqrt{\theta})$. However, such result should have the cost of requiring the function $F$ (as the notation use in Corollary B.1) to be $C^4$ and $g$ to be $C^2$, a stronger condition[1]. Formally, we have that

$$\sigma^2 \nabla \log p_\sigma(x) + (x - \mathrm{P}_\mathcal{M}(x))$$
$$= \frac{\int_\mathcal{M} \mathcal{N}(x; u, \sigma^2 I) p_{\text{data}}(u) \left(\Phi(u) - \mathrm{P}_\mathcal{M}(x)\right) du}{\int_\mathcal{M} \mathcal{N}(x; u, \sigma^2 I) p_{\text{data}}(u) du},$$

and we want to prove its $L^\infty\left(T_\mathcal{M}(\epsilon)\right)$ norm is $O(1)$. For any $x \in T_\mathcal{M}(\epsilon)$ and $v \in \{v \mid \|v\| = 1\}$, we have that

$$v^\mathsf{T}\left(\sigma^2 \nabla \log p_\sigma(x) + (x - \mathrm{P}_\mathcal{M}(x))\right)$$
$$= \frac{\int_\mathcal{M} \mathcal{N}(x; u, \sigma^2 I) p_{\text{data}}(u) v^\mathsf{T}\left(\Phi(u) - \mathrm{P}_\mathcal{M}(x)\right) du}{\int_\mathcal{M} \mathcal{N}(x; u, \sigma^2 I) p_{\text{data}}(u) du}$$
$$= \frac{\int_\mathcal{M} \mathcal{N}(x; u, \sigma^2 I) p_{\text{data}}(u) \left(v^\mathsf{T}\left(\Phi(u) - \mathrm{P}_\mathcal{M}(x)\right) + 1\right) du}{\int_\mathcal{M} \mathcal{N}(x; u, \sigma^2 I) p_{\text{data}}(u) du} - 1.$$

The last step where we add 1 is a simple trick because the Laplace's method we will use does not allow the prefactor to be 0 at the minimizer. Next, we multiply the numerator and denominator by $\exp\left(\frac{\|x - \mathrm{P}_\mathcal{M}(x)\|^2}{2\sigma^2}\right) \frac{\sqrt{|\hat{H}(\Phi^{-1}(\mathrm{P}_\mathcal{M}(x)), x)|}}{(2\pi\sigma^2)^{(n-d)/2}}$, so that their limit does not diminishing to 0. For the numerator, we apply Majerski (2015, Theorem 2.4) with their $n = 1/\sigma^2$, $t = u$, $f(u) = \|x - \Phi(u)\|^2/2$, $\alpha = 2$ ($f(u)$ is $C^4$ since $\Phi(u)$ is $C^4$), $B_\delta$ can be selected the same as in the proof of Theorem B.2, $g(u) = p_{\text{data}}(u)(v^\mathsf{T}\left(\Phi(u) - \mathrm{P}_\mathcal{M}(x)\right) + 1)$ ($g(u)$ is $C^2$ since $p_{\text{data}}(u)$ is $C^2$), and the minimizer is $\Phi^{-1}(\mathrm{P}_\mathcal{M}(x))$. The upper boundedness of the constants can be easily verified by compactness and one can show that they are uniform for $x$ and $v$. Crucially, $g(\Phi^{-1}(\mathrm{P}_\mathcal{M}(x))) = p_{\text{data}}(\Phi^{-1}(\mathrm{P}_\mathcal{M}(x)))$ is uniformly lower bounded. The lower boundedness of $\lambda_{\min}$ can be reasoned in the same way as in the proof of Theorem B.2. Therefore, we have

$$v^\mathsf{T}\left(\sigma^2 \nabla \log p_\sigma(x) + (x - \mathrm{P}_\mathcal{M}(x))\right) = \frac{1 + O(\sigma^2)}{1 + O(\sigma^2)} - 1 = O(\sigma^2).$$

Since the bound is uniformly for $x$ and $\|v\| = 1$, we have that

$$\sup_{x \in T_\mathcal{M}(\epsilon)} \left\|\sigma^2 \nabla \log p_\sigma(x) + (x - \mathrm{P}_\mathcal{M}(x))\right\|$$
$$\leq \sup_{x \in T_\mathcal{M}(\epsilon)} \sup_{\|v\|=1} v^\mathsf{T}\left(\sigma^2 \nabla \log p_\sigma(x) + (x - \mathrm{P}_\mathcal{M}(x))\right) = O(\sigma^2),$$

which proves Equation (20).

$\square$

## B.6 Proof for Section 6

*Proof of Theorem 6.1.* The proof follows the same as Theorem 5.2, except that now we have $f_1 = v$ when applying Theorem B.3 and Theorem B.1. $\square$

## C Experimental Details and Further Experiments

### C.1 Numerical Simulations on Ellipse

**Loss function.** In our experiments, we train the score network to predict

$$\hat{s}(x, \sigma) := \sigma^2 s(x, \sigma),$$

---

[1]Weaker condition such as $C^{1,1}$ is also possible, see Majerski (2015, Theorem 2.4).

instead of $s(x, \sigma)$ directly. This formulation is more stable across noise levels, since the leading term in the score expansion is of order $1/\sigma^2$, making $\hat{s}(x, \sigma)$ an $O(1)$ target. With this choice, the training objective becomes

$$\frac{1}{2} \mathbb{E}_{u \sim p_{\text{data}}} \mathbb{E}_{x \sim \mathcal{N}(\Phi(u), \sigma^2 I)} \left[ \sigma^2 \left\| s(x, \sigma) + \frac{x - \Phi(u)}{\sigma^2} \right\|^2 \right]$$

$$= \frac{1}{2} \mathbb{E}_{u \sim p_{\text{data}}} \mathbb{E}_{x \sim \mathcal{N}(\Phi(u), \sigma^2 I)} \left[ \frac{1}{\sigma^2} \| \hat{s}(x, \sigma) + x - \Phi(u) \|^2 \right].$$

The score function $s$ is parameterized by a neural network consisting of four transformer blocks, each with hidden dimension 128.

**Data and noise.** Training data is generated from a von Mises distribution with parameter $\kappa = 1$. The injected Gaussian noise variance $\sigma^2$ is sampled from a range $\sigma \in [0.01, 50]$.

**Optimization.** We use AdamW with weight decay $1 \times 10^{-4}$ and global gradient clipping at norm 1.0. The initial learning rate is $3 \times 10^{-3}$, decayed cosine-schedule over $4 \times 10^{-4}$ steps down to 1% of its initial value, after which training continues with a constant learning rate of $4 \times 10^{-4}$. The batch size is set to 1024.

**Sampling.** For sampling, we run Langevin dynamics

$$dx_t = \hat{s}(x_t, \sigma_{\min}) \, dt + \sqrt{2\sigma_{\min}^2} \, dW_t,$$

with $\sigma_{\min} = 0.01$. This process has the same stationary distribution as

$$dx_t = s(x_t, \sigma_{\min}) \, dt + \sqrt{2} \, dW_t.$$

For the TS Langevin dynamics, the diffusion coefficient is $\sqrt{2\sigma_{\min}^{2-\alpha}}$ instead of $\sqrt{2\sigma_{\min}^2}$. We employ the Euler–Maruyama scheme with a step size of $0.1$, running $10{,}000$ steps with $10{,}000$ runs.

## C.2 IMAGE GENERATION WITH DIFFUSION MODELS

**Algorithm details.** We use a pre-trained Stable Diffusion 1.5 model with a DDPM sampler in a predictor–corrector (PC) scheme. The pre-trained network provides a denoiser $\epsilon(x, t, y)$, and the corresponding classifier-free guidance (CFG) score at time $t$ is

$$s_t(x, y) = \underbrace{\nabla_x \log p_t(x)}_{\text{unconditional score}} + w \underbrace{\left( \nabla_x \log p_t(x \mid y) - \nabla_x \log p_t(x) \right)}_{\text{conditional increment}}$$

$$= -\frac{1}{\sigma_t} \left[ \epsilon(x, t, \emptyset) + w \big( \epsilon(x, t, y) - \epsilon(x, t, \emptyset) \big) \right],$$

where $y$ is the conditioning input (prompt embedding), $w$ is the guidance scale, $\sigma_t = \sqrt{1 - \bar{\alpha}_t}$, and $\bar{\alpha}_t$ is as in Ho et al. (2020). Our tempered-score framework applies to this PC sampler by modifying only the unconditional component while leaving the guided increment unchanged:

$$\tilde{s}_t(x, y) = -\frac{1}{\sigma_t} \left[ \sigma_t^\alpha \, \epsilon(x, t, \emptyset) + w \big( \epsilon(x, t, y) - \epsilon(x, t, \emptyset) \big) \right],$$

which is consistent with Equation (8). Let $\{t_i\}$ denote the discrete reverse-time schedule. After each DDPM predictor update at level $t_i$, we perform $n_{\text{corr.}}$ *corrector* steps of Langevin dynamics with the tempered score:

$$x_{k+1} = x_k + \delta_i \, \tilde{s}_{t_i}(x_k, y) + \sqrt{2\delta_i} \, \xi_k, \qquad \xi_k \sim \mathcal{N}(0, I),$$

where the step size $\delta_i$ follows Song et al. (2021, Algorithm 5). After the entire reverse process, we apply an additional $n_{\text{corr.}}$ deterministic projection steps using the unconditional score (no guidance, no noise) to further project onto the data manifold:

$$dx_\tau = \nabla \log p_{t_0}(x_\tau) d\tau.$$

We use the same number of projection steps for both the original PC baseline and our TS to ensure a fair comparison.

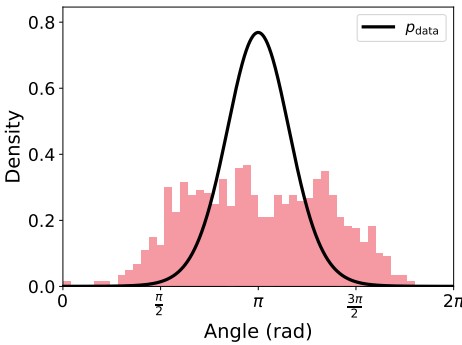
(a) Distribution generated by Diffusion Model

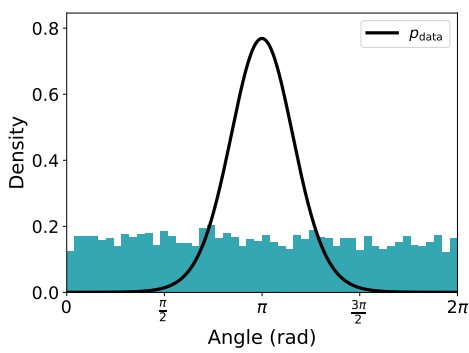
(b) Distribution generated by TS-1

Figure 4: Comparison of distributions generated with VE diffusion model versus our TS Langevin dynamics Equation (8) with $\alpha = 1$.

**Hyperparameter setting.** We adopt the default configuration of Stable Diffusion 1.5 (`https://huggingface.co/stable-diffusion-v1-5/stable-diffusion-v1-5`). Unless otherwise noted, all results in Section 7.2 use guidance scale $w = 7.5$ and 30 inference steps. For the best-results reported in Table 1, we perform a grid search over the number of corrector steps in $\{5, 10, 15, 20, 30\}$ and $\alpha \in \{0.1, 0.5, 1.0, 1.5\}$. The original PC baseline is tuned over the same set of numbers of corrector step for fairness. For CLIP evaluations, we generate 512 images per setting and downscale each to $256 \times 256$ before computing the scores.

### C.3 CONTROLLED EXPERIMENT WITH GROUND TRUTH SCORES

To empirically validate the rate separation results in Theorems 4.1 and 5.1, we designed a controlled experiment using synthetic data where the manifold and ground truth scores are known analytically.

We consider the unit circle manifold $\mathcal{M} = \{x \in \mathbb{R}^2 \mid \|x\| = 1\}$ with a Von Mises distribution $p_{\text{data}}(\theta) \propto \exp(\kappa \cos(\theta - \theta_0))$, where we used $\kappa = 4$ and $\theta_0 = \pi$. This setup allows us to compute the analytic ground truth score $s^*(x, \sigma)$. We then inject a deterministic error field $e(x)$ into the true score:

$$\hat{s}(x, \sigma) = s^*(x, \sigma) + e(x), \quad \text{with} \quad e(x) = -\nabla \left( \frac{1}{2} \left\| x - \begin{bmatrix} 1 \\ 0 \end{bmatrix} \right\|^4 \right).$$

The magnitude of this error term $e(x)$ is $O(1)$ with respect to $\sigma$.

We compare the performance of the standard reverse diffusion process against our proposed TS Langevin dynamics using this corrupted score $\hat{s}$. As shown in Figure 4, the standard reverse diffusion process using $\hat{s}$ produces samples that deviate significantly from the ground truth $p_{\text{data}}$, confirming that $O(1)$ score errors are sufficient to corrupt distributional recovery, while the TS Langevin dynamics with $\alpha = 1$ robustly recovers the uniform distribution on the circle.

### C.4 SENSITIVITY ANALYSIS OF HYPERPARAMETER $\alpha$

To evaluate the sensitivity of the hyperparameter $\alpha$, we performed an ablation study using the Stable Diffusion 1.5 model, under the same setting as in Section 7.2 of our paper. We tested $\alpha \in \{0, 0.1, 0.5, 1.0, 1.5\}$ across three prompt categories, with the number of corrector steps fixed at 10 and 20. Note that $\alpha = 0$ corresponds to the standard predictor-corrector baseline.

As shown in Tables 3 and 4, our method yields consistent improvements over the baseline ($\alpha = 0$) once $\alpha$ is sufficiently large ($\alpha \geq 0.5$), demonstrating that the performance gains are robust and not limited to a narrow hyperparameter setting. The performance is particularly stable for $\alpha \in [1.0, 1.5]$, which aligns well with our theoretical framework (Theorems 5.1 and 5.2) that guarantees convergence to the uniform distribution for any $\alpha < 2$. While we utilized $\alpha = 1$ in Table 2 for simplicity, these results suggest that slightly more aggressive tempering ($\alpha = 1.5$) can provide further gains in diversity and quality.

| Prompt | $\alpha = 0$ P-sim↑ | I-sim↓ | $\alpha = 0.1$ P-sim | I-sim | $\alpha = 0.5$ P-sim | I-sim | $\alpha = 1.0$ P-sim | I-sim | $\alpha = 1.5$ P-sim | I-sim |
|---|---|---|---|---|---|---|---|---|---|---|
| **Architecture** | 27.13 | 81.81 | 27.12 | 81.73 | 27.14 | 81.67 | 27.27 | 81.57 | **27.32** | **81.52** |
| **Furniture** | 29.30 | 81.24 | 29.32 | 81.37 | 29.33 | 81.06 | 29.58 | 80.95 | **30.16** | **80.76** |
| **Car** | 26.30 | 87.57 | 26.30 | 87.58 | 26.31 | 87.44 | 26.37 | 87.42 | **26.50** | **87.34** |

Table 3: Ablation of $\alpha$ for 10 corrector steps.

| Prompt | $\alpha = 0$ P-sim↑ | I-sim↓ | $\alpha = 0.1$ P-sim | I-sim | $\alpha = 0.5$ P-sim | I-sim | $\alpha = 1.0$ P-sim | I-sim | $\alpha = 1.5$ P-sim | I-sim |
|---|---|---|---|---|---|---|---|---|---|---|
| **Architecture** | 26.87 | 81.60 | 26.85 | 81.56 | 26.97 | 81.49 | 27.06 | **80.97** | **27.10** | 81.13 |
| **Furniture** | 28.98 | 81.72 | 28.99 | 81.65 | 29.07 | 81.40 | 29.52 | **81.15** | **30.20** | 81.39 |
| **Car** | 26.26 | 88.06 | 26.26 | 88.09 | 26.25 | 87.95 | 26.28 | 88.07 | **26.62** | **87.70** |

Table 4: Ablation of $\alpha$ 20 corrector steps.

# D  CONVERGENCE OF TS LANGEVIN

In this section, we deduce the mixing time analysis, i.e. the convergence analysis for a stochastic process, of the TS Langevin to the estimation of the Poincaré constant. The goal is to show that TS Langevin is not necessarily slower—and can in fact be significantly faster—than the standard Langevin dynamics in terms of mixing time. To carry out such an analysis, we assume that the score network is a gradient field, i.e. $s(\cdot, \sigma) = \nabla \log p_\theta$ for some parameterized density function. WLOG, we assume $p_\theta$ is normalized as the normalizing factor does not affect the velocity field $s$.

## D.1  CONVERGENCE ANALYSIS OF LANGEVIN DYNAMICS USING FUNCTIONAL INEQUALITY

To analyze the convergence of Langevin dynamics, it is customary to use a functional inequality satisfied by the invariant measure $p_\infty$ of the Langevin dynamics (Here, $p_t$ denotes the density of the process at time $t$, and $p_\infty$ is its stationary distribution. This notation differs from $p_\theta$, and the distinction should be clear from context). In this response, we focus on the Poincaré inequality (PI): We say $p_\infty$ satisfies $\text{PI}(C_{\text{PI}})$ if for all $f \in H^1(p_\infty)$ (Sobolev space weighted by $p_\infty$),

$$\int \left(f - \int f \, dp_\infty\right)^2 dp_\infty \leq \frac{1}{C_{\text{PI}}} \int |\nabla f|^2 dp_\infty,$$

where we call $C_{\text{PI}} > 0$ is the Poincaré constant.

Consider the overdamped Langevin dynamics with potential $U_\sigma : \mathbb{R}^d \to \mathbb{R}$:

$$dX(t) = -\nabla U_\sigma(X(t))dt + \sqrt{2}dW(t),$$

and let $p_t = \text{Law}(X(t))$. Under mild assumptions, $p_\infty \propto \exp(-U_\sigma)$ is the unique invariance measure of the above dynamics. If $p_\infty \propto \exp(-U_\sigma)$ satisfies $\text{PI}(C_{\text{PI}})$, then

$$\chi^2(p_t, p_\infty) \leq e^{-C_{\text{PI}}t}\chi^2(p_0, p_\infty),$$

where $\chi^2$ denotes the $\chi^2$-divergence. In particular, to ensure $\chi^2(p_t, p_\infty) \leq \eta$ for some target accuracy $\eta > 0$, it suffices to take $t = O(\frac{1}{C_{\text{PI}}} \log \frac{1}{\eta})$. *Thus, the larger the Poincaré constant, the faster the convergence.*

## D.2  ANALYZING THE EFFECT OF DRIFT SCALING TO THE POINCARÉ CONSTANT.

Under the assumptions of our paper, the comparison between the mixing of standard Langevin and TS Langevin therefore reduces to comparing their Poincaré constants. We illustrate how drift scaling affects the Poincaré constant in the simple case where the data manifold is the unit circle:

$$\mathcal{M} = \{x \in \mathbb{R}^d : \|x\| = 1\}.$$

In this case, the squared distance function can be computed in a closed form:

$$d(x) = \frac{1}{2}\text{dist}^2(x, \mathcal{M}) = \frac{1}{2}\|x - \frac{x}{\|x\|}\|^2 = \frac{1}{2}(\|x\| - 1)^2.$$

Following section 5 of our paper, we assume the score error is $O(\sigma^\beta)$ for some $-2 < \beta < 0$. Recall that we assume the learned score is a gradient field, i.e. $s(\cdot, \sigma) = \nabla \log p_\theta$. Let us further suppose that the problem dimension is $d = 2$, i.e. $x \in \mathbb{R}^2$, and the density function $p_\theta$ (corresponding to the learned score $s(\cdot, \sigma)$) has the following form

$$-\log p_\theta = \frac{1}{\sigma^2}d(x) + \sigma^\beta \phi(x), \text{ where } \phi(x) = (|x_1| - 1)^2,$$

where $x_1$ denotes the first coordinate of $x$. Clearly, this function satisfies all requirement in our paper. Crucially, such a construction ensures that the score error is $O(\sigma^\beta)$.

**Standard Langevin dynamics.** We restate the standard Langevin dynamics for the ease of reference:

$$dX(t) = \nabla \log p_\theta(X(t))dt + \sqrt{2}dW(t).$$

Without temperature scaling, the error function $\phi(x)$ introduces two separated modes $(-1, 0)$ and $(+1, 0)$. For such a multimodal measure, classical Eyring-Kramers law or the large deviation principle results imply that the Poincaré constant can scale as

$$C_{\text{PI}}^{\text{LD}} = O(\exp(-\sigma^\beta)).$$

Consequently, the mixing time of the original Langevin dynamics can become *exponentially large* as $\sigma \to 0$.

**TS Langevin.** We restate the standard Langevin dynamics for the ease of reference:

$$dX(t) = \sigma^\alpha \nabla \log p_\theta(X(t))dt + \sqrt{2}dW(t) = \nabla \log p_\theta^{\sigma^\alpha}(X(t))dt + \sqrt{2}dW(t).$$

Under mild conditions, the unique equilibrium measure is $p_\theta^{\sigma^\alpha}$. We show that, under our standing assumptions and $\alpha > -\beta$, that its Poincaré constant, denoted as $C_{\text{PI}}^{\text{TS}}$, is *uniformly bounded away from zero*, *independent of $\sigma$* for sufficiently small $\sigma$. Here we summarize the main steps:

- Recall the Holley–Stroock perturbation principle (Holley & Stroock, 1987): Let $U$ and $\tilde{U}$ be two potential functions defined on $\mathbb{R}^d$. Suppose that the corresponding Gibbs measures $p_\infty \propto \exp(-U)$ and $\tilde{p}_\infty \propto \exp(-\tilde{U})$ satisfy Poincaré inequality with constants $C_{\text{PI}}$ and $\tilde{C}_{\text{PI}}$ respectively. One has

$$\tilde{C}_{\text{PI}} \geq \exp(-osc(\tilde{U}, U))C_{\text{PI}},$$

  where the oscillation between $U$ and $\tilde{U}$ is defined as

$$osc(\tilde{U}, U) := \sup_{x \in \mathbb{R}^d}(\tilde{U} - U) - \inf_{x \in \mathbb{R}^d}(\tilde{U} - U).$$

Since $2 > \alpha > -\beta$, a Holley–Stroock perturbation argument implies that the PI constant of $p_\theta^{\sigma^\alpha}$ is comparable (up to a fixed factor) to that of the measure $\mu_d \propto \exp(-d(x)/\sigma^{2-\alpha})$ for small $\sigma$. We denote the Poincaré constant of this ideal potential as $C_{\text{PI}}^{\text{dist}}$.

A short proof for the above statement: Pick

$$\tilde{U} = \log p_\theta^{\sigma^\alpha} \text{ and } U = d(x)/\sigma^{2-\alpha}.$$

One can bound $osc(\tilde{U}, U)$ using Theorem 3.1 of our submission. Apply the above principle to yield

$$C_{\text{PI}}^{\text{TS}} \geq \exp(-O(\sigma^{\alpha+\beta}))C_{\text{PI}}^{\text{dist}} \geq \exp(-1)C_{\text{PI}}^{\text{dist}},$$

for a sufficiently small $\sigma$.

- We note that the distance function $d(x)$ is locally Polyak–Łojasiewicz, and hence one can expect the recent results (Gong et al., 2024) on the temperature-independent Poincaré constant for locally log-PL measure can be applied. The only requirement in (Gong et al., 2024) that is not satisfied by $\mu_d$ is that it is not $C^2$ at $x = 0$.

- We therefore introduce a smoothed potential

$$V_c(x) := \frac{\|x\|^2}{2} + \frac{1}{2} - \sqrt{\|x\|^2 + c^2},$$

and apply Holley–Stroock again to compare the PI constant of $\mu_d$ with that of $\mu_c \propto \exp(-V_c/\sigma^{2-\alpha})$. Choosing $c = \sigma^{3-\alpha}$, we can verify that $V_c$ satisfies the assumptions of the log-PL result (Gong et al., 2024), which implies that the corresponding Poincaré constant (denoted as $C_{\mathrm{PI}}^{\mathrm{smooth}}$) is *independent of $\sigma$*.

A short proof to bound $C_{\mathrm{PI}}^{\mathrm{dist}}$ with $C_{\mathrm{PI}}^{\mathrm{smooth}}$: Pick

$$\tilde{U}(x) = d(x)/\sigma^{2-\alpha} \text{ and } U(x) = V_c(x)/\sigma^{2-\alpha}.$$

To bound $osc(\tilde{U}, U)$, notice that

$$|d(x) - V_c(x)| = |\|x\| - \sqrt{\|x\|^2 + c^2}| = \frac{c^2}{\|x\| + \sqrt{\|x\|^2 + c^2}} \leq c = \sigma^{3-\alpha}.$$

Apply the perturbation principle to yield

$$C_{\mathrm{PI}}^{\mathrm{dist}} \geq \exp(-O(\sigma))C_{\mathrm{PI}}^{\mathrm{smooth}} \geq \exp(-1)C_{\mathrm{PI}}^{\mathrm{smooth}},$$

for a sufficiently small $\sigma$.

- Combining these comparisons shows that the Poincaré constant of $p_\theta^{\sigma^\alpha}$, i.e., $C_{\mathrm{PI}}^{\mathrm{TS}}$, differs from $C_{\mathrm{PI}}^{\mathrm{dist}}$ and $C_{\mathrm{PI}}^{\mathrm{smooth}}$ only by a constant factor.

- In this point, we discuss on proving $C_{\mathrm{PI}}^{\mathrm{smooth}}$ is independent of $\sigma$. First, we note that directly apply the result in (Gong et al., 2024) on the potential $V_c$ already yields that the Poincaré constant $C_{\mathrm{PI}}^{\mathrm{smooth}}$ is of order $\Omega(c)$: It is easy to verify the assumptions in (Gong et al., 2024), i.e. local PL, non-saddle point, growth condition beyond a compact set, and the boundedness of $|\Delta V_c|$, i.e. the absolution value of the Laplacian of $V_c$ within a compact set. We can hence directly use Theorem 2 in (Gong et al., 2024). However, the quantity $|\Delta V_c|$ is of order $\frac{1}{c}$ in this vanilla analysis and hence we would yield that the Poincaré constant $C_{\mathrm{PI}}^{\mathrm{smooth}}$ is of order $\Omega(c)$. It turns out that by exploiting the particular structure of $V_c$, we can further improve this result: We note that $|\Delta V_c|$ does *not* need to hold in the neighborhood of the local maximum set and their analysis still goes through. We hence pick this neighborhood as a ball centered around the local maximum $x = 0$ with radius $0.1$. One can see that outside of this neighborhood but within a compact set, $|\Delta V_c|$ is bounded by a $\sigma$-independent constant. Then $C_{\mathrm{PI}}^{\mathrm{smooth}}$ could be proved to be $\Omega(1)$. We highlight that even the vanilla $\Omega(c)$ bound already establishes the exponential difference between $C_{\mathrm{PI}}^{\mathrm{TS}}$ (lower bounded by a polynomial in $\sigma$) and $C_{\mathrm{PI}}^{\mathrm{LD}}$ (upper bounded by exponential of $-1/poly(\sigma)$). Of course, the $\Omega(1)$ one leads to even bigger separation.

Putting these estimates together, we see that, at least in this unit-circle example, *TS Langevin mixes strictly faster* than the original Langevin dynamics in the small-$\sigma$ regime. This illustrates that temperature-scaled Langevin is not necessarily slower—and can in fact be significantly faster—than the standard Langevin dynamics in terms of mixing time.

## D.3 A Refined Analysis for $C_{\mathrm{PI}}^{\mathrm{smooth}}$

Directly applying the result in (Gong et al., 2024), we have that $C_{\mathrm{PI}}^{\mathrm{smooth}} = \Omega(\frac{1}{\sigma})$ for a sufficiently small $\sigma$. In this subsection, we show that this can be improved to $C_{\mathrm{PI}}^{\mathrm{smooth}} = \Omega(1)$ with a small modification to the analysis of the Lyapunov function in (Gong et al., 2024).

**Proposition D.1.** *(Menz & Schlichting, 2014, Theorem 3.8) Consider the Langevin dynamics*

$$dX(t) = -\nabla V(X(t))dt + \sqrt{2\epsilon}dW(t).$$

*Define the associated infinitesimal generator $\mathcal{L}$ as*

$$\mathcal{L} := -\nabla V \cdot \nabla + \epsilon \Delta \tag{21}$$

*A function $\mathcal{W} : \mathbb{R}^d \to [1, \infty)$ is a Lyapunov function for $\mathcal{L}$ if there exists $U \subseteq \mathbb{R}^d$, $b > 0$, $\sigma > 0$, such that*

$$\forall x \in \mathbb{R}^d, \ \epsilon^{-1}\mathcal{L}\mathcal{W}(x) \leq -\sigma\mathcal{W}(x) + b1_U(x). \tag{22}$$

*Given the existence of such a Lyapunov function $\mathcal{W}$, if one further has that the truncated Gibbs measure $\mu_{\epsilon,U}$ satisfies PI with constant $\mathrm{PI}_{\epsilon,U} > 0$, the Gibbs measure $\mu_\epsilon$ satisfies PI with constant*

$$\rho_\epsilon \geq \frac{\sigma}{b + \rho_{\epsilon,U}} \rho_{\epsilon,U}. \tag{23}$$

In the context of this section, $\epsilon = \sigma^{2-\alpha}$ and $V = V_c$. In (Gong et al., 2024), the Lyapunov function is chosen to be $\mathcal{W} = \exp(\frac{V}{2\epsilon})$ and eq. (22) can be simplified to

$$\frac{\mathcal{L}\mathcal{W}}{\epsilon \mathcal{W}} = \frac{\Delta V}{2\epsilon} - \frac{|\nabla V|^2}{4\epsilon^2} \leq -\sigma + b1_U. \tag{24}$$

To establish the above inequality, Gong et al. (2024) partition the whole domain $\mathbb{R}^d$ into multiple disjoint parts: (1) $U$, (2) a neighborhood of the global minimum but outside of $U$, (3) neighborhoods of local maximum, (4) beyond a compact set that contains all critical points, and (5) the rest. We discuss our treatment of each subdomain.

- On (1), we follow the choice of $U$ in (Gong et al., 2024) so the local Poincaré inequality there directly holds.

- On (2), i.e. in the neighborhood of the global minimum (note that under the assumptions of (Gong et al., 2024), all local minima are global minima), but outside of the neighborhood $U$, we follow the argument as (Gong et al., 2024).

- On (4), Beyond a compact set that contains all the local minima and maximum, we can verify that $V_c$ above fulfilles the requirements of $V$ in (Gong et al., 2024) and hence the argument directly carries over.

- On (3), i.e. in a neighborhood of the local maximum, since the Laplacian is already negative, one can directly obtain eq. (24). Note that we will pick this neighborhood to be the ball centered at $x = 0$ with radius $0.1$ for $V_c$, denoted by $\mathbb{B}(0, 0.1)$.

- On (5), i.e. within the said compact set, but outside of the neighborhoods of the global minimum and local maximum, (Gong et al., 2024) requires the Laplacian to be bounded. We note that the analysis in (Gong et al., 2024) is a bit loose and they require the boundedness to hold on the whole compact set. However, there is no need to assume the boundedness of the Laplacian on $\mathbb{B}(0, 0.1)$ as eq. (24) is already established in (3).

Based on the above discussion, we notice that the global bound on the Laplacian of $V_c$ is only required within a compact set, but outside of $\mathbb{B}(0, 0.1)$, which is hence a constant independent of $\epsilon$. We hence obtain the $\Omega(1)$ bound on the Poincaré constant.

