# OpenReview forum: "When Scores Learn Geometry: Rate Separations under the Manifold Hypothesis"
_ICLR.cc/2026/Conference — ICLR 2026 Poster_

### Official Review · Reviewer_DVRd · 2025-10-31

**Soundness:** 4
**Presentation:** 4
**Contribution:** 4
**Rating:** 8
**Confidence:** 3

**Summary:**

The paper studies diffusion models and argues that their success is due to learning the manifold supporting the data distribution rather than finely learning the data distribution itself. The authors support their claim by showing that, in the small noise limit, the dominating term in the noised log density is determined purely by the manifold and does not involve the data distribution. The authors go on to argue that from a methodological perspective, the aim should be to estimate the manifold rather than the distribution since this task can be done successfully even with larger errors in the score estimation; in this sense, geometric learning is easier than distribution learning. Additionally, the authors propose a sampling algorithm to sample from the uniform distribution on the manifold given a score function estimator. Finally, the authors present experimental results to support their claims.

**Strengths:**

The paper makes a good case for switching paradigms to geometric learning (i.e. learning the uniform distribution on the manifold) as the score error requirements are much less stringent than that required for full distribution learning. It also seems natural to me from a generalization perspective that the manifold ought to be the target. It is also quite nice that a simple (one-line!) modification to the sampling dynamics can tolerate higher score errors and give uniformly distributed samples on the manifold. The paper is written very clearly and the authors provide good exposition. The experiments indeed show better diversity as one expects from targeting the manifold rather than the data generating distribution.

**Weaknesses:**

1) The result of Theorem 4.1 requires an $L^\infty$ bound on the score estimation error. This is quite stringent and undesirable, especially since diffusion models are typically trained via score matching which is an $L^2$ loss. Furthermore, there is some empirical evidence that $L^\infty$ assumptions are not typically satisfied in practice (e.g. see Section 3.1 of "Fast Sampling of Diffusion Models with Exponential Integrator", Zhang & Chen 2023). The authors do note this in their Limitations section, but it is a weakness nonetheless.

2) The results in Theorem 3.1 are somewhat qualitative (i.e. consistency results and weak convergence), and it would be a stronger paper if quantitative rates in some distance (e.g. Wasserstein distance) were available. For example, if E(\sigma) = o(1), is it meaningfully easier to estimate the uniform distribution on $M$ rather than the true $p_{\text{data}}$? If it is not, then perhaps the justification for geometric learning becomes weaker.

3) I'm confused about the justification for learning the uniform distribution from the Bayesian Inverse problem perspective. I agree that choosing it as the uniform yields a weaker score estimation condition for correct posterior sampling. But I'm not convinced this is by itself a good enough reason. There is strong information that is being lost by switching to the uniform distribution which might have resulted in a "better" posterior distribution but harder to sample from. It's not obviously clear why this is less desirable than a less informed prior with easier sampling requirements.

**Questions:**

1) Can the authors discuss regularity conditions on $p_{\text{data}}$? It seems that learning the uniform distribution on the manifold can get quite difficult if there are meaningful regions on $M$ where $p_{\text{data}}$ is very small. But this doesn't seem to be reflected in any of the results. Is this due to the qualitative nature of Theorem 3.1? It would be very clarifying if the authors could discuss this point in some detail.

2) Can the authors provide some high-level discussion on sample complexity? With finite data, is it the case that learning the uniform distribution on $M$ is only going to be feasible when $p_{\text{data}}$ is bounded above and below by a constant on its support (i.e. there's enough mass everywhere on the manifold)?

3) What are the difficulties in getting a quantitative version of Theorem 3.1 (say in Wasserstein distance or something else)? It would be nice to add a bit of discussion about this in the paper.

---

> ### Author Response · Authors · 2025-11-24
> **Response to Reviewer DVRd - Part 1**
>
> We thank the reviewer for the positive assessment and constructive comments. We are delighted that you appreciated the clarity of our exposition and found the motivation for switching to a geometric learning paradigm convincing. We address your specific concerns and questions point-by-point below.
>
> Regarding the reviewer's comment on the weaknesses:
> > **1. The result of Theorem 4.1 requires an $L^{\infty}$ bound on the score estimation error. This is quite stringent and undesirable, especially since diffusion models are typically trained via score matching which is an $L^2$ loss. Furthermore, there is some empirical evidence that $L^{\infty}$ assumptions are not typically satisfied in practice (e.g. see Section 3.1 of "Fast Sampling of Diffusion Models with Exponential Integrator", Zhang \& Chen 2023). The authors do note this in their Limitations section, but it is a weakness nonetheless.**
>
> We thank the reviewer for raising this important point regarding the $L^{\infty}$ assumption and the connection to the practical behavior of diffusion models. We agree that the standard training objective is $L^2$ loss, and we address the implications of this below.
> - **Relaxation to $L^2$.**
> Regarding the theoretical gap between $L^{\infty}$ and $L^2$, we kindly refer the reviewer to our detailed response to Reviewer G9vr's last question.
> In summary, we argue that the central Rate Separation phenomenon (Theorem 3.1) is an intrinsic property of the ground truth score decomposition. Consequently, the insight that geometric information is encoded at a significantly stronger scale ($\Theta(\sigma^{-2})$) than density information persists even under the $L^2$ norm.
> - **Distinction between Global $L^{\infty}$ and Local $L^{\infty}$.**
> We appreciate the reference to Zhang \& Chen (2023), which stated that
> "the score model works well in the region where $p_t(x)$ is large but suffers from large error in
> the region where $p_t(x)$ is small."
> While we agree that learning the score with low $L^{\infty}$ error globally on $\mathbb{R}^d$ is infeasible, our Theorems 4.1, 5.1, and 5.2 only require the error bound to hold on a high-density compact set $K$.
> - **Alignment with Empirical Evidence.**
> Importantly, empirical evidence showing that trained scores often fail to achieve high accuracy aligns closely with the central insight of our work. Our findings reveal that recovering the data manifold (i.e., geometric learning) does not require a perfectly accurate score; it only requires the error to be $o(\sigma^{-2})$. This observation offers an explanation for why diffusion models are able to generate high-quality samples despite large score errors—errors that may even diverge as $\sigma \to 0$.
>
> > **2. The results in Theorem 3.1 are somewhat qualitative (i.e. consistency results and weak convergence), and it would be a stronger paper if quantitative rates in some distance (e.g. Wasserstein distance) were available. For example, if $E(\sigma) = o(1)$, is it meaningfully easier to estimate the uniform distribution on $\mathcal{M}$ rather than the true $p_{\mathrm{data}}$? If it is not, then perhaps the justification for geometric learning becomes weaker.**
>
> We agree that deriving explicit convergence rates in metrics like the Wasserstein distance would further strengthen the paper, which requires complex technical work. As our primary contribution is identifying the fundamental rate separation phenomenon and the resulting algorithmic paradigm, we believe establishing these precise quantitative rates is a substantial theoretical undertaking best reserved for future work.
>
> Regarding the reviewer's specific question: If the score error is very small ($E(\sigma) = o(1)$), is geometric learning still "easier"? We thank the reviewer for this insightful question. We offer two perspectives:
> - **The region is usually impractical.** In the challenging low-noise limit ($\sigma \to 0$), achieving $o(1)$ accuracy is notoriously difficult and often computationally intractable. Our results show that geometric learning remains possible in the "intermediate" error regime ($o(1) < E(\sigma) \ll \sigma^{-2}$), where distributional learning provably fails. This is the practical regime where the "advantage" lies.
> - **Statistical complexity.** Even in the regime where the score approximation error is small ($E(\sigma)=o(1)$), we believe that geometric learning remains "meaningfully easier" from the perspective of statistical sample complexity. While formal derivation is left for future work, we conjecture that the statistical estimation error for the geometric term decays significantly faster with respect to the number of samples than the error for the density term. Consequently, with a fixed dataset, a model can estimate the geometry of the manifold with much higher precision than the specific data density.

---

> > ### Author Response · Authors · 2025-11-24
> > **Response to Reviewer DVRd - Part 2**
> >
> > > **3. I'm confused about the justification for learning the uniform distribution from the Bayesian Inverse problem perspective. I agree that choosing it as the uniform yields a weaker score estimation condition for correct posterior sampling. But I'm not convinced this is by itself a good enough reason. There is strong information that is being lost by switching to the uniform distribution which might have resulted in a "better" posterior distribution but harder to sample from. It's not obviously clear why this is less desirable than a less informed prior with easier sampling requirements.**
> >
> > We thank the reviewer for this insightful comment. We fully agree that replacing the informative data prior ($p_{\text{data}}$) with the uniform distribution on the manifold entails a loss of information, which is not always desirable. We structure our justification for this into two parts:
> > - **A "Negative Result" regarding Prior Information:**
> > We acknowledge that in many scenarios, access to informative priors is highly valuable. In such cases, our theory functions as a cautionary “negative result.” Specifically, the rate separation quantifies the cost of leveraging that information: achieving it requires score estimation with an error of order $o(1)$. This insight reveals that practitioners aiming to incorporate the true prior are confronted with a more challenging optimization problem than previously recognized, emphasizing the need for highly accurate score estimation.
> > - **The "Geometry-First" Use Case:** While information loss is a drawback in some settings, the uniform prior is actively desirable in others. When the strict $o(1)$ accuracy is unattainable (the common case), our proposed TS dynamics offer a "safe harbor." It guarantees a valid solution on the correct manifold geometry with only $o(\sigma^{-2})$ accuracy, preventing the model from collapsing to the arbitrary/corrupted modes. Also, in many scientific and fairness-sensitive applications, the density $p_{\mathrm{data}}$ reflects selection bias in the training set. For instance, in material science or molecular discovery, we often wish to explore the full space of chemically valid structures (the manifold) rather than being constrained to the specific distribution of materials discovered historically. In such cases, intentionally "unbiasing" the prior to sample uniformly is a desirable property that facilitates principled manifold exploration and diverse solution finding.
> >
> > Regarding the reviewer's questions:
> > > **1. Can the authors discuss regularity conditions on $p_{\\mathrm{data}}$? It seems that learning the uniform distribution on the manifold can get quite difficult if there are meaningful regions on $\\mathcal{M}$
> >  where $p_{\\mathrm{data}}$ is very small. But this doesn't seem to be reflected in any of the results. Is this due to the qualitative nature of Theorem 3.1? It would be very clarifying if the authors could discuss this point in some detail.**
> >
> > We appreciate the reviewer's insight regarding the impact of low-density regions. Our analysis relies on Assumption 2.2, which posits that $p_{\mathrm{data}}$ is strictly positive on the compact manifold $\mathcal{M}$; this guarantees a global lower bound $c_{\min} > 0$. As the reviewer suspects, this lower bound plays a role in the validity of the expansion. Specifically, in the proof of Theorem 3.1 (formalized as Theorem C.2 in the Appendix), the remainder term for the log-density takes the form:
> > $$\hat{p}(x, \sigma) = \log \left( 1 + \frac{O(\sigma)}{p\_{\mathrm{data}}(\Phi\^{-1}(P\_{\mathcal{M}}(x)))} \right).$$
> > Consequently, the "difficulty" of low-density regions is mathematically reflected in this term.
> > The error grows as $\log (1 + O(\sigma)/c_{\min})$.

---

> > ### Author Response · Authors · 2025-11-24
> > **Response to Reviewer DVRd - Part 3**
> >
> > > **2. Can the authors provide some high-level discussion on sample complexity? With finite data, is it the case that learning the uniform distribution on $\mathcal{M}$ is only going to be feasible when $p_{\\mathrm{data}}$ is bounded above and below by a constant on its support (i.e. there's enough mass everywhere on the manifold)?**
> >
> > - **Coverage is Essential:**
> > We agree with the reviewer’s intuition: effectively learning the manifold geometry requires the data samples to "cover" the manifold. From a statistical learning perspective, if $p_{\mathrm{data}}$ is vanishingly small in a specific region of $\mathcal{M}$ such that no samples are observed there in a finite dataset, the score function in that region cannot be learned from data.
> >
> > - **Sample Efficiency:**
> > Our theoretical results rely on a given score error bound. In practice, achieving a specific error bound is a function of sample complexity—typically, a tighter error requirement necessitates more samples.
> > Therefore, our results suggest a significant advantage for the uniform sampling objective once the coverage condition is met. For a fixed sample size $N$, the estimation error might be too large to recover the exact $p_{\mathrm{data}}$ (the $o(1)$ condition fails), but small enough to satisfy the $o(\sigma^{-2})$ condition. In other words, we can "afford" a much larger statistical error, and thus use fewer samples, to learn the geometry (uniform distribution) than to learn the density.
> >
> > > **3. What are the difficulties in getting a quantitative version of Theorem 3.1 (say in Wasserstein distance or something else)? It would be nice to add a bit of discussion about this in the paper.**
> >
> > We thank the reviewer for this suggestion. We agree that a quantitative bound (e.g., in Wasserstein distance) would be a valuable addition.
> > We discussion here a few challenges:
> > - Theorem 4.1/5.1/5.2 relies on Laplace’s method and WKB expansions. These are asymptotic tools designed to characterize the limit behavior of integrals and PDEs. Converting these into non-asymptotic quantitative bounds requires explicitly bounding the remainder terms of the asymptotic series. This would significantly complicate the analysis without necessarily changing the qualitative insight of rate separation.
> > - Our current analysis is local (inside the tubular neighborhood). A global Wasserstein bound would require a quantitative control of the probability mass outside the tubular neighborhood.

---

> > > ### Comment · Reviewer_DVRd · 2025-11-25
> > >
> > > Thanks very much to the authors for their very detailed responses to my questions! I am very satisfied with the answers. It would be great if the authors can include in the paper (where appropriate) some small expository lines commenting on these aspects. I will maintain my score.

---

### Official Review · Reviewer_LahT · 2025-11-01

**Soundness:** 2
**Presentation:** 3
**Contribution:** 2
**Rating:** 4
**Confidence:** 3

**Summary:**

In this work, the authors studies a rate separation when score-based methods learn manifold data, namely that recovering the data manifold is easier than recovering the exactly underlying distribution on the manifold, where the difficulty is measured by the error tolerance for score approximation. The authors thus argues for a paradigm shift from distribution learning to geometry learning - more specifically, targeting uniform distributions on manifolds - and introduces the Tampered Score (TS) dynamics to generate distributions on manifolds that are close to uniform.

**Strengths:**

1. The authors' main hypothesis - that recovering manifolds is easier than recovering the exact distributions on manifolds for Langevin-type dynamics with approximate score functions - is novel to my knowledge. If fully validated on score-based generative models, it can potentially shape our understanding of how such models fundamentally work.

2. The authors derive precise theorems to quantitatively characterize the hypothesis, enabled by a rigorous framework based on differential geometry and impressive mathematical techniques (not fully checked).

3. Although the idea of tempering the drift term in Langevin dynamics is not entirely new, the proposal to use it for learning uniform distributions on sub-manifolds is novel to the best of my knowledge, and also as a modification to the corrector step in predictor-corrector sampling.

4. The presentation is clear overall.

**Weaknesses:**

1. The analysis is performed towards the stationary distributions of the various score function, rather than the distributions obtained from the denoising dynamics (i.e. reverse ODE / SDE) as in score-based generative models. Hence it is unclear to what extent the theoretical insights derived for the former idealized setting also hold for the latter (even in the continuous-time setup). In the latter case, for example, even small deviations in the score function can cause the final distribution to be supported beyond the manifold.

2. In particular, I wonder whether it tends to take longer for the TS dynamics to converge to the stationary distribution than the (untempered) Langevin dynamics. Intuitively I suspect this is the case since the drift is reduced with $\alpha > 0$. As the theoretical analysis is concerned only with the stationary distributions, the results are unable to address questions regarding the rate of convergence, which are nevertheless quite relevant for practice.

3. In the paragraph after Theorem 5.2, the authors claim that the TS scheme helps with recovering "the uniform distribution on the data manifold from samples of $p_{data}$", but I don't see a justification of this by the theoretical results. In fact, all the main theoretical results require the regularity assumption (Assumption 2.2) that the noiseless distribution $p_{data}$ has a continuously differentiable density on the manifold, which does not hold in the case of empirical distributions of finite samples.

4. In Remark 4.1 on Page 5, the authors claim that the assumption of a compact support of the limiting distribution is reasonable because "many diffusion models apply clipping to generated samples". I think this is a bit misleading - applying clipping as a post-processing step after sampling doesn't affect the properties of the stationary distributions of the score functions, about which the assumptions are stated.

Typos:
1. Line 202, "difficult" -> "difficulty"
2. Line 336, "satisfie" -> "satisfy"

**Questions:**

See the "Weaknesses" section above.

---

> ### Author Response · Authors · 2025-11-19
> **Response to Reviewer LahT - Part 1**
>
> We thank the reviewer for the careful reading and constructive feedback.
>
> **We believe the current assessment mainly stems from a misunderstanding that our framework applies only to stationary distributions; clarifying that this is not the case is the central goal of our rebuttal.**
>
> We elaborate on this point in detail below, and we will make sure to incorporate this stimulating discussion into the revision.
>
>
> Regarding the reviewer's concerns:
> > **1. The analysis is performed towards the stationary distributions of the various score function, rather than the distributions obtained from the denoising dynamics (i.e. reverse ODE / SDE) as in score-based generative models. Hence it is unclear to what extent the theoretical insights derived for the former idealized setting also hold for the latter (even in the continuous-time setup). In the latter case, for example, even small deviations in the score function can cause the final distribution to be supported beyond the manifold.**
>
> There seems to be an important misunderstanding here: our analysis in Section 4 **applies directly to the denoising dynamics**. The central takeaway is that the denoising dynamics is guaranteed to generate samples on the manifold **even if the score error is infinitely large at the limit $\sigma\rightarrow 0$**, as long as it is $o(\\sigma^{-2})$. It also identifies *coverage* as the main issue at this stage: the denoising dynamics may produce Dirac-like measures, as in memorization phenomena.
>
> Addressing such degenerate scenarios, while still imposing only minimal requirements on the score error, is precisely the goal of Section 5. There, we indeed make use of stationary distributions, but only to obtain a much stronger conclusion about coverage—namely, that the limiting distribution achieves (in an appropriate sense) the best possible coverage: the uniform distribution over the manifold.
>
>
> Specifically, our paper treats two conceptually separate setups:
>
> 1. **Section 4: Analysis of existing generative models (including diffusion models).**
>    Here, $\\pi_{\\sigma}$ denotes the distribution actually **produced by a generative method at noise level $\\sigma$** (e.g., the terminal law of the reverse SDE/ODE, a predictor–corrector sampler, or discretized Langevin). As a concrete example, consider the (inference time) denoising dynamics of the VE–SDE [1]:
>    $$
>    d X(t) = - \beta(t) s_\theta(t, X(t)) dt + \sqrt{\beta(t)} d W(t),
>    $$
>    where $s_\\theta$ is the **learned score**. Let $\\pi_\\sigma$ denote the corresponding **generated distribution at noise level $\\sigma$**, i.e. starting from $X(T) \sim \mathcal{N}(0, \sigma^2(T) I_d)$ and run the above dynamics backward in time from $T$ to $t$ such that $\sigma(t) = \sigma$, where $\beta(t) = \frac{d\sigma^2(t)}{dt}$. We then denote
>    $$
>    \pi_{\sigma} = \mathrm{law}(X(t)).
>    $$
>    To quantify how far $\\pi_\sigma$ is from the **ideal** Gaussian-convolved data distribution $p_\sigma$, we consider
>    $$
>    d(\pi_\sigma, p_\sigma) := ||\nabla \log \pi_\sigma - \nabla \log p_\sigma||_{\infty},
>    $$
>    which can be shown to be a valid distance on probability measures (other metrics, such as Wasserstein distances, are also possible, but we focus on this choice since it yields the cleanest statements).
>
>    In this setting, Theorem 4.1 establishes a **scale separation** in $d(\pi_\sigma, p_\sigma)$: if $d(\pi_\sigma, p_\sigma) = o(\sigma^{-2})$, then as $\sigma \to 0$, the distribution $\pi_{\sigma}$ produced by the diffusion model **provably** concentrates on the data manifold $\mathcal{M}$. At the same time, our theory also reveals a limitation: the denoising dynamics in diffusion models may still produce degenerate samples (Dirac measures) unless we impose the much more stringent condition $d(\pi_\sigma, p_\sigma) = o(1)$. Removing this requirement, while still maintaining a meaningful (non-degenerate) distribution over the data manifold, is the main goal of Section 5. There, we show that this is achievable via a simple modification of existing denoising dynamics.
>
>    Exactly the same reasoning applies to **any** denoising dynamics beyond the mentioned VE-SDE, discretized or not, with $d(\pi_\sigma, p_\sigma)$ implicitly capturing the cumulative errors due to imperfect score learning (e.g., via denoising score matching), discretization error, etc. **In particular, we make no stationarity assumptions in Section 4.**

---

> > ### Author Response · Authors · 2025-11-28
> > **Proof Sketch for the mixing time of TS vs. vanilla Langevin dynamics - Part 2**
> >
> > 2. **TS Langevin.**
> >     We restate the standard Langevin dynamics for the ease of reference:
> >     $$
> >         dX(t) = \sigma^{\alpha}\nabla \log p_\theta (X(t)) dt + \sqrt{2} dW(t) = \nabla \log p^{\\sigma^{\\alpha}}_\theta (X(t)) dt + \sqrt{2} dW(t).
> >     $$
> >
> >     Under mild conditions, the unique equilibrium measure is $p_\\theta^{\\sigma^\\alpha}$.
> >     We show that, under our standing assumptions and $\alpha > -\beta$, that its Poincaré constant,
> >     denoted as $C_{\mathrm{PI}}^{\mathrm{TS}}$, is **uniformly bounded away from zero**, **independent of $\sigma$** for sufficiently small $\sigma$. Here we summarize the main steps:
> >
> >    * Recall the Holley–Stroock perturbation principle [4]:
> >    Let $U$ and $\tilde U$ be two potential functions defined on $\mathbb{R}^d$.
> >    Suppose that the corresponding Gibbs measures $p_\infty \propto \exp(-U)$ and $\tilde p_\infty \propto \exp(-\tilde U)$ satisfy Poincaré inequality with constants $C_\mathrm{PI}$ and $\tilde C_\mathrm{PI}$ respectively. One has
> >    $$
> >         \tilde C_{\mathrm{PI}} \geq \exp(-osc(\tilde U, U))C_{\mathrm{PI}},
> >    $$
> >    where the oscillation between $U$ and $\tilde U$ is defined as
> >    $$
> >         osc(\tilde U,  U):=\sup_{x \in \mathbb{R}^d}(\tilde U -  U) - \inf_{x \in \mathbb{R}^d}(\tilde U - U).
> >    $$
> >    Since $2 > \alpha > -\beta$, a Holley–Stroock perturbation argument implies that the PI constant of $p_\theta^{\sigma^\alpha}$ is comparable (up to a fixed factor) to that of the measure $\mu_d \propto \exp(-d(x)/\sigma^{2-\alpha})$ for small $\sigma$. We denote the Poincaré constant of this ideal potential as $C_{\mathrm{PI}}^{\mathrm{dist}}$. [Short proof]: Pick
> >    $$
> >         \tilde U = \log p_\theta^{\sigma^\alpha} \text{ and } U = d(x)/\sigma^{2-\alpha}.
> >    $$
> >    One can bound $osc(\tilde U, U)$ using Theorem 3.1 of our submission.
> >    Apply the above principle to yield
> >    $$
> >         C_{\mathrm{PI}}^{\mathrm{TS}} \geq \exp(- O(\sigma^{\alpha+\beta})) C_{\mathrm{PI}}^{\mathrm{dist}} \geq \exp(- 1) C_{\mathrm{PI}}^{\mathrm{dist}},
> >    $$
> >     for a sufficiently small $\sigma$.
> >
> >    * We note that the distance function $d(x)$ is locally Polyak–Łojasiewicz, and hence one can expect the recent results [5] on the temperature-independent Poincaré constant for locally log-PL measure can be applied. The only requirement in [5] that is not satisfied by $\mu_d$ is that it is not $C^2$ at $x=0$.
> >
> >
> >    * We therefore introduce a smoothed potential
> >      $$
> >      V_c(x) := \frac{\\|x\\|^2}{2} + \frac{1}{2} - \sqrt{\\|x\\|^2 + c^2},
> >      $$
> >      and apply Holley–Stroock again to compare the PI constant of $\mu_d$ with that of $\mu_c \propto \exp( -V_c/\sigma^{2-\alpha})$. Choosing $c = \sigma^{3-\alpha}$, we can verify that $V_c$ satisfies the assumptions of the log-PL result [5], which implies that the corresponding Poincaré constant (denoted as $C_{\mathrm{PI}}^{\mathrm{smooth}}$) is **independent of $\sigma$**.
> >      [Short proof to bound $C_{\mathrm{PI}}^{\mathrm{dist}}$ with $C_{\mathrm{PI}}^{\mathrm{smooth}}$]: Pick
> >    $$
> >         \tilde U(x) = d(x) /\sigma^{2-\alpha} \text{ and } U(x) = V_c(x)/\sigma^{2-\alpha}.
> >    $$
> >    To bound $osc(\tilde U,  U)$, notice that
> >    $$
> >         |d(x) - V_c(x)| = |\|x\| - \sqrt{\|x\|^2 + c^2}| = \frac{c^2}{\|x\| + \sqrt{\|x\|^2 + c^2}} \leq c = \sigma^{3 - \alpha}.
> >    $$
> >    Apply the perturbation principle to yield
> >    $$
> >         C_{\mathrm{PI}}^{\mathrm{dist}} \geq \exp(- O(\sigma)) C_{\mathrm{PI}}^{\mathrm{smooth}} \geq \exp(- 1) C_{\mathrm{PI}}^{\mathrm{smooth}},
> >    $$
> >     for a sufficiently small $\sigma$.
> >
> >     * Combining these comparisons shows that the Poincaré constant of $p_\theta^{\sigma^\alpha}$, i.e., $C_{\mathrm{PI}}^{\mathrm{TS}}$, differs from $C_{\mathrm{PI}}^{\mathrm{dist}}$ and $C_{\mathrm{PI}}^{\mathrm{smooth}}$ only by a constant factor.

---

> > ### Author Response · Authors · 2025-11-28
> > **Proof Sketch for the mixing time of TS vs. vanilla Langevin dynamics - Part 3**
> >
> > * In this point, we discuss on proving $C_{\mathrm{PI}}^{\mathrm{smooth}}$ is independent of $\sigma$.
> >     First, we note that directly apply the result in [5] on the potential $V_c$ already yields that the Poincaré constant $C_{\mathrm{PI}}^{\mathrm{smooth}}$ is of order $\Omega(c)$: It is easy to verify the assumptions in [5], i.e. local PL, non-saddle point, growth condition beyond a compact set, and the boundedness of $|\Delta V_c|$, i.e. the absolution value of the Laplacian of $V_c$ within a compact set.
> >     We can hence directly use Theorem 2 in [5]. However, the quantity $|\Delta V_c|$ is of order $\frac{1}{c}$ in this vanilla analysis and hence we would yield that the Poincaré constant $C_{\mathrm{PI}}^{\mathrm{smooth}}$ is of order $\Omega(c)$.
> >     It turns out that by exploiting the particular structure of $V_c$, we can further improve this result: We note that $|\Delta V_c|$ does **not** need to hold in the neighborhood of the local maximum set and their analysis still goes through. We hence pick this neighborhood as a ball centered around the local maximum $x=0$ with radius $0.1$. One can see that outside of this neighborhood but within a compact set, $|\Delta V_c|$ is bounded by a $\sigma$-independent constant. Then $C_{\mathrm{PI}}^{\mathrm{smooth}}$ could be proved to be $\Omega(1)$.
> >     We highlight that even the vanilla $\Omega(c)$ bound already establishes the exponential difference between $C_{\mathrm{PI}}^{\mathrm{TS}}$ (lower bounded by a polynomial in ${\sigma}$) and $C^\mathrm{LD}_{\mathrm{PI}}$ (upper bounded by exponential of $-1/{poly(\sigma)}$). Of course, the $\Omega(1)$ one leads to even bigger separation.
> >
> >
> > Putting these estimates together, we see that, at least in this unit-circle example, **TS Langevin mixes strictly faster** than the original Langevin dynamics in the small-$\sigma$ regime. This illustrates that temperature-scaled Langevin is not necessarily slower—and can in fact be significantly faster—than the standard Langevin dynamics in terms of mixing time.
> >
> >
> > **References**
> >
> > [4] Richard Holley and Daniel Stroock. Logarithmic Sobolev inequalities and stochastic Ising models. J. Stat. Phys. 1987
> >
> > [5] Yun Gong, Niao He, and Zebang Shen. Poincare Inequality for Local Log-Polyak-\L ojasiewicz Measures: Non-asymptotic Analysis in Low-temperature Regime. arXiv preprint, 2024.

---

> ### Author Response · Authors · 2025-11-19
> **Response to Reviewer LahT - Part 2**
>
> 2) **Section 5: Uniform sampling.** The main result of this section is that the aforementioned degeneracy can be remedied by a simple modification of the widely adopted predictor–corrector (PC) sampler. As the Reviewer correctly points out, our analysis here is carried out at stationarity. However, we argue that this is not a significant restriction, for the following reasons:
>
> - In practice, we approximate stationarity with only a finite number of iterations. As demonstrated in our experiments, the resulting algorithm still achieves superior performance in terms of both sample diversity and sample quality.
> - We also emphasize that this is fully consistent with the original influential PC sampler of [1], which likewise relies on approximate stationarity in practice; we do not introduce any stronger assumptions than those already implicit in that work. In particular, our computational cost is the same as in standard PC sampling.
>
>
>
>
> -----------------------
>
> > **2. In particular, I wonder whether it tends to take longer for the TS dynamics to converge to the stationary distribution than the (untempered) Langevin dynamics. Intuitively I suspect this is the case since the drift is reduced with $\\alpha > 0$. As the theoretical analysis is concerned only with the stationary distributions, the results are unable to address questions regarding the rate of convergence, which are nevertheless quite relevant for practice.**
>
>
> We thank the Reviewer for this remarkably insightful question: Upon careful examination, we were led to the somewhat counterintuitive conclusion that our TS Langevin is **not** inherently slower; in fact, we managed to identify concrete examples in which TS Langevin converges *provably exponentially faster* than untempered Langevin. Because the relevant arguments rely on advanced functional-inequality techniques, we will provide the proof sketch in a separate comment and summarize here only the conclusions and the key intuition.
>
>
> - **High-level intuition.**
> The key reason tempering can accelerate convergence is that *the two dynamics target different equilibrium distributions*. For example, given a score of the form $s(x,\sigma) = \nabla \log p_\theta(x)$, the standard Langevin dynamics converges to $p_\theta$, whereas TS Langevin converges to $p_\theta^{\sigma^\alpha}$ (up to normalization). This stands in contrast to typical scenarios in the discretization of Langevin sampling, where reducing the step size or the drift magnitude slows convergence because the algorithm still targets the *same* limiting distribution.
>
> - **A concrete separation.**
> We can construct explicit examples in which the mixing time of TS Langevin for the target $p_\theta^{\sigma^\alpha}$ remains *constant*, while that of the standard Langevin dynamics for $p_\theta$ is *exponentially slower*. The proof relies on a careful analysis of the associated *Poincaré constants*, which determine the spectral gaps and hence the mixing times. The main argument will be provided in a separate comment.
>
> - **General case remains open.**
> A systematic comparison of convergence rates between standard Langevin and TS Langevin on *general* data manifolds (beyond our constructed examples) is a challenging and important problem that merits a dedicated treatment. We plan to investigate this direction in future work. At present, the theoretical comparison of their convergence rates in full generality remains inconclusive.
>
> We have included the discussion and the full proof in the revised manuscript.

---

> ### Author Response · Authors · 2025-11-19
> **Response to Reviewer LahT - Part 3**
>
> > **3. In the paragraph after Theorem 5.2, the authors claim that the TS scheme helps with recovering "the uniform distribution on the data manifold from samples of $p_{\\mathrm{data}}$", but I don't see a justification of this by the theoretical results. In fact, all the main theoretical results require the regularity assumption (Assumption 2.2) that the noiseless distribution $p_{\\mathrm{data}}$ has a continuously differentiable density on the manifold, which does not hold in the case of empirical distributions of finite samples.**
>
> We believe this comment arises from a simple misunderstanding:
>
> - **$p_{\mathrm{data}}$ represents the population density:** In our theoretical framework (and specifically in Theorems 4–6), the symbol $p_{\mathrm{data}}$ denotes the *continuous population distribution* supported on the manifold, not the discrete empirical distribution defined by finite samples. Consequently, Assumption 2.2 (regularity) is applied to this underlying population density, which is compatible with existing work analyzing score error in diffusion models; see, for example, [2, 3].
>
> - **The role of samples in score estimation:** While our theoretical results analyze the properties of the diffusion process with respect to the population $p_{\mathrm{data}}$, in practice, the score function is learned using **finite samples** drawn from $p_{\mathrm{data}}$. Our claim regarding the TS scheme refers to its ability to recover the uniform distribution of the *population manifold* given a sufficiently accurate score estimator trained on these samples.
>
> - **Statistical sample complexity is future work:** As in the existing literature [2, 3], we explicitly separate the *approximation error* (how the diffusion process behaves given a learned score) from the *estimation error* (how well the score can be learned from $N$ samples). Our current results focus on the former: bounding the sampling error given a bound on the score error $E_\sigma$. As noted in our conclusion section, the question of statistical sample complexity—deriving the bound on $E_\sigma$ as a function of finite sample size $N$—is a distinct theoretical challenge that we leave for future work.
>
> -----
>
> > **4. In Remark 4.1 on Page 5, the authors claim that the assumption of a compact support of the limiting distribution is reasonable because "many diffusion models apply clipping to generated samples". I think this is a bit misleading - applying clipping as a post-processing step after sampling doesn't affect the properties of the stationary distributions of the score functions, about which the assumptions are stated.**
>
>
>
> As noted in our response to Point  1, $\pi_{\sigma}$ in **Remark 4.1** denotes the distribution produced by the generative model (including any built-in post-processing, such as clipping). In particular, it is unrelated to any notion of a stationary distribution.
>
> We agree that the wording can be improved, and we thank the Reviewer for bringing this to our attention.
>
>
> ------
>
> **References:**
>
>
> [1] Yang Song, Jascha Sohl-Dickstein, Diederik P Kingma, Abhishek Kumar, Stefano Ermon, and Ben Poole.
> Score-based generative modeling through stochastic differential equations. ICLR, 2021.
>
>
> [2] Sitan Chen, Sinho Chewi, Jerry Li, Yuanzhi Li, Adil Salim, and Anru R Zhang. Sampling is as easy as
> learning the score: theory for diffusion models with minimal data assumptions. ICLR, 2023.
>
> [3] Holden Lee, Jianfeng Lu, and Yixin Tan. Convergence of score-based generative modeling for general data
> distributions. ALT, 2023.

---

> > ### Comment · Reviewer_LahT · 2025-11-28
> >
> > Thanks to the authors for the detailed and helpful response! Some follow-up comments are added below.
> >
> > > Point 1.
> >
> > Let me clarify a bit what I meant by "_the analysis is performed towards the stationary distributions of the various score function, rather than the distributions obtained from the denoising dynamics (i.e. reverse ODE / SDE) as in score-based generative models_" in the original review. I apologize for not having made this very clear.
> >
> > I wanted mainly to emphasize the difference between two types of scores: (1) the learned score functions used in the denoising dynamics and (2) the log-gradient of the generated distributions. In the VE-SDE case described in the authors' response, for instance, they correspond to $s_{\theta}$ and $\nabla \log \pi_{\sigma}$, respectively. In the paper, the authors study the score approximation error defined as the deviation between $\nabla \log \pi_{\sigma}$ and $\nabla \log p_{\sigma}$, instead of between $s_{\theta}(\sigma, \cdot)$ and $\nabla \log p_{\sigma}$. However, wouldn't it be even more relevant to study the latter, because it is $s_{\theta}$ that the neural networks express, instead of $\nabla \log \pi_{\sigma}$, which is a theoretical object.
> >
> > The difference between $s_{\theta}(\sigma, \cdot)$ and $\nabla \log p_{\sigma}$ can also be phrased in the following way. The observation that "$p_{\sigma}$ agrees with the law of $X(t)$ where $X(t)$ is obtained from the reversed SDE driven by $\nabla \log p_{\sigma}$ as the score function" relies on the fact that $\\{ p_{\sigma} \\}\_{\sigma > 0}$ is a series of distributions obtained by Gaussian convolution, but this does *not* necessarily hold for $\\{ \pi_{\sigma} \\}\_{\sigma > 0}$, which are defined from $s_{\theta}$. In other words, even if we have access to $\nabla \log \pi_{\sigma'}$ for all $\sigma' \geq \sigma$, running the backward SDE driven by $\nabla \log \pi_{\sigma'}$ will *not* necessarily generate $\pi_{\sigma}$. Rather, to obtain $\pi_{\sigma}$, we would have to run Langevin dynamics (the corrector step) with $\nabla \log \pi_{\sigma}$ until stationarity, using the property that its stationary distribution agrees with $\pi_{\sigma}$ (as the authors already mentioned in Section 2.3). It is in this sense that I made the remark about "$\pi\_{\sigma}$ being the stationary distribution corresponding to the score function $\nabla \log \pi\_{\sigma}$".
> >
> > > Point 2.
> >
> > Very interesting results, thanks for the further analyses! Looking forward to the fully developed results with the constructed example.
> >
> > > Point 3.
> >
> > I suppose there are two ways of phrasing things:
> >
> > 1. $p_{data}$ represents the population density, as the authors described. In this case, the question of what happens in finite-data cases is left for the _estimation error_ aspect of the analysis, which is separated from the current paper's focus on the _approximation error_.
> >
> > 2. $p_{data}$ represents the empirical distribution of the training set, the view that I took in my review. In this case, we have the issue that the regularity assumption is not satisfied.
> >
> > Either way, I would still say that the case of learning from finite samples is not covered by the current work.
> >
> > > Point 4.
> >
> > Thanks for the clarification. I don't have additional questions here beyond the comment under Point 1.

---

> > > ### Author Response · Authors · 2025-12-02
> > >
> > > We thank the reviewer for the follow-up comments. To ensure clarity for the disucssion, we briefly recap the definitions used in our framework: $\pi_{\sigma}$ is the marginal distribution generated by the sampler (e.g., the reverse SDE/ODE at time $t$), and $p_{\mathrm{data}}$ is the continuous population distribution supported on the manifold. Those defintions are standard in the literature, and we will make them explicit in the revised manuscript.
> > >
> > > Regarding Point 1 (The interpretation of the score $\nabla \log \pi_\sigma$):
> > > - **Direct relevance to generative practice.** We emphasize that $\pi_{\sigma}$ is defined simply as the distribution actually produced by the diffusion model at a specific noise level. It represents the immediate output of the sampler and does not require any additional simulation (such as running Langevin dynamics to stationarity). The discussion of stationary distributions in Section 2.3 was specific to the uniform sampling method in Section 5.
> > > - **Practical implication.** Because $\pi_{\sigma}$ corresponds directly to the samples generated by the model in practice, our analysis in Section 4 directly characterizes the geometry/density recovery of real-world diffusion outputs.
> > > - **Alignment with standard theory.** Existing theoretical works, e.g., [6,7], establish bounds between the generated law $\pi_{\sigma}$ and the target $p_{\sigma}$ in various metrics (TV, KL, Wasserstein). Our work operates in this same standard setting by considering the difference of their scores. Therefore, although a network does not learn this distribution or its score directly (as in the reviewer's concern), the error of $\pi_{\sigma}$ is widely accepted as the correct object of study. Our specific contribution is going beyond standard bounds to reveal the rate separation phenomenon: showing that geometry is recovered at a much faster rate ($o(\sigma^{-2})$) than the density. We believe this is novel and insightful.
> > >
> > > **References:**
> > >
> > > [6] Valentin De Bortoli. Convergence of denoising diffusion models under the manifold hypothesis. TMLR, 2022.
> > >
> > > [7] Sitan Chen, Sinho Chewi, Jerry Li, Yuanzhi Li, Adil Salim, and Anru R Zhang. Sampling is as easy
> > > as learning the score: theory for diffusion models with minimal data assumptions. ICLR, 2023.

---

> ### Author Response · Authors · 2025-11-28
> **Proof Sketch for the mixing time of TS vs. vanilla Langevin dynamics - Part 1**
>
> **Convergence analysis of Langevin dynamics using functional inequality** To analyze the convergence of Langevin dynamics, it is customary to use a functional inequality satisfied by the invariant measure $p_\infty$ of the Langevin dynamics (Here, $p_t$ denotes the density of the process at time $t$, and $p_\infty$ is its stationary distribution. This notation differs from $p_\theta$, and the distinction should be clear from context). In this response, we focus on the Poincaré inequality (PI): We say $p_\infty$ satisfies $\mathrm{PI}(C_\mathrm{PI})$ if for all $f \in H^1(p_\infty)$ (Sobolev space weighted by $p_\infty$),
> $$
> \int \bigl(f - \textstyle\int f d p_\infty\bigr)^2  d p_\infty
> \leq \frac{1}{C_{\mathrm{PI}}}  \int \bigl|\nabla f\bigr|^2  d p_\infty,
> $$
> where we call $C_{\mathrm{PI}} > 0$ is the Poincaré constant.
>
> Consider the overdamped Langevin dynamics
> $$
> dX(t) = -\nabla U_{\sigma}(X(t))dt + \sqrt{2}dW(t),
> $$
> and let $p_t = \mathrm{Law}(X(t))$. If the invariant measure $p_\infty \propto \exp(-U_\sigma)$ satisfies a PI with constant $C_{\mathrm{PI}} = \rho_\sigma$, then
> $$
> \chi^2(p_t, p_\infty) \le e^{-\rho_\sigma t}\chi^2(p_0, p_\infty),
> $$
> where $\chi^2$ denotes the $\chi^2$-divergence. In particular, to ensure $\chi^2(p_t, p_\infty) \le \eta$ for some target accuracy $\eta > 0$, it suffices to take $t = O(\frac{1}{C_{\mathrm{PI}}}  \log \frac{1}{\eta})$. Thus, the **larger** the Poincaré constant, the **faster** the convergence.
>
> **Analyzing the effect of drift scaling to the Poincaré constant.** Under the assumptions of our paper, the comparison between the mixing of standard Langevin and TS Langevin therefore reduces to comparing their Poincaré constants. We illustrate how drift scaling affects the Poincaré constant in the simple case where the data manifold is the unit circle:
> $$
> \mathcal{M} = \\{x \in \mathbb{R}^d : \\|x\\| = 1\\}.
> $$
> In this case, the squared distance function can be computed in a closed form:
> $$
>     d(x) = \frac{1}{2}\mathrm{dist}^2(x, \mathcal{M}) = \frac{1}{2} \\|x - \frac{x}{\\|x\\|}\\|^2 = \frac{1}{2}(\\|x\\|-1)^2.
> $$
> Following section 5 of our paper, we assume the score error is $O(\sigma^{\beta})$ for some $-2 < \beta < 0$.
> Recall that we assume the learned score is a gradient field, i.e. $s(\cdot, \sigma) = \nabla \log p_\theta$. Let us further suppose that the problem dimension is $d=2$, i.e. $x\in\mathbb{R}^2$, and the density function $p_\theta$ (corresponding to the learned score $s(\cdot, \sigma)$) has the following form
> $$
>      -\log p_\theta = \frac{1}{\sigma^2} d(x) + \sigma^{\beta} \phi(x), \mathrm{where}\ \phi(x) = (|x_1| - 1)^2,
> $$
> where $x_1$ denotes the first coordinate of $x$. Clearly, this function satisfies all requirement in our paper.
> Crucially, such a construction ensures that the score error is $O(\sigma^{\beta})$.
>
> 1. **Standard Langevin dynamics.**
>     We restate the standard Langevin dynamics for the ease of reference:
>     $$
>         dX(t) = \nabla \log p_\theta (X(t)) dt + \sqrt{2} dW(t).
>     $$
>    Without temperature scaling, the error function $\phi(x)$ introduces two separated modes $(-1, 0)$ and $(+1, 0)$. For such a multimodal measure, classical Eyring-Kramers law or the large deviation principle results imply that the Poincaré constant can scale as
>    $$
>    C^\mathrm{LD}_{\mathrm{PI}} = O(\exp\bigl(-\sigma^{\beta}\bigr)).
>    $$
>     Consequently, the mixing time of the original Langevin dynamics can become **exponentially large** as $\sigma \to 0$.

---

### Official Review · Reviewer_G9vr · 2025-11-01

**Soundness:** 3
**Presentation:** 3
**Contribution:** 3
**Rating:** 8
**Confidence:** 3

**Summary:**

While score-based models such as diffusion models are typically thought as approximating the data distribution, this paper hypothesizes that---assuming the popular manifold hypothesis---their success arises from implicitly learning the manifold instead ("geometry learning").
The authors provide theoretical results that show that (1) recovering the data distribution (as the noise level $\sigma \to 0$) is *difficult* as it requires a strict $o(1)$ error on the learned scores, while (2) mere *concentration* on the data support (i.e., the manifold) can be achieved with a much larger score error of $o(\sigma^{-2})$. However, in the latter case, it is shown that the learned distribution supported on the manifold can be arbitrary. To this end, the paper proposes a simple modification to Langevin-based samplers that draws from the *uniform distribution* on the manifold, which still only requires $o(\sigma^{-2})$ score errors. Similarly, the authors show that in Bayesian inverse problems, sampling from the posterior induced by a prior that is uniform on the manifold also tolerates $o(\sigma^{-2})$ score error, while $o(1)$ error must be assumed when picking the (Gaussian smoothed) data distribution as the prior.
The author's theoretical results are empirically validated using low-dimensional synthetic data, as well as pre-trained large-scale diffusion models.

**Strengths:**

+ **Motivation and Relevance.** The work is clearly motivated: Understanding learning dynamics in score-based models is highly relevant for contemporary generative modeling and Bayesian inference, but also for potential downstream tasks such as using the learned density for out-of-distribution detection etc.
+ **Presentation.** The theoretical results are presented well. The Taylor expansion in Theorem 3.1 gives a good intuition for the results, while being less technical.
+ **Novel, important theoretical results.** The rate separation results (Theorem 4.1) are novel and important for this line of research. Under standard assumptions, the results give new insight into why learning geometry information is "easier" than distribution learning in score-based models.
+ **Uniform-on-manifold sampling.** Tampered Score (TS) Langevin dynamics is a surprisingly straightforward adaptation that comes with relatively strong theoretical guarantees regarding convergence to the uniform distribution on the manifold. Specifically, the guarantees also hold in the practical case where $s(\cdot, \sigma)$ is a non-conservative vector field (as in most diffusion models that directly output the score).

**Weaknesses:**

+ **Empirical Validation.**
	+ The empirical results are limited in scope, and do not directly support the main rate separation results in Theorem 4.1. The paper would benefit from such experiments (e.g., synthetic data with known manifold and ground truth scores, controlled injection of score error, systematic analysis on how manifold concentration and distribution recovery behaves).
	+ The experiment in Section 7.1 would benefit from a quantitative evaluation of the qualitative results in Figure 2. Moreover, higher dimensional synthetic experiments (with known manifolds) would shed light onto more practical settings, as score errors can be very different in 2D when compared to high-dimensional problems.
	+ The experiment in Section 7.2 seems weak as it shows only slight quantitative improvements on merely three prompts. No error bars are provided, and it is unclear if the results are significant. While in Table 2, $\alpha=1$ is fixed, in Table 1, TS was tuned on both the number of correction steps *and* $\alpha$ , while PC was only tuned on the former, which makes the tuning budget unbalanced.

+ **Discretization Error.** All theoretical analysis assume continuous time models, while discretization error is disregarded. However, in practice, discretization error has an empirically large influence on the performance. Extending the theory in this direction would make it more practically applicable.

**Questions:**

+ How sensible are the results in Section 7 to the choice of $\alpha$?
+ A possible relaxation of the $L^{\infty}$ error assumption is mentioned in the limitations. Can you elaborate on the importance of this relaxation? How could the presented results be related to e.g. the denoising score matching loss (Fisher divergence) of trained models?

---

> ### Author Response · Authors · 2025-11-24
> **Response to Reviewer G9vr - Part 1**
>
> We thank the reviewer for the detailed and encouraging feedback. We are glad that the motivation, clarity of presentation, rate separation results, and the TS Langevin scheme, including its applicability to non-conservative scores, were found to be novel and relevant.
>
> Regarding the reviewer's comment on the weaknesses:
>
> 1. **Empirical Validation**
>
> > **The empirical results are limited in scope, and do not directly support the main rate separation results in Theorem 4.1. The paper would benefit from such experiments (e.g., synthetic data with known manifold and ground truth scores, controlled injection of score error, systematic analysis on how manifold concentration and distribution recovery behaves).**
>
> We thank the reviewer for this constructive suggestion.
> In response, we have conducted a new analysis on a synthetic manifold (unit circle with Von Mises distribution) where we have access to the closed-form score. We injected a controlled score error of order $\Theta(1)$. According to our theory (Theorem 3.1), this error scale is sufficient to disrupt the $\Theta(1)$ information required for density recovery, but is negligible compared to the $\Theta(\sigma^{-2})$ term governing manifold concentration and geometry.
>
> The results, detailed in the new Appendix D.3 in the revised paper confirm our theoretical predictions:
>
> - Distribution Learning Fails: When running the standard reverse process with the corrupted score, the generated distribution deviates significantly from the ground truth $p_{\mathrm{data}}$, confirming that distributional recovery is highly sensitive to $O(1)$ errors (validating Theorem 4.1, part 3).
> - Geometric Learning Succeeds: Conversely, when running our proposed Tempered Score (TS) Langevin dynamics ($\alpha=1$) with the exact same corrupted score, the method robustly recovers the uniform distribution on the manifold (validating Theorem 5.1).
> We have added these results to Appendix D.3 of the revised paper.
>
>
> > **The experiment in Section 7.1 would benefit from a quantitative evaluation of the qualitative results in Figure 2. Moreover, higher dimensional synthetic experiments (with known manifolds) would shed light onto more practical settings, as score errors can be very different in 2D when compared to high-dimensional problems.**
>
> We appreciate the insight. We agree that extending the synthetic experiments to higher dimensions and including quantitative metrics would be beneficial, and we plan to explore these settings to further verify our theoretical results.
>
> > **The experiment in Section 7.2 seems weak as it shows only slight quantitative improvements on merely three prompts. No error bars are provided, and it is unclear if the results are significant. While in Table 2,
> $\alpha = 1$ is fixed, in Table 1, TS was tuned on both the number of correction steps and $\alpha$, while PC was only tuned on the former, which makes the tuning budget unbalanced.**
>
> We appreciate the reviewer's detailed scrutiny of our experimental setup. We agree that evaluating the statistical significance and stability of the results in Section 7.2 is important, and we plan to investigate this further.
>
> Regarding the concern about the tuning budget for $\alpha$, we would like to clarify two key points:
> - In Table 2, we explicitly fixed $\alpha=1$ (performing no tuning on $\alpha$) and compared our method against the baseline across varying numbers of corrector steps. The results demonstrate that our method achieves consistent improvements over PC without requiring specific tuning of $\alpha$.
> - As we will detail in our response to your Question 1 (in the Questions section), our ablation study confirms that the improvement of TS over PC remains consistent across a wide range of $\alpha$ values, suggesting that the performance gain is robust and not merely the result of a larger tuning budget.
>
>
> 2. **Discretization Error**
>
> > **All theoretical analysis assume continuous time models, while discretization error is disregarded. However, in practice, discretization error has an empirically large influence on the performance. Extending the theory in this direction would make it more practically applicable.**
>
> We appreciate the reviewer raising the practical issue of discretization.
> - Our theoretical framework in Section 4 is robust to discretization. The error term $E_\sigma := ||s_{\pi_\sigma} - s^\*||\_{L^\infty}$ measures the deviation of the actual effective score of the generated distribution $\pi_\sigma$ from the ideal score. In this view, discretization errors can be accounted in $E_\sigma$. Our theorem implies that while discretization error might distort the final density, the recovery of the manifold geometry is robust to much larger discretization errors (up to $o(\sigma^{-2})$).
> - While we focus on the continuous-time analysis in Section 5, we agree with the reviewer and mentioned in the conclusion that explicitly modeling the discretization error is indeed a valuable direction for future work.

---

> > ### Author Response · Authors · 2025-11-24
> > **Response to Reviewer G9vr - Part 2**
> >
> > Regarding the reviewer's questions:
> > > **1. How sensible are the results in Section 7 to the choice of $\alpha$?**
> >
> > We thank the reviewer for raising this important practical question. To evaluate the sensitivity of the hyperparameter $\alpha$, we performed an ablation study using the Stable Diffusion 1.5 model, under the same setting as in Section 7.2 of our paper. We tested $\alpha \in \\{0, 0.1, 0.5, 1.0, 1.5\\}$ across three prompt categories, with the number of corrector steps fixed at 10 and 20.
> >
> > The tables below summarize the CLIP Prompt Similarity (P-sim, higher is better) and Inter-Image Similarity (I-sim, lower is better). Note that $\alpha=0$ corresponds to the standard predictor-corrector baseline.
> >
> > Table 1: Ablation for 10 corrector steps
> >
> > | Category | **$\alpha = 0$** | | **$\alpha = 0.1$** | | **$\alpha = 0.5$** | | **$\alpha = 1.0$** | | **$\alpha = 1.5$** | |
> > | :--- | :---: | :---: | :---: | :---: | :---: | :---: | :---: | :---: | :---: | :---: |
> > | **Metric** | P-sim $\uparrow$ | I-sim $\downarrow$ | P-sim $\uparrow$ | I-sim $\downarrow$ | P-sim $\uparrow$ | I-sim $\downarrow$ | P-sim $\uparrow$ | I-sim $\downarrow$ | P-sim $\uparrow$ | I-sim $\downarrow$ |
> > | **Architecture** | 27.13 | 81.81 | 27.12 | 81.73 | 27.14 | 81.67 | 27.27 | 81.57 | **27.32** | **81.52** |
> > | **Furniture** | 29.30 | 81.24 | 29.32 | 81.37 | 29.33 | 81.06 | 29.58 | 80.95 | **30.16** | **80.76** |
> > | **Car** | 26.30 | 87.57 | 26.30 | 87.58 | 26.31 | 87.44 | 26.37 | 87.42 | **26.50** | **87.34** |
> >
> > Table 2: Ablation for 20 corrector steps
> >
> > | Category | **$\alpha = 0$** | | **$\alpha = 0.1$** | | **$\alpha = 0.5$** | | **$\alpha = 1.0$** | | **$\alpha = 1.5$** | |
> > | :--- | :---: | :---: | :---: | :---: | :---: | :---: | :---: | :---: | :---: | :---: |
> > | **Metric** | P-sim $\uparrow$ | I-sim $\downarrow$ | P-sim $\uparrow$ | I-sim $\downarrow$ | P-sim $\uparrow$ | I-sim $\downarrow$ | P-sim $\uparrow$ | I-sim $\downarrow$ | P-sim $\uparrow$ | I-sim $\downarrow$ |
> > | **Architecture** | 26.87 | 81.60 | 26.85 | 81.56 | 26.97 | 81.49 | 27.06 | **80.97** | **27.10** | 81.13 |
> > | **Furniture** | 28.98 | 81.72 | 28.99 | 81.65 | 29.07 | 81.40 | 29.52 | **81.15** | **30.20** | 81.39 |
> > | **Car** | 26.26 | 88.06 | 26.26 | 88.09 | 26.25 | 87.95 | 26.28 | 88.07 | **26.62** | **87.70** |
> >
> > As shown in the tables, our method yields consistent improvements over the baseline ($\alpha=0$) once $\alpha$ is sufficiently large ($\alpha \ge 0.5$), demonstrating that the performance gains are robust and not limited to a narrow hyperparameter setting. The performance is particularly stable for $\alpha \in [1.0, 1.5]$, which aligns well with our theoretical framework (Theorem 5.1/5.2) that guarantees convergence to the uniform distribution for any $\alpha < 2$. While we utilized $\alpha=1$ in the main text for simplicity, these results suggest that slightly more aggressive tempering ($\alpha=1.5$) can provide further gains in diversity and quality.

---

> > ### Author Response · Authors · 2025-11-24
> > **Response to Reviewer G9vr - Part 3**
> >
> > > **2. A possible relaxation of the $L^{\\infty}$ error assumption is mentioned in the limitations. Can you elaborate on the importance of this relaxation? How could the presented results be related to e.g. the denoising score matching loss (Fisher divergence) of trained models?**
> >
> > We thank the reviewer for raising this insightful point regarding the relaxation of the error assumption. We agree that connecting our analysis to the $L^2$ error (Fisher divergence) is a vital step in bridging the gap between our theoretical findings and practical training objectives. We address the importance of this relaxation and the connection to Fisher divergence below:
> >
> > - **Importance of the Relaxation.**
> > As the reviewer correctly notes, diffusion models are trained via Denoising Score Matching, which minimizes a Fisher-divergence-type loss (a weighted $L^2$ norm of the score error). Relaxing this to an $L^2$ assumption would align our theoretical guarantees with the actual training objective of standard diffusion models.
> > - **The Rate Separation Insight Persists in $L^2$.**
> > While a rigorous proof of the weak limit under $L^2$ error requires handling technical challenges,
> > we argue that the fundamental insight of rate separation holds in the $L^2$ sense.
> >   - *Expansion of $\log p_{\sigma}$:* Our Theorem 3.1 establishes that the ground truth log-density $\log p_{\sigma}$ expands into a geometric term (scaled by $\sigma^{-2}$) and a density term (scaled by $1$), with a remainder that is uniformly bounded. Because the remainder is uniformly bounded, this expansion naturally holds in the $L^2$ norm as well.
> >   - *Triangle Inequality Argument:* Let $s_{\pi_{\sigma}}$ be the score corresponding to the generated distribution and $s^\*$ be the ground truth score. If the model is trained such that the Fisher divergence (score matching loss) is bounded, i.e., $\\|s_{\pi_{\sigma}} - s^\*\\|\_{L^2} \le \delta$, then by the triangle inequality:
> >   $$\\|s_{\pi_{\sigma}} - \underbrace{\text{GeometricTerm}}\_{\Theta\left(\sigma^{-2}\right)} \\|\_{L^2} \le \delta + \\|s^\* - \text{GeometricTerm} \\|\_{L^2},$$
> > where the second term is of order $o(\sigma^{-2})$ as per Theorem 3.1.
> > Therefore, as long as $\delta = o(\sigma^{-2})$, the function $s_{\pi_{\sigma}}$ accurately captures the geometric information in terms of the $L^2$ error. Likewise, if $\delta = o(1)$, then $s_{\pi_{\sigma}}$ also preserves the density information.
> >   - *Conclusion:* Therefore, even in the $L^2$ sense, our central insight holds: ensuring $s_{\pi_{\sigma}}$ is accurate to $o(\sigma^{-2})$ is sufficient to learn the geometry (distance function), whereas learning the specific data density $p_{\mathrm{data}}$ requires the much stricter $o(1)$ accuracy.
> >
> > - **"No Deep Holes": Why $L^\infty$ is a Reasonable Proxy on $K$.**
> > Finally, we provide intuition on why a small Fisher divergence likely controls the error point-wise within the region of interest.
> >   - *Analysis Restricted to High-Density Set $K$:* Our convergence theorems (Theorems 4.1/5.1/5.2) rely on error bounds within a compact set $K$ containing the tubular neighborhood instead of $L^{\infty}$ bound over $\mathbb{R}^d$.
> >   - *Strictly Positive Density Lower Bound:* Inside this set $K$, the ground truth density $p_\sigma(x)$ is the convolution of a strictly positive data density $p_{\mathrm{data}}$ with a Gaussian kernel. If $K$ has radius $R$, the density is bounded below by roughly $\int p_{\mathrm{data}}(u) \exp(-R^2/(2\sigma^2)) du$.
> >   - *Effective Uniform Control:* This implies there are no "deep holes" (regions of vanishing density) inside $K$. The weighting function in the Fisher divergence, $p_\sigma(x)$, remains bounded away from zero throughout $K$. Consequently, the training objective applies a "fair" penalty to score errors at all points in $K$. While $L^2$ control does not mathematically imply $L^\infty$ control, the lack of degenerate regions suggests that assuming small $L^\infty$ error on $K$ is a plausible idealization of a model that has successfully minimized the Fisher divergence.

---

### Author Response · Authors · 2025-12-03
**Rebuttal Summary - Part 1**

Dear Area Chair,

We are pleased to provide a summary of our rebuttal, during which we incorporated new experimental validations and theoretical analyses that address the reviewers' feedback.

**Summary of Revisions**
In response to the constructive feedback, we have revised the manuscript to include:
- Convergence Analysis: A new theoretical analysis demonstrating that our Tempered Score (TS) Langevin dynamics can converge exponentially faster than standard Langevin dynamics in certain settings.
- Validated Rate Separation: A new controlled experiment on a synthetic manifold with injected score errors, empirically confirming that $O(1)$ errors disrupt distributional recovery while our method robustly recovers geometry.
- Ablation Studies: We included sensitivity analysis for the hyperparameter $\alpha$ in our image generation experiments, demonstrating robustness across a wide range of values.
- Clarifications: General improvements to clarity and presentation, including fixing typos and refining theoretical definitions to prevent misunderstandings .

Our responses to each reviewer's concerns are summarized below.

**Reviewer G9vr (score: 8)**
- *Reviewer:* Empirical results were limited and did not directly support the rate separation theorem; specifically requested synthetic experiments with known manifolds.
*Response:* We added a new experiment on a unit circle manifold with analytic ground truth scores. We demonstrated that injecting an $O(1)$ error corrupted the distribution (validating Theorem 4.1) while our TS dynamics successfully recovered the uniform manifold distribution (validating Theorem 5.1).
- *Reviewer:* The image generation results might result from an unbalanced tuning budget for $\alpha$. *Response:* We clarified that we fixed $\alpha=1$ in Table 2 (no tuning) and still outperformed the baseline. We conducted new experiments and provided a new ablation study showing robust improvements for $\alpha \ge 0.5$.
- *Reviewer:* The theory assumes continuous time and disregards discretization error.
*Response:* We explained that our error term $E_\sigma$ captures the effective score deviation of the generated distribution, thereby implicitly accounting for discretization errors.
- *Reviewer:* The $L^\infty$ assumption is stringent compared to the $L^2$ training objective.
*Response:* We argued that the rate separation is intrinsic to the score decomposition and persists in the $L^2$ sense via triangle inequality.
We also noted that $L^\infty$ is a reasonable proxy in high-density regions where models operate effectively.

**Reviewer LahT (score: 4)**
- *Reviewer:* Unclear if insights on stationary distributions apply to the dynamic reverse SDE used in generative models.
*Response:* We clarified that our analysis in Section 4 applies directly to the marginals $\pi_\sigma$ of the denoising process, instead of stationary distributions. In the follow-up discussion, we further emphasized that $\pi_\sigma$ represents the immediate output of the sampler (e.g., at time $t$) and aligns with standard theoretical frameworks in existing work; our contribution lies in revealing the rate separation phenomenon, providing insights beyond existing convergence bounds.
- *Reviewer:* TS dynamics might converge slower than standard Langevin due to reduced drift.
*Response:* We provided a proof showing the result that TS Langevin can mix exponentially faster than standard Langevin because the tempered drift mitigates metastability issues on multimodal densities on the manifold.
- *Reviewer:* Justification for uniform sampling given finite samples of $p_{\mathrm{data}}$ (regularity assumption).
*Response:* We clarified that $p_{\mathrm{data}}$ refers to the continuous population density, consistent with standard theory in the literature.
We distinguished approximation error (our focus) from statistical estimation error (future work).

---

> ### Author Response · Authors · 2025-12-03
> **Rebuttal Summary - Part 2**
>
> **Reviewer DVRd (score: 8)**
> - *Reviewer:* The $L^\infty$ assumption is undesirable given $L^2$ training losses.
> *Response:* We addressed the concern regarding the strict $L^\infty$ assumption by arguing that the core rate separation phenomenon is intrinsic to the score decomposition and persists under the $L^2$ norm typically used in training. Furthermore, our theorems only require error bounds on a high-density compact set $K$ rather than a global $L^\infty$ bound.
> In addition, we noted that even large score errors align with our findings, as geometric learning remains feasible with $o(\sigma^{-2})$ errors.
> - *Reviewer:* Quantitative rates (e.g., Wasserstein) would be stronger; is geometric learning meaningfully easier if errors are small?
> *Response:* We argued geometric learning is easier in the practical "intermediate" error regime ($o(1) < E(\sigma) \ll \sigma^{-2}$) where distribution learning fails.
> We also conjectured it has better sample complexity.
> - *Reviewer:* Replacing the data prior with a uniform prior in Bayesian Inverse Problems causes information loss.
> *Response:* We framed our result as a "negative result" warning that using the data prior requires practically much more efforts, and presented the uniform prior as a "safe harbor" or a tool for unbiasing discovery.
> - *Reviewer:* Impact of low-density regions of $p_{\mathrm{data}}$.
> *Response:* We confirmed that low-density regions increase difficulty, mathematically reflected in our remainder terms.

---

### Meta-Review · Area_Chair_nkHg · 2026-01-07

**Summary:**

The paper presents a new perspective on diffusion models, namely, given the manifold hypothesis, their success arises from implicitly learning the data manifold (geometric learning), rather than the full data distribution. Theoretical results on the rate separation (geometric learning versus distribution learning) are presented, along with numerical experiments.

Reviewers generally appreciate the novel theoretical results presented in the paper. While there are limitations on the current theoretical analysis and empirical validation, as acknowledged by the authors themselves, the results currently reported would be of sufficient interest to the community.

**Reviewer Concerns:**

Some of the concerns by the reviewers include

- The $L^{\infty}$ score error assumption is stringent compared to the $L^2$ training objective: Reviewer G9vr, Reviewer DVRd

The authors acknowledge this as a limitation in the current analysis, while arguing that the rate separation would persist under an $L^2$ assumption.

- The theoretical analysis assumes continuous time models,  disregarding discretization error: Reviewer G9vr

The authors acknowledge this as a limitation in the current analysis, while arguing that the current error analysis implicitly accounts for discretization  errors.

- Limited empirical validation:  Reviewer G9vr.
The authors reported further experimental results and agree that higher-dimensional experiments would be beneficial.

- The proposed Tempered Score (TS) Langevin dynamics might converge slower than standard Langevin: Reviewer LahT.
The author rebuttal provided a proof for the case of the unit circle showing TS Langevin can mix strictly faster than standard Langevin in the small $\sigma$ regime.

**Reviewer Scores:**

The scores are 8 4 8

I believe Reviewer LahT (score 4) would have raised their score after the author rebuttal, while the other reviewers would have kept their scores.

---

### Decision · Program_Chairs · 2026-01-26

Accept (Poster)